# Toward targeted observations of the meteorological initial state for improving the PM$_{2.5}$ forecast of a heavy haze event that occurred in the Beijing-Tianjin-Hebei region

Lichao Yang[1], Wansuo Duan[1,2], Zifa Wang[3], and Wenyi Yang[3]

[1]LASG, Institute of Atmospheric Physics, Chinese Academy of Sciences, Beijing, 100029, China

[2] Collaborative Innovation Center on Forecast and Evaluation of Meteorological Disasters (CIC-FEMD), Nanjing University of Information Science & Technology, Nanjing, 210044, China

[3]LAPC, Institute of Atmospheric Physics, Chinese Academy of Sciences, Beijing 100029, China

*Corresponding to*: Wansuo Duan (duanws@lasg.iap.ac.cn)

**Abstract.** An advanced approach of Conditional Nonlinear Optimal Perturbation (CNOP) was adopted to identify the sensitive area for targeted observations of meteorological fields associated with PM$_{2.5}$ concentration forecasts of a heavy haze event that occurred in the Beijing-Tianjin-Hebei (BTH) region, China, from 30 November to 4 December 2017. The results show that a few specific regions in the southern and northwestern directions close to the BTH region represent the sensitive areas. Numerically, when predetermined artificial observing arrays (i.e., possible "targeted observations") in the sensitive areas were assimilated, the forecast errors of PM$_{2.5}$ during the accumulation and dissipation processes were aggressively reduced; in particular, these assimilations, compared with those in other areas that have been thought of as being important for the PM$_{2.5}$ forecasts in the BTH region in previous studies, exhibited a more obvious decrease in the forecast errors of PM$_{2.5}$. Physically, the reason why these possible "targeted observations" can significantly improve the forecasting skill of PM$_{2.5}$ was interpreted by comparing relevant meteorological fields before and after assimilation. Therefore, we conclude that preferentially deploying additional observations in the sensitive areas identified by the CNOP approach can greatly improve the forecasting skill of PM$_{2.5}$, which, beyond all doubt, provides theoretical guidance for practical field observations of meteorological fields associated with PM$_{2.5}$ forecasts.

## 1 Introduction

Air pollution is one of the most severe environmental problems that China is facing. Among various air pollutants, fine particulate matter (PM$_{2.5}$) has been considered as the most serious pollutant, frequently engulfing northern China, such as the Beijing-Tianjin-Hebei (BTH) region. Exposure to heavy PM$_{2.5}$ episodes not only increases the risks of various respiratory diseases, but also induces the possibility of

diabetes and other metabolic dysfunction-related diseases (Guan et al., 2016; Lim and Thurston, 2019). Accurate $PM_{2.5}$ concentration forecasts are essential since they can remind people to reduce exposure during haze days and can assist policy-makers in making effective emission reduction measure decisions. The atmospheric chemical transport model (CTM) is one of the most widely used and effective ways to forecast $PM_{2.5}$ concentrations. However, relevant chemical and physical processes are complex, and

associated parameterization schemes of turbulent processes and meteorological and emission conditions cannot describe exactly the real world, causing model forecasts to have great uncertainty, especially on heavy haze days (Hu et al., 2010; Kong et al., 2021).

   The uncertainties of CTM output, as mentioned above, are primarily attributed to the uncertainties of meteorological and emission inputs, in addition to those occurring in the chemical model formulation

(Roman et al., 2004; Gilliam et al., 2015). Meteorological conditions including wind, temperature, and relative humidity, which are crucial for the transformation, formation, diffusion and removal of pollutants in the atmosphere, have a great impact on $PM_{2.5}$ forecasts of the BTH region in CTMs (Godowitch et al., 2011; Chen et al., 2020). Using an artificial neural network model combined with wavelet transformation, He et al. (2017) demonstrated that meteorological conditions explained more than 70% of the variance

in daily $PM_{2.5}$ concentrations over the major cities in China. Therefore, regional $PM_{2.5}$ concentrations rely on meteorological variations to a large extent. Thus, to improve the $PM_{2.5}$ forecasting skill, it is necessary to understand the sensitivity of the CTM results to the inputted meteorological fields and to reduce meteorological uncertainty. It has been demonstrated that uncertainties in the meteorological initial field substantially influence pollution simulations, including their temporal variations and peak

time concentrations (Zhang et al., 2007; Bei et al., 2017; Liu et al., 2018). Then increasing the accuracy of the meteorological initial conditions is an effective way to improve the $PM_{2.5}$ forecasting skill.

   Data assimilation is recognized as a useful technique for improving the accuracy of initial conditions. To obtain reliable initial meteorological conditions, sufficient and effective observations are essential. However, conventional observations, which are distributed at a low resolution in both oceans and islands,

have a limitation in improving the accuracy of initial conditions (Li et al., 2015). Assimilating additional field observations has been proven to be an effective way to obtain a reliable initial field (Sydney, 1996; Mu et al., 2015). Since field observations are costly and never sufficiently dense, one can consider placing a preferentially limited number of observations in key areas to have the most positive impacts on

improving forecast skill. This idea is just one of the new observational strategies of "target observation",

also called "adaptive observation", which has been developed over the past two decades (Snyder, 1996;

Palmer et al., 1998; Majumdar, 2016). The "target observation" mainly serves the demand of forecasts

on observations. The idea is as follows. To better predict an event at a future time $t_2$ (i.e., verification

time) in a focused area (i.e., verification area), additional observations are deployed at a future time

$t_1$(i.e., target time; $t_1 < t_2$) in some key areas (i.e., sensitive areas) where additional observations are

expected to have a large contribution in reducing the prediction errors in the verification area. These

additional observations are assimilated by a data assimilation system to provide a more reliable initial

state, which would be supplied to the model to obtain a more accurate prediction. Targeted observations

have become a hot topic in atmospheric science due to their successful applications in improving the

prediction skills of extreme weather events, such as typhoons (Wu et al. 2009; Mu et al., 2009), winter

storms (Kren et al., 2020), and high-impact climatic events, such as the El Niño-Southern Oscillation

(ENSO; Kramer and Dijkstra., 2013; Duan et al., 2018) and Indian Ocean Dipole (IOD; Feng et al., 2017;

Beal et al., 2020). As we stated above, the meteorological initial fields have great impacts on the $PM_{2.5}$

forecasts of the BTH region (Bei et al., 2017; Liu et al., 2018); meanwhile, our results also showed that

the $PM_{2.5}$ forecasts are sensitive to the uncertainties of meteorological initial conditions (see Section 3.1).

Based on these findings, we would propose the following question: can we apply the targeted observation

strategy to improve the meteorological condition forecasts, which then further improve the $PM_{2.5}$

forecasts of BTH region? It has also been argued that sufficient satellite observations can be used to yield

the meteorological initial field by using a data assimilation approach. However, assimilating more

observations may not lead to higher forecast benefits. Therefore, even if there are sufficient observations,

one should also consider observations in which area and how many observations should be preferentially

assimilated to improve the $PM_{2.5}$ forecast skill to a larger degree. When the observations in the area with

high sensitivity are assimilated to the initial values of the forecast, the forecasting skills will be greatly

increased; conversely, if the observations in the area where the forecast is not sensitive to the initial values

are assimilated, the forecasting skills will be improved slightly or even become worse (Yu et al., 2012;

Janjić et al., 2018; Zhang et al., 2018). Then the present study would explore the relevant sensitive area

and examine the role of possible "targeted observations" on meteorological fields in improving the $PM_{2.5}$

forecast skill during a heavy haze event that occurred from 30 November to 4 December 2017 in the

BTH region, eventually suggesting the usefulness of implementing targeted observations on meteorological fields for improving air quality forecasts.

The key for the targeted observation is the determination of sensitive areas mentioned above and the design of the observation network. That is, when implementing the targeted observations, one should first make clear where to preferentially implement targeted observations and how to display these additional observations. To obtain the sensitive areas of meteorological fields for $PM_{2.5}$ forecasting, an advanced optimization method, Conditional Nonlinear Optimal Perturbation (CNOP), is used (Mu et al.,

2003; Mu and Zhang, 2006), which overcomes the linear limitation of the traditional singular vector approach (Lorenz, 1965). The CNOP represents the initial perturbation that causes the largest error growth at a given future time over the verification area. The CNOP is therefore the most sensitive initial perturbation; therefore, it would have potential for providing the sensitive area for targeting observations. In fact, the CNOP has been adopted to identify sensitive areas for targeting observations in both

Observations System Simulation Experiments (OSSEs) and/or practical observation tasks associated with typhoons, ENSO, Kuroshio, and marine environments over the coast of China (Mu et al., 2015; Da et al., 2019) and has gained great success in improving the forecasting skills of the concerned high-impact weather or climatic events.

      In the present study, we would consider the importance of the meteorological initial conditions on

$PM_{2.5}$ forecasting and apply the targeted observation strategy of meteorological fields with the CNOP approach to study the $PM_{2.5}$ forecast of a heavy haze episode. As mentioned above, during the period from 30 November to 4 December 2017, a heavy air pollution event occurred in the BTH region, with hourly maximum $PM_{2.5}$ concentrations greater than 250 $\mu g/m^3$, exceeding the standard of severe pollution (Feng et al., 2016). However, the Beijing Municipal Ecological and Environmental Monitoring Center

did not provide a warning of this event in time (see the link http://www.bjmemc.com.cn/). We utilize this event as an example to explore the possible "targeted observations" of meteorological fields and to investigate whether they can help improve the $PM_{2.5}$ forecasting skill. Specifically, the following questions are addressed.

      (a) Which area represents the sensitive area of initial meteorological fields for targeted observations

associated with the $PM_{2.5}$ forecast of the concerned event?

      (b) What is the optimal observation array for targeted observations in meteorological fields (in terms

of locations and coverage density)?

(c) Why can the "targeted observations" in the sensitive areas lead to a larger improvement of the PM$_{2.5}$ forecasting skill of the event?

The paper is organized as follows. The model, methodology and data used in the study are introduced in the next section. Then, the CNOP-type errors of the meteorological field forecasting of the haze event are calculated in Sect. 3. In Sect. 4, the sensitive areas of the meteorological field for the PM$_{2.5}$ forecasts are identified, and relevant OSSEs are designed to verify the validity of the targeted observation in improving the forecasting skill of PM$_{2.5}$ in the haze event. In Sect. 5, the reasons why the "targeted observations" can result in a larger improvement of PM$_{2.5}$ forecasts are interpreted. Finally, a summary and discussion are presented in Sect. 6.

## 2 Model, Methodology and Data

In this study, we adopt the Nested Air Quality Prediction Modeling System (NAQPMS) and Weather Research and Forecasting (WRF) model to explore the role of targeted observations on meteorological fields in improving the surface air concentrations of PM$_{2.5}$ forecasts by building an optimization problem associated with the CNOP approach.

### 2.1 Models

The NAQPMS is a three-dimensional regional Eulerian chemical transport model developed by the Institute of Atmospheric Physics, Chinese Academy of Sciences (Wang et al., 1997; 2006). It includes modules that address horizontal and vertical advection and diffusion, dry-wet deposition, gaseous phases, aqueous phases, aerosols and heterogeneous chemical reactions. The NAQMPS has been widely applied to forecast air pollutants and to study the source apportionment of pollutants (Yang et al., 2020). The anthropogenic emissions of PM$_{2.5}$ and other pollutants are from Multi-resolution Emission Inventory for China in 2017 (MEIC 2017) (Li et al., 2014) (http://meicmodel.org/). The model integration is conducted in a single model domain of 95×95 grids at a resolution of 30 km with 20 vertical levels. The components of PM$_{2.5}$ simulation include black carbon (BC), organic carbon (OC), secondary inorganic aerosol (sulfate, nitrate, ammonium) and primary PM$_{2.5}$ emitted directly from various sources. The mass of aerosol liquid water is not included in the simulated PM$_{2.5}$ mass concentrations so that the PM$_{2.5}$ simulations are dry

mass concentrations.

The NAQPMS is driven by the meteorological field generated through WRFV3.6.1 (http://www.wrf-model.org/). The WRF model used in the present study adopts the Lin microphysics scheme (Lin et al. 1983), RRTMG longwave radiation (Iacono et al. 2008), Dudhia shortwave radiation schemes (Dudhia, 1989) and Yonsei University planetary boundary layer parameterization scheme (Hong et al. 2006). These parameterization schemes are also adopted in the adjoint model of the WRF, which is used to calculate the CNOP (see Sect. 2.2). To enhance the computing efficiency of the CNOP, a horizonal resolution of 30 km is used in the present study for an initial attempt. The model domain of the WRF and its adjoint model are the same as that in the NAQPMS. The assimilation system we used is a 3-D variational data assimilation system of the WRF, which has been proven to be an efficient assimilation tool for PM$_{2.5}$ simulations (Kumar et al., 2019; Zhang et al., 2021).

**2.2 Conditional Nonlinear Optimal Perturbation (CNOP)**

The CNOP represents the initial perturbation (or error) that can lead to the largest forecast error in the focused area (verification area) at verification time. Suppose a nonlinear model is expressed as Eq. (1),

$$\begin{cases} \frac{\partial x}{\partial t} + F(x) = 0 \\ x|_{t=0} = x_0 \end{cases}, \qquad (1)$$

where $x$ is the state vector with an initial value $x_0$ and $F$ is a nonlinear partial differential operator. The solution of Eq. (1) can be described as $x(t) = M(x_0)$, in which $M$ is the nonlinear propagator. If $x(t)$ is a reference state and an initial perturbation $\delta x_0$ is added to its initial state $x_0$, a forecast will be made with $x(t) + \delta x(t) = M(x_0 + \delta x_0)$, where $\delta x(t) = M(x_0 + \delta x_0) - M(x_0)$ represents the evolution of the initial perturbation $\delta x_0$. Then, an initial perturbation is CNOP $(\delta x_0^*)$ if and only if

$$J(\delta x_0^*) = \max_{\delta x_0^T C_1 \delta x_0 \leq \beta} [M(x_0 + \delta x_0) - M(x_0)]^T C_2 [M(x_0 + \delta x_0) - M(x_0)], \ (2)$$

where $\delta x_0^T C_1 \delta x_0 \leq \beta$ is the constraint condition that the initial perturbation should satisfy and $\beta$ is a positive value that is comparable to the initial analysis error variance of the considered variables. $C_1$ and $C_2$ are coefficient matrices, which define the amplitudes of initial perturbations $\delta x_0$ and its evolution $M(x_0 + \delta x_0) - M(x_0)$, with $x$ consisting of zonal and meridional wind ($U$ and $V$, respectively), temperature ($T$), water vapor mixing ratio ($Q$) and pressure ($P$) components in the present study, and they play their role by calculating the total perturbation energy from surface to top (i.e., 100 hPa), as in Eq. (3) (Ehrendorfer et al., 1999; Chen et al., 2020),

$$\text{Total energy} = \frac{1}{D}\int_0^1 \int_D [U'^2 + V'^2 + \frac{C_p}{T_r}T'^2 + \frac{L^2}{C_pT_r}Q'^2 + R_aT_r(\frac{P'}{P_r})^2]\, d\eta dD, \quad (3)$$

where $C_p$ (=1005.7 Jkg$^{-1}$ K$^{-1}$), $R_a$ (=287.04 Jkg$^{-1}$ K$^{-1}$), $T_r$ (=270 K), $L(= 2.5105 \times 10^6$ Jkg$^{-1}$) and $P_r$ (=1000 hPa) are constant values; and $U'$, $V'$, $T'$, $Q'$ and $P'$ denote the perturbations superimposed on meteorological fields of zonal and meridional wind, temperature, water vapor mixing ratio and pressure, respectively. $D$ denotes the verification area, which is the BJH region in this study and $\eta$ signifies the vertical coordinate.

The optimization problem in Eq. (2) is solved by using the spectral projected gradient 2 (SPG2) method (Birgin et al., 2001) in the present study. A first guess is assigned to the initial perturbation $\delta x_0$. The WRF model is integrated forward with the initial state $x_0 + \delta x_0$ to obtain the forecast $M(x_0 + \delta x_0)$. The cost function $J$ is calculated by using $M(x_0 + \delta x_0)$ and $M(x_0)$. The adjoint model of the WRF is integrated backward to calculate the gradient of the cost function with respect to the initial perturbation $\delta x_0$. The gradient represents the fastest descending direction of the cost function $J$ in Eq. (2). Based on the iteratively forward and backward integration governed by the SPG2 algorithm, the initial perturbation $\delta x_0$ is optimized and updated until the convergence condition of the algorithm is satisfied. Here, the convergence condition is $\|P(\delta x_0 - g(\delta x_0)) - \delta x_0\|_2 \leq \varepsilon_1$, where $\varepsilon_1$ is an extremely small positive number, $P(\delta x_0)$ projects the $\delta x_0$ outside the constraint to the boundary of the constraint condition and $g(\delta x_0)$ represents the gradient of the cost function $J$ with respect to $\delta x_0$. Then, the resultant initial perturbation $\delta x_0^*$ is the CNOP. The details for the SPG2 algorithm can be seen in Birgin et al. (2001).

**2.3 Data**

Surface PM$_{2.5}$ observation datasets for verification are obtained from national environmental monitoring stations. There are 1287 national stations across China, 80 of which are located in the BTH region. The distribution of the 80 observation sites within the BTH region is shown in Figure 1. We retrieved the hourly measurements of PM$_{2.5}$ from 80 air quality monitoring stations from 30 November to 4 December 2017, where the PM$_{2.5}$ observations are for dry mass concentrations and there are no missing values during the time period we considered.

The fifth generation ECMWF reanalysis for the global climate and weather (ERA5) (https://www.ecmwf.int/en/forecasts/datasets/reanalysis-datasets/era5) and National Centers for

Environmental Prediction (NCEP) GFS historical archive forecast data (GFS, https://rda.ucar.edu/datasets/ds084.1/) are both used to produce the initial and boundary meteorological conditions for the WRF simulations. Both the ERA5 and GFS data have a 0.25° spatial resolution (approximately 25 km) and 6-hour temporal resolution.

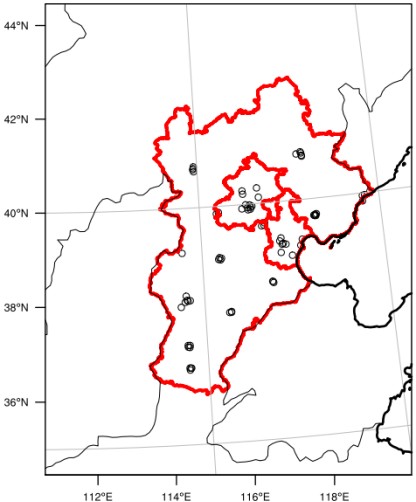

**Figure 1 The map of current environmental monitoring stations (hollow circles) within the BTH domain. The black line presents the boundary of province in China, and the thick black line presents the coastline. The boundaries of the Beijing City, Tianjin City and Hebei province are marked in red.**

## 3 The CNOP of the PM$_{2.5}$ forecasting

In this section, we use the CNOP approach to identify the sensitive areas for targeted observations associated with the PM$_{2.5}$ forecast in the heavy haze event in BTH occurred from 30 November to 4 December 2017. Figures 2(a) and 2(b) plot the time series of the PM$_{2.5}$ concentration observed at Baoding (in Hebei) and Dongsi (in Beijing) environmental monitoring stations. The haze started to develop at approximately 02:00 BJT (Beijing Time, UTC + 8 hours) on 1 December and dispersed at 14:00 BJT on 3 December. Specifically, the PM$_{2.5}$ concentrations of most cities in the BTH region exceeded 250 ug/m$^3$ at 12:00 on 2 December; then, starting from 01:00 on 3 December, the PM$_{2.5}$ dissipated rapidly within several hours. In Beijing, from 00:00 on 1 December, it took almost one day to accumulate PM$_{2.5}$ from 77 ug/m$^3$ to 160 ug/m$^3$ according to the Dongsi station; then, from 01:00 on 3 December, the PM$_{2.5}$ concentration decreased from 256 ug/m$^3$ to 19 ug/m$^3$ in 7 hours.

### 3.1 Simulations of the PM$_{2.5}$ variability in the heavy haze event

After a 10-day spin-up of the WRF-NAQPMS, the ERA5 and GFS meteorological data are separately

adopted to initialize the WRF at 00:00 BJT on 30 November 2017, and the simulations of $PM_{2.5}$

concentrations at the Baoding and Dongsi stations are plotted in Fig. 2. Since the two simulations are

generated by the same model using the same emission inventory, the $PM_{2.5}$ forecast uncertainties are only

attributed to the uncertainties of meteorological initial fields. The simulation initialized by ERA5 can

better reproduce the pollution event. During the period between 00:00 BJT on 30 November and 23:00

BJT on 1 December, the simulations initialized by ERA5 almost overlap with the observations. In the

remaining time period, although the highest $PM_{2.5}$ concentration simulated by ERA5 occurs

approximately 12 hours earlier and more than 50 ug/m$^3$ lower than those in the observations, the

simulation can represent well the accumulation and dissipation processes of $PM_{2.5}$.

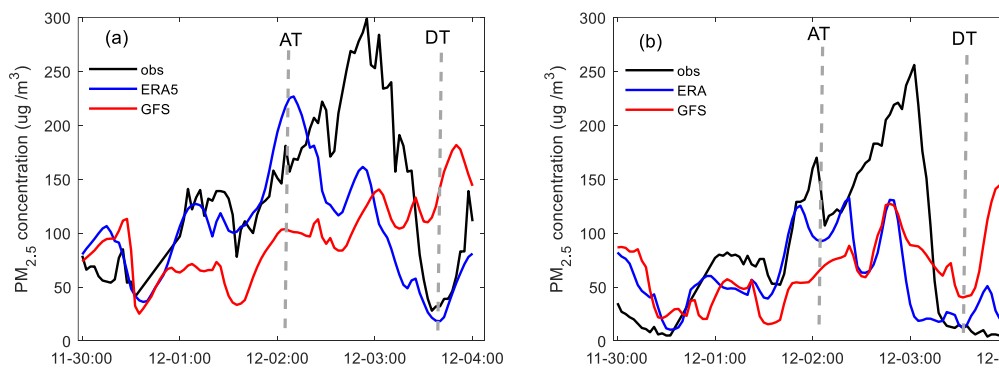

**Figure 2. Time series of the dry PM2.5 concentrations at (a) Baoding station (Hebei Province) and (b) Dongsi**
**station (Beijing city) of observations and simulations initialized by ERA5 and GFS meteorological data during**
**the period between 30 November and 4 December 2017. The Accumulation Time (AT) and Dissipation Time**
**(DT) are marked by dashed lines.**

The simulations initialized by the GFS do not perform well in representing the episode of $PM_{2.5}$.

They underestimate the $PM_{2.5}$ concentrations during the accumulation process, and the simulated highest

$PM_{2.5}$ concentration (176 μg/m$^3$) occurs at approximately 21:00 on 3 December in Baoding, which is

exactly in the dissipation process of the observed event. The simulation of Beijing $PM_{2.5}$ also shows a

large deviation from the observational $PM_{2.5}$ concentration, especially during the dissipation process.

To quantify the differences between simulations and observations, mean RMSEs and correlations

of the 80 grids during the whole event (from 00:00 BJT on 30 November to 00:00 BJT on 4 December

2017) are calculated against the observations. As shown in Table 1, the mean RMSE of the simulations

initialized by ERA5 is 60.09 ug/m$^3$ for the $PM_{2.5}$ concentration, which is 19.87% lower than that of the

GFS simulations (i.e., 74.99 ug/m$^3$). The correlation between the ERA5 simulation and the observation is 0.47 and 20.51% higher than that of GFS simulations (i.e., 0.39). More specifically, we select two time points to show the PM$_{2.5}$ differences between simulations and observations, which are at 2:00 BJT on 2 December (hereafter defined as Accumulation Time; AT) and 14:00 BJT on 3 December (hereafter defined as Dissipation Time; DT). Almost all GFS simulations show an underestimation of the PM$_{2.5}$ at the AT and an overestimation at the DT. The mean deviations are -47.88 ug/m$^3$ at the AT and 55.02 ug/m$^3$ at the DT. The ERA5 simulation performs much better at the two time points, with mean deviations of -30.57 ug/m$^3$ and 41.58 ug/m$^3$, although it also shows an underestimation at the AT and an overestimation at the DT.

**Table 1 The Root Mean Square Error (RMSE) (ug/m$^3$) and correlation coefficient (CC) of PM$_{2.5}$ concentrations between simulations initialized by ERA5 and GFS and observations averaged over 80 stations.**

| Measurements | ERA5 | GFS |
|:---:|:---:|:---:|
| RMSE | 60.09 | 74.99 |
| CC | 0.47 | 0.39 |

It is known that a bad forecast made by a numerical model is attributed to errors in both models and initial conditions. The study of targeted observations aims to improve the forecast by reducing the errors in the initial conditions, which is usually implemented with perfect model assumptions (Mu et al., 2015). A perfect model is assumed to limit forecast errors that result only from errors in the initial conditions, thus simplifying the complexity of problems. However, there are no perfect models in reality. Thus, when implementing the targeted observation tasks, we choose the model that exhibits relatively small model errors and is able to present good simulations to determine where (i.e., the sensitive area) to deploy the targeted observations by calculating the CNOP. The WRF is one of the most advanced weather-forecasting models currently and exhibits small model errors (Liu et al., 2012). Therefore, we apply the WRF, together with the NAQPMS model, to explore the role of targeted observations in PM$_{2.5}$ forecasts. When we use different initial conditions to simulate PM$_{2.5}$, a better simulation is taken as the "truth run", and the CNOP is calculated based on that. As shown above, the simulations initialized by ERA5 have better performances in presenting the PM$_{2.5}$ variability; particularly, they show the best simulation at the

AT for the accumulation process of PM2.5 and at the DT for the dissipation process. Thus, the simulations initialized by ERA5, especially at AT and DT, are taken as the "truth run" to determine the sensitive area for targeted observations by calculating the CNOP. Even though, the calculated sensitive area is actually an approximation of the real sensitive area. If such approximation is valid, then for any forecast, preferentially assimilating additional observations in the sensitive area will help improve the $PM_{2.5}$ forecasting skill greatly. The validity of the above approximate sensitive area is often tested by prescribing a good simulation to observation (for example, the simulation initialized by ERA5) and then assimilating the simulated observations located in the sensitive area to a bad forecast (for example, the control forecast) to examine whether the assimilation forecast will be much closer to the good simulation, which, actually, is a kind of OSSEs (see Masutani et al., 2010; Qin et al., 2013). In our study, to verify the validity of the sensitive area, the simulated targeted observations are assimilated to the GFS forecasts to improve their $PM_{2.5}$ forecasts, where the GFS forecasts are taken as the "control run" and those after assimilating targeted observations are regarded as the "assimilation run". If the sensitive area is valid, the $PM_{2.5}$ forecasts in the assimilation run will be much closer to the truth run. It can also be inferred that if the real observations are available, assimilating the real targeted observations to the initial field of the meteorology of the control forecast would improve the $PM_{2.5}$ forecast skill greatly against the observations. In the present study, we will adopt assimilating simulated observations to verify the validity of the sensitive area due to the lack of available observations.

**3.2 CNOP-type errors of meteorological field forecasting**

We select the AT and DT as verification times separately to determine the sensitive areas by calculating the CNOP-type errors. When the AT is taken as the verification time, we explore the forecast starting from 2:00 on 1 December, with a lead time of 24 hours, and the forecast starting from 14:00 on 1 December, with a lead time of 12 hours. When the DT is taken as the verification time, the forecasts starting from 14:00 on 2 December and 02:00 on 3 December, with lead times of 24 hours and 12 hours, respectively, are investigated. Then, there are a total of 4 $PM_{2.5}$ forecasts concerned here for the heavy haze event that occurred in the BTH region from 30 November to 4 December 2017, which are all initialized by ERA5.

As we described in Section 2.2, the CNOP-type initial errors which include the variables of wind, temperature, pressure and water vapor mixing ratio cause the largest forecast error of concerned

meteorological fields measured by the total energy at the verification time in the verification area, which

may perturb the $PM_{2.5}$ forecast to the greatest extent when considering the combined effect of different

meteorological components and thus represent the most disturbing initial error of the meteorological field.

The CNOP-type errors are calculated separately for these 4 forecasts. Figures 3-6 plot the horizontal

structures of the CNOP-type errors (including wind, temperature and water vapor perturbations) at

ground level (approximately 1000 hPa), low level (approximately 850 hPa and 750 hPa), middle level

(approximately 500 hPa) and upper level (approximately 200 hPa) for the 4 forecasts. All wind,

temperature and water vapor components of the CNOP-type errors, either for the AT or DT, are mainly

concentrated at ground and low levels, with large errors lying at the low level for a lead time of 24 hours

and ground level for a lead time of 12 hours.

When it is the CNOP-type errors for the AT, their dominant anomalies, as mentioned above, occur

at the low level (i.e., 850 hPa) for the forecast with a lead time of 24 hours; furthermore, the horizontal

pattern mainly presents two areas that cover the large CNOP-type errors despite small position

differences among the respective large-error areas of wind, temperature and water vapor components at

the 850 hPa level (see Fig. 3). One area is near the southern part of the BTH region, with southerly wind,

positive temperature and water vapor biases, while the other area is in central Mongolia, with southerly

wind, positive temperature and negative water vapor biases. However, at ground level, the horizontal

patterns present different areas with large errors for the three meteorological components: the wind

presents large errors in the southern and western parts of the BTH region, while the temperature and

water vapor present large errors in the western part of the BTH region. For the forecast with a lead time

of 12 hours, the CNOP-type errors are dominant at ground level but mainly confined in Beijing city, with

large northerly wind and positive temperature and water vapor biases (see Fig. 4). In addition, the wind

and water vapor also present large errors in Shandong Province; at the low level (i.e., 850 hPa), the

maximum errors of wind and temperature are located in the northwestern part of the BTH region, near

the region of Nart, but the maximum error of water vapor is found in Shandong Province in the

southeastern part of the BTH region.

When the DT is the verification time, it can be seen that the CNOP-type errors mainly occur at the

low level (i.e., 850 hPa and 750 hPa) for a lead time of 24 hours and large northerly wind, negative

temperature and water vapor biases occur in southern Mongolia despite their specific positions having

small differences, with the location of large water vapor errors further west to that of the large errors of wind and temperature (see Fig. 5). For a lead time of 12 hours, the large northwesterly wind errors are concentrated at the ground level, while the large positive temperature and water vapor errors occur at the low level; furthermore, there are also large temperature and water vapor errors occurring at the low and middle levels (see Fig. 6).

It is clear that the CNOP-type errors peak at different vertical levels for the 4 forecasts. Even for the meteorological fields of wind, temperature, and water vapor, even at the same vertical level, the areas with large errors of different variables are somewhat different. The errors in the areas where the CNOP-type errors are concentrated could make the largest contribution to the forecast errors of the verification area at the verification time and therefore can be regarded as a sensitive area for targeted observations associated with $PM_{2.5}$ forecasts. However, from the above CNOP-type errors, it is known that such areas are dependent on different meteorological variables and are located at different vertical levels and regions, which therefore confuses us which meteorological variables, levels and areas should be identified to be preferentially observed and provides challenges to real field campaigns. Then, in this situation, how do we address the problems related to targeted observations for the meteorological fields associated with $PM_{2.5}$ forecasting? We will address this question in the next section.

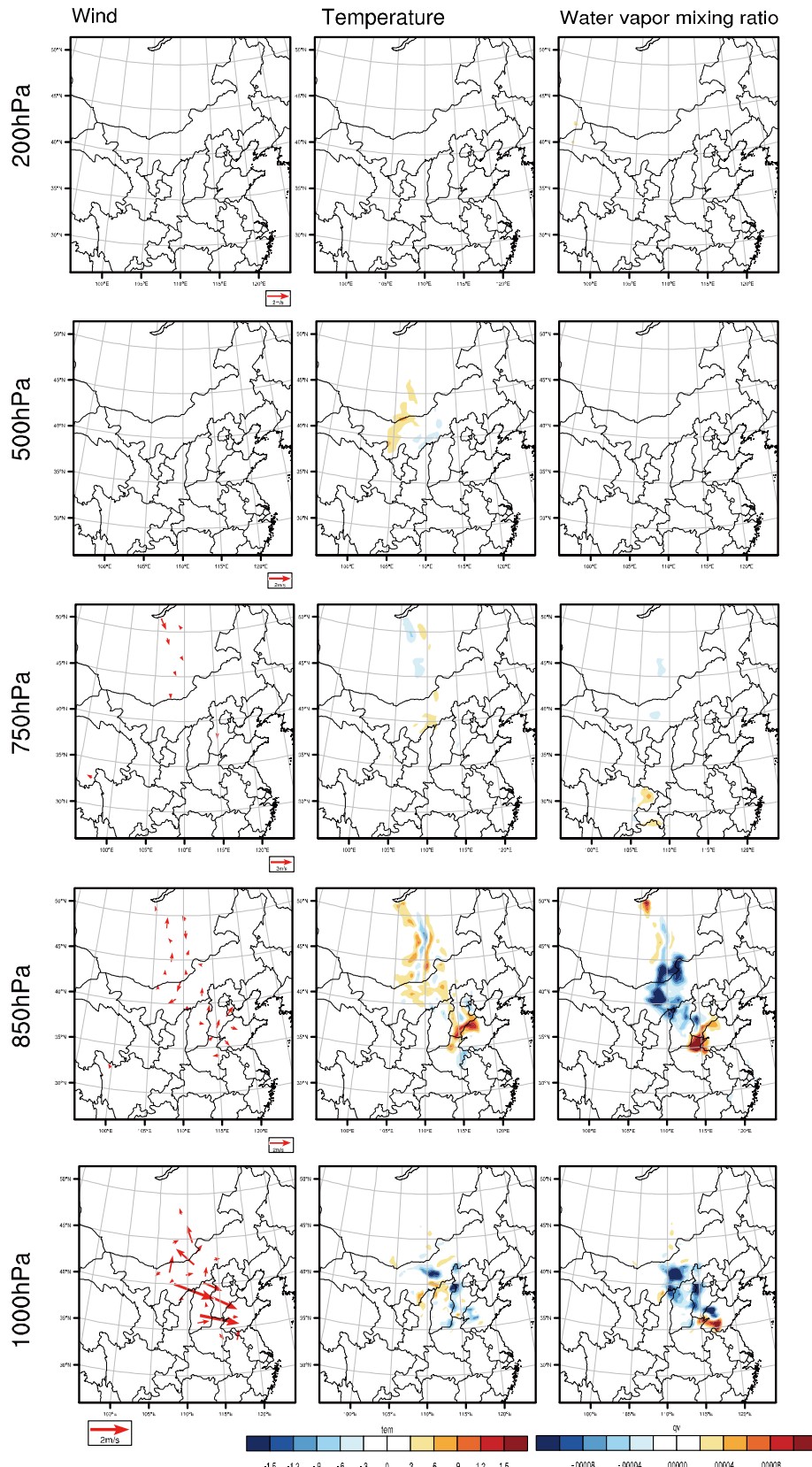

**Figure 3. The horizontal distribution of the CNOP-type errors including wind component (vector, left column, unit: m/s), temperature component (shaded, middle column, unit: ℃) and water vapor mixing ratio component (shaded, right column, unit: kg/kg) at an upper pressure level (approximately 200 hPa), middle**

**pressure level (approximately 500 hPa), low pressure level (approximately 850 hPa) and ground level (approximately 1000 hPa) for the forecast starting from 2:00 on 1st December, with a lead time of 24 hours.**

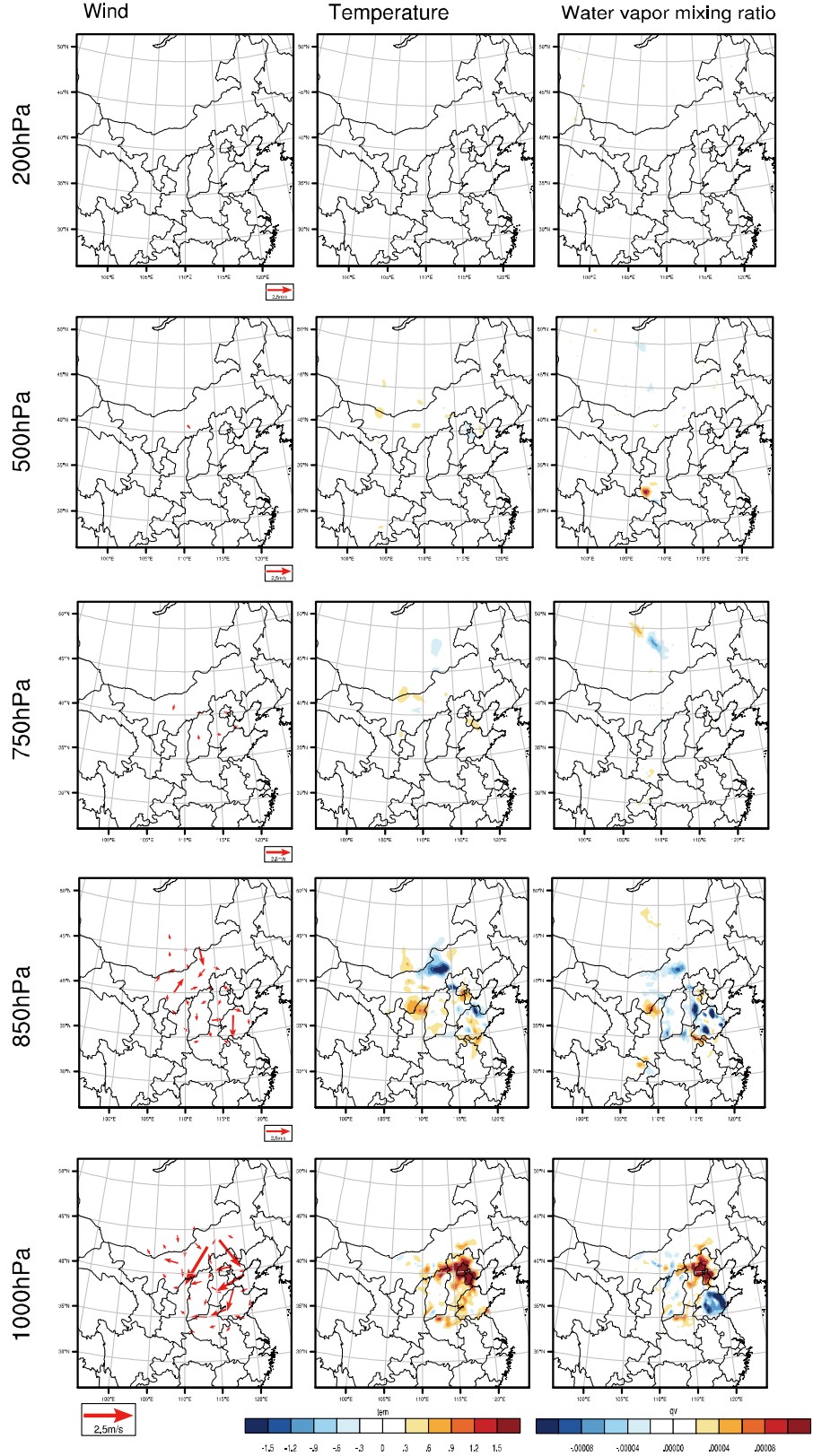

**Figure 4. The same as in Figure 3, but for the forecast starting from 14:00 on 1st December, with a lead time of 12 hours.**

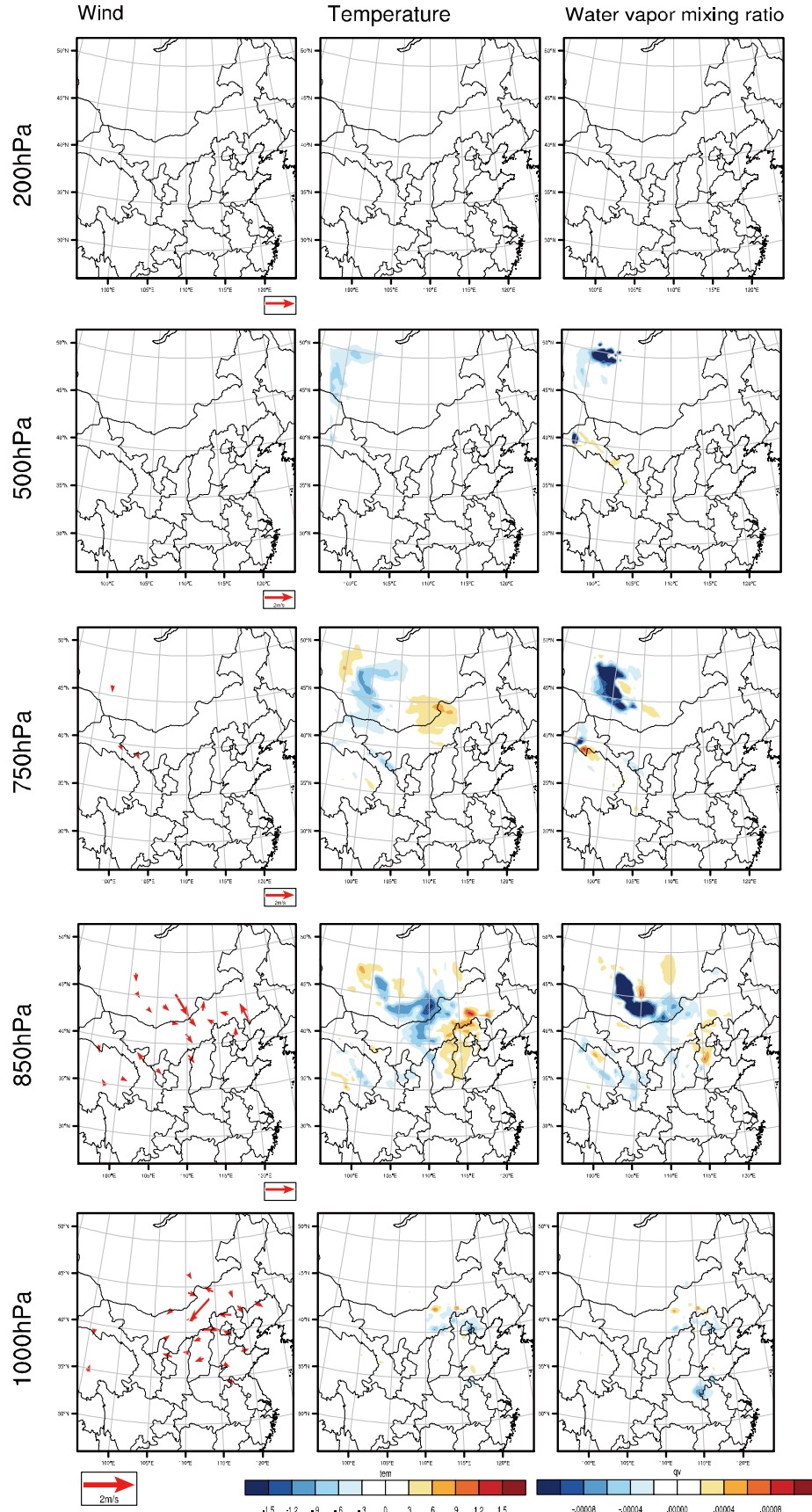

**Figure 5. The same as in Figure 3, but for the forecast starting from 14:00 on 2ⁿᵈ December, with a lead time of 24 hours.**

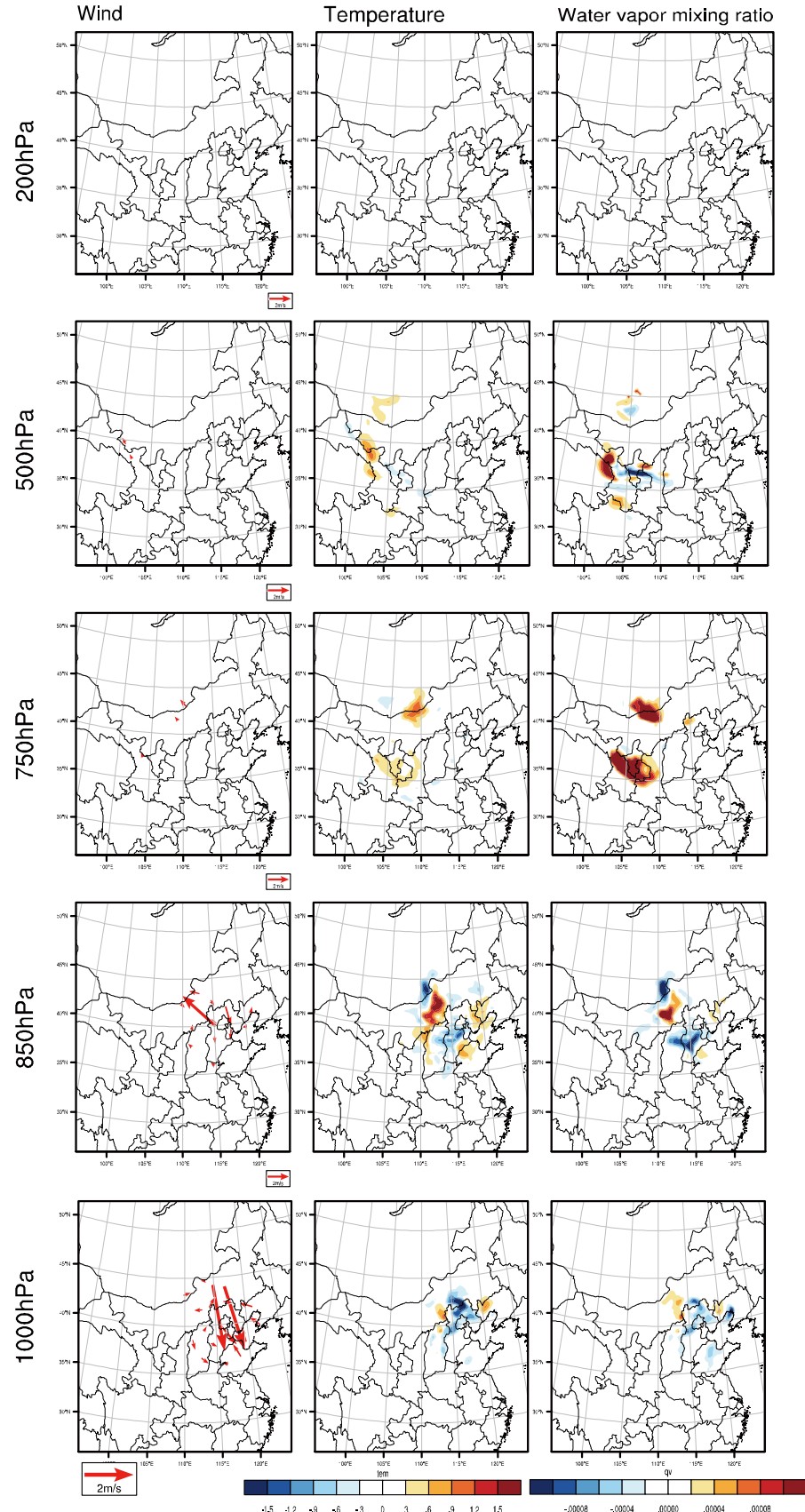

**Figure 6. The same as in Figure 3, but for the forecast starting from 2:00 on 3$^{rd}$ December, with a lead time of 12 hours.**

**4. The sensitive area for targeted observations and associated validity verification on improving the PM₂.₅ forecasts**

In this section, we propose an approach to measure the comprehensive sensitivity of initial errors occurring in different vertical levels and horizontal areas for different meteorological variables. Then, the sensitive areas for targeted observations can be identified by this comprehensive sensitivity that considers the information of all meteorological variables at all pressure levels.

**4.1 The sensitive areas for targeted observations associated with PM₂.₅ forecasts**

To evaluate the comprehensive sensitivity of the CNOP-type initial errors occurring at different vertical levels and areas for different meteorological fields, a vertical integral (VI) of the CNOP-type errors, as in Eq. (4), is calculated.

$$\text{VI} = \int_0^1 \frac{1}{2}\left(U'^2 + V'^2 + \frac{C_p}{T_r}T'^2 + \frac{L^2}{C_p T_r}Q'^2 + R_a T_r (\frac{P'}{P_r})^2\right) d\eta. \qquad (4)$$

The VI consists of all concerned meteorological variables and their vertical distributions and measures the comprehensive sensitivity of forecasting uncertainties on initial errors of different meteorological variables. In this situation, the PM₂.₅ forecast could be very sensitive to the combined effect of initial errors of the meteorological fields in the area of a larger VI, and preferentially reducing the meteorological initial errors in these sensitive areas will lead to much larger improvements of the meteorological forecasts over the BTH region, which then significantly improves the regional PM₂.₅ forecasts.

Figure 7 shows the horizontal distribution of the VI for the 4 forecasts. When the AT is the verification time, two areas are identified to have large VIs for the forecast starting from 2:00 on 1 December, with a lead time of 24 hours. One area is near Dezhou city, which lies to the southeast of Hebei Province; the other area is located in central Inner Mongolia, extended to Mongolia. Then, we regard these two areas as the sensitive areas for meteorological field forecasting and then we regard the PM₂.₅ forecast of the BTH region at the AT with a lead time of 24 hours. Similarly, we identify the sensitive area for the forecast with a lead time of 12 hours in Beijing and Tianjin cities. For the verification time DT, the sensitive areas are determined as the region from Huhhot in Inner Mongolia to the Altai Mountains in Mongolia for a lead time of 24 hours. For a lead time of 12 hours, the sensitive areas are mainly located in Zhangjiakou and Chengde cities, which lie in the northern part of the BTH

region.

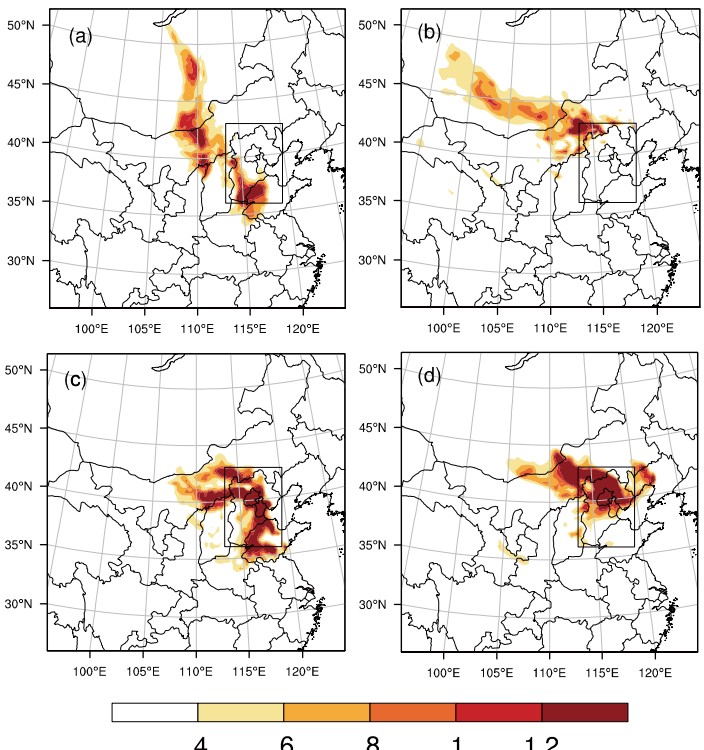

**Figure 7. The horizontal distribution of the VI (unit: J kg$^{-1}$) for the forecasts at the AT with lead times of (a) 24 hours and (c) 12 hours and for the forecasts at the DT with lead times of (b) 24 hours and (d) 12 hours. The black rectangle is the verification area.**

## 4.2 Validity of "targeted observations" in improving PM$_{2.5}$- forecasting skill

According to the definition of targeted observations, deploying additional observations in the sensitive areas and assimilating them to the initial field will improve the forecasting skill of the meteorological field and then the PM$_{2.5}$. If such improvement is significantly larger than those of assimilating the additional observations in other areas, the sensitivity of the targeted observations in the sensitive area determined by the CNOP is confirmed numerically. With the above argument, the better simulation of PM$_{2.5}$ with the meteorological field forecast by ERA5 is assumed to be the "truth run", and the worse simulation initialized by the GFS is the "control run" (see Sect. 3.1); thus, the differences between the PM$_{2.5}$ concentrations in the control and truth runs can be regarded as forecast errors of the control run with respect to the truth run. Figure 8 shows the spatial distributions of forecast errors of PM$_{2.5}$ at the AT and DT. This shows that the control run has an obvious underestimation of the PM$_{2.5}$ concentrations over the whole BTH region at the AT and an overestimation at the DT. If taking the absolute value of the biases, then the mean biases of the whole BTH region are 34.22 and 64.13 ug/m$^3$ at the AT and DT, respectively. To verify the validity of the targeted observations in the sensitive areas, we

take relevant meteorological fields in the truth run but confine them to the above identified sensitive areas as "additional observations" (i.e., artificial "targeted observations") and assimilate them to the initial fields of the control run by the 3D-Var assimilation system of the WRF (see Sect. 2.1), finally obtaining an updated forecast of the PM$_{2.5}$ concentration, which, as defined in Sect. 3.1, is called the "assimilation run". The validity of targeted observations in improving PM$_{2.5}$ forecasts of the control run is quantified by two indices defined by Eqs. (5) and (6),

$$\text{AE}_\text{V} = (\frac{|P_C - P_T| - |P_A - P_T|}{|P_C - P_T|})_{t=T} \times 100\%, \quad (5)$$

$$\text{AE}_\text{M} = \frac{1}{T}\sum_{i=t_0}^{i=T}(\frac{|P_C - P_T| - |P_A - P_T|}{|P_C - P_T|})_{t=i} \times 100\%, \quad (6)$$

where $\text{AE}_\text{V}$ and $\text{AE}_\text{M}$ are the percent change of the forecast errors at verification times [see Eq. (5)] and that during the whole forecast period [see Eq. (6)] after assimilating the control forecast, respectively; and $P_C, P_T,$ and $P_A$ denote the PM$_{2.5}$ concentration in the control run, truth run and assimilation run, respectively. The sign $|\cdot|$ measures the amplitude of forecast errors averaged over the BTH region, $T$ represents the verification time and $t_0$ is the initial time of the forecast. A positive value of AE$_V$ and AE$_M$ indicates an improvement in forecast skills, and the larger the positive values are, the more significant the improvements. A negative value of AE$_V$ and AE$_M$ indicates a decline in forecast skills.

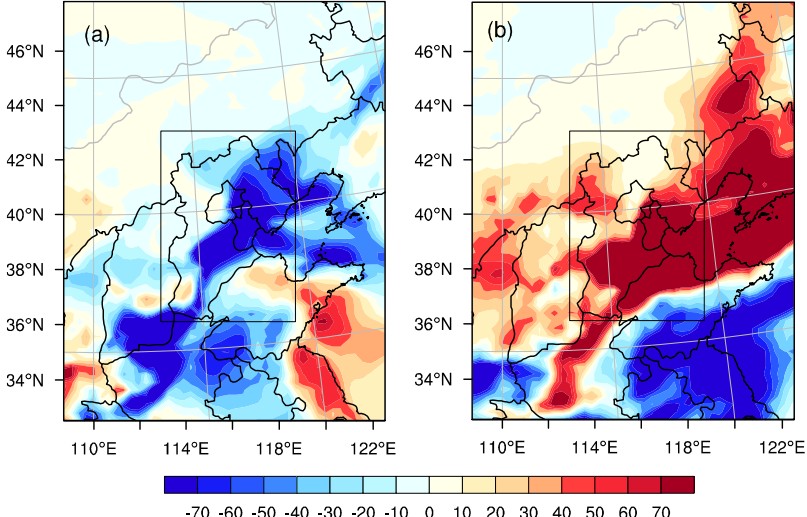

**Figure 8. The spatial distributions of PM$_{2.5}$ forecast errors (unit: $\mu\text{g/m}^3$) in the control run at the (a) AT and (b) DT. The black rectangle is the verification area.**

We take the artificial "additional observations" of meteorological fields located at a fixed number of 15 horizontal observation positions, which are located through the vertical 950, 850, 750 and 500 hPa levels (60 observations at the four pressure levels in total) and include horizontal wind, temperature, and

relative humidity; in particular, these observation positions are considered to be covered by the sensitive areas identified by the VI of the CNOP-type errors. To determine the optimal observation array in the sensitive areas, "additional observations" are experimentally distributed every 30, 60, 90, 120 and 150 km. Specifically, we take the observation distance of 150 km as an example. The grid point with the largest VI is taken as the first observation position. Then, we exclude the grids that are no further than 150 km away from the first observation position and determine one of the largest VIs among the remaining grids as the second observation position. After the second observation position is fixed, we exclude the grids that are no further than 150 km away from the second observation position, and the grid of the largest VI among the remaining grids is determined as the third observation position. The other 12 observation positions can be similarly determined. Note that the fixed number 15 of the observation positions is experimentally selected, and one can choose other numbers to conduct experiments. In accordance with the above approach, we can obtain five observation arrays with 15 predetermined observation positions.

By assimilating the five observation arrays to the initial fields of control runs, new forecasts (i.e., the assimilation runs) of $PM_{2.5}$ are obtained. The improvements of the forecasting skills against the truth runs are shown in Tables 2 and 3. For a 24-hour lead time of the forecast at the AT, assimilating the five observation arrays can improve the $PM_{2.5}$ forecast skill by reducing the forecast errors ranging from 4.29 ug/m$^3$ to 6.91 ug/m$^3$, accounting for 12.54% to 20.20% of the forecast errors in control runs measured by $AE_V$ at the AT; the mean forecast errors during the whole forecast period can decrease from 19.79% to 29.20% measured by $AE_M$ (exactly from 3.58 ug/m$^3$ to 5.28 ug/m$^3$) (Table 2). Of the five observation arrays, the array with observation positions every 90 km shows the largest improvement measured by $AE_V$ and $AE_M$. When the 15 observation positions are deployed every 90 km, approximately 68% of the grids over the BJH region show positive $AE_V$ values, and the largest improvement in $PM_{2.5}$ forecasts reaches 73.80 ug/m$^3$, located in Cangzhou city, southeastern Hebei Province (Fig. 9a). When the observation arrays are deployed 12 hours before the AT, a larger improvement in forecasting skills can be found (Table 2). Of the five observation arrays, the improvements in forecasting skills at the AT measured by $AE_V$ range from 24.53% to 43.26%, and the mean improvement during the whole forecast period measured by the $AE_M$ ranges from 32.84% to 50.81%, where the observation array deployed at a distance of 150 km shows the largest improvements in terms of both $AE_V$ and $AE_M$ despite the

observations being relatively sparse in this array. Overall, the observations deployed 12 hours before the AT in the sensitive areas identified by the CNOP-type errors measured by the VI show better performances than those deployed 24 hours before the AT. Thus, if we care about improving the $PM_{2.5}$ forecast at the AT and the number of observation positions is fixed at 15 (only accounting for 0.17% of the grids over the domain), the observation array with an observation position distance of 150 km deployed in the sensitive areas (i.e., locations in Beijing and Tianjin cities) at 12 hours before the AT might be the optimal choice for targeted observations; in this case, the forecast error of $PM_{2.5}$ could decrease by as much as 43.26% at the AT in terms of the $AE_V$ and 50.81% during the whole forecast period in terms of the $AE_M$ (see also Table 2).

Table 2 The $AE_V$/$AE_M$ of the forecasts at the AT with lead times of 24 and 12 hours, when the additional observations in the sensitive region (CNOP), Region-W and Region-N are assimilated (unit: %). The respective optimal observation array is marked in bold.

| Lead times | Region | 30 km | 60 km | 90 km | 120 km | 150 km |
|---|---|---|---|---|---|---|
| 24 hour | CNOP | 12.54/19.79 | 17.52/24.83 | **20.20/29.20** | 17.12/26.60 | 15.02/25.44 |
| | Region-W | 3.16/5.12 | 6.51/8.61 | **7.60/11.30** | 5.46/9.42 | 5.13/8.22 |
| | Region-N | -2.03/0.78 | **-0.76/2.45** | -0.79/2.34 | -1.73/1.09 | -5.70/-3.83 |
| 12 hour | CNOP | 24.53/32.84 | 32.48/37.43 | 38.79/46.31 | 42.66/50.73 | **43.26/50.81** |
| | Region-W | 15.14/18.39 | 11.52/13.11 | 11.18/13.42 | 14.95/16.13 | **17.61/18.71** |
| | Region-N | **3.67/7.32** | -2.88/-0.30 | 0.37/2.82 | -0.95/1.73 | -1.84/0.46 |

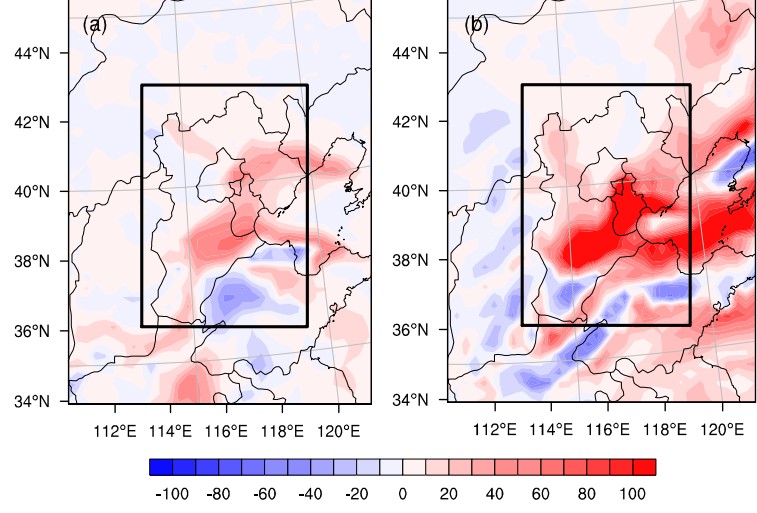

Figure 9. The spatial distributions of the improvement in $PM_{2.5}$ forecasts (unit: $\mu g/m^3$) at the (a) AT and (b)

**DT with a lead time of 24 hours. The black rectangle is the verification area.**

To improve the $PM_{2.5}$ forecast at the DT, five observation arrays in the corresponding sensitive areas can be similarly obtained, and of these arrays, their assimilation runs improve the $PM_{2.5}$ forecast skills with the $AE_V$ varying from 20.87% to 44.72% (exactly from 13.39 to 28.77 ug/m$^3$) and the $AE_M$ varying from 27.31% to 40.83% (exactly from 8.27 to 11.90 ug/m$^3$; Table 3) for a lead time of 24 hours. The assimilation run with the observation array of the observation positions every 150 km shows the largest improvement in both $AE_V$ and $AE_M$. Specifically, when the observation arrays are deployed every 150 km, an area of approximately 81% of the grids over the BJH region shows positive $AE_V$ values, and the largest improvement in the $PM_{2.5}$ forecast, reaching 202.64 ug/m$^3$, occurs in Tianjin city (Fig. 9b). However, when the lead time is reduced to 12 hours, the mean improvements are less than the forecast with a lead time of 24 hours, with the $AE_V$ varying from 20.92% to 31.01% (exactly from 11.24 to 16.66 ug/m$^3$) and $AE_M$ varying from 27.81% to 40.00% (exactly from 6.95 to 10.00 ug/m$^3$, Table 3). Among the 5 observation arrays, the observations with an observation position distance of 90 km show the largest improvement in both $AE_V$ and $AE_M$, which is different from the optimal observation array of observation positions every 150 km deployed 24 hours before the DT. In contrast, the last array has the worst performance. Overall, if we care about improving the $PM_{2.5}$ forecast skills at the DT, the optimal observation arrays should be deployed over the sensitive areas (i.e., locations in Mongolia) with an observation position distance of 150 km 24 hours before the DT, and assimilating the observations could reduce the forecast errors by as much as 44.72% at the DT measured by $AE_V$ and 40.83% during the forecast period measured by the $AE_M$. All these results are also summarized in Table 3.

**Table 3 The same as in Table 2, but for the forecast at the DT.**

| Lead times | Region | 30 km | 60 km | 90 km | 120 km | 150 km |
|---|---|---|---|---|---|---|
| 24 hour | CNOP | 20.87/27.31 | 30.69/34.28 | 34.90/35.79 | 36.89/37.20 | **44.72/40.83** |
| | Region-W | 20.49/14.75 | **22.01/16.93** | 18.18/11.23 | 17.00/10.94 | 15.74/9.54 |
| | Region-N | -0.60/**-0.49** | -0.92/-0.80 | -0.25/-0.91 | -0.50/-3.43 | **-0.15**/-2.48 |
| 12 hour | CNOP | 26.78/35.44 | 23.62/31.72 | **31.01/40.00** | 23.49/32.60 | 20.92/27.81 |
| | Region-W | -0.45/-1.16 | -1.49/-2.86 | **4.83/2.62** | 1.09/-0.71 | 1.81/0.73 |
| | Region-N | **15.07/16.64** | 13.77/15.00 | 14.11/15.74 | 14.68/16.39 | 12.52/15.51 |

Through a series of OSSEs, the effectiveness of targeted observation is conducted by deploying a

fixed number of observations (15 horizontal grids through 4 pressure levels) and observations deployed at different distances are evaluated to determine the optimal observation array. The results shows that when the observation number is fixed, an appropriate observing distance, not necessarily a large observing distance, is essential to obtain the largest improvement of $PM_{2.5}$ forecast skills. To further examine the role of appropriate observing distance, we also conducted the following experiments that observations are deployed within a limited area with different observing distances (which corresponds to different observation numbers in the limited area). Specifically, we first select a number of 120 most sensitive grids as the sensitive area in each of the four forecasts according to the VI value. Within the given size of the sensitive area, the observing arrays with the distance of 30, 60, 90, 120 and 150km are determined, with the same method as the experiments described above. The additional observations are assimilated to the control run and the improvements of $PM_{2.5}$ forecast skills are shown in Figure 10. For the two forecasts at the AT and the forecast at the DT with the lead time of 24 hours, the observation arrays with a distance of 30km shows the largest improvement in both $AE_V/AE_M$. It implies that in the given size of sensitive area, denser observation sites can better resolve the synoptic initial conditions within the sensitive area, which in turn enhance the forecasting skills more effectively. However, for the forecast at the DT with lead time of 12 hour, the observations with the distance of 90km show the largest improvement. It implies that in this forecast, it is not necessarily much denser observation locations but an appropriate one that is much important for improving the $PM_{2.5}$ forecasts. Thus, it emphasized that the observations deployed at a large distance or a high density, will not necessarily result in the largest improvement of PM2.5 forecast skills. It is suggested that, when we implement the field campaigns, the observations should be deployed carefully with an appropriate distance to get the largest benefits.

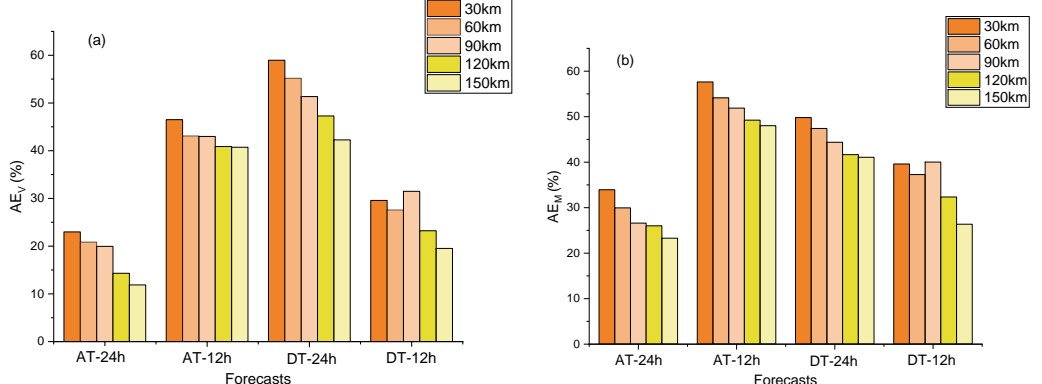

**Figure 10 The bar plots of (a)$AE_V$ and (b)$AE_M$ values of the four forecasts, when the additional observations are deployed within a limited size of area with different observing distances**

**4.3 A comparison between targeted observations and other additional observations in improving PM$_{2.5}$ forecasts**

The results in Sect. 3.2 show that assimilating targeted observations in the sensitive areas determined by the CNOP-type errors can largely improve the PM$_{2.5}$ forecasting skills (hereafter CNOP-EXPs). To further illustrate the usefulness of CNOP in identifying the sensitive area for targeted observations, in this section, we compare the sensitive areas and other areas surrounding the BTH region.

Apart from the sensitive areas identified by CNOP-type errors, other areas surrounding the BTH region are mainly located in the southwestern, southeastern, eastern and northern parts of the BTH region. Previous studies demonstrated that the PM$_{2.5}$ concentrations in the BTH region are continuously influenced by weather conditions (especially wind anomalies) in the southwestern and northern parts of the BTH region (Sun et al., 2019; Zhang et al., 2018). Specifically, they showed that southwesterly wind anomalies tend to transport the polluted air from the southwestern part to the BTH region and that northerly wind anomalies blow away BTH pollution. It therefore seems that the PM$_{2.5}$ forecasts are more sensitive to the meteorological conditions along the southwestern (i.e., Shanxi province) and northern (i.e., Inner Mongolia province) directions of the BTH region. To examine this sensitivity, we select two areas in these two directions, which are similar to the sensitive areas identified by the CNOP-type errors and surround the BTH region. Specifically, we refer to these two areas as Region-W (100.5-113.5$^{o}$E, 29.5-36.0$^{o}$N) and Region-N (115.5-126.0$^{o}$E, 42.5-51.0$^{o}$N), whose area sizes are approximately the same as those of the sensitive areas identified by the CNOP-type errors. In each region, we calculate the initial errors of meteorological conditions that lead to the largest forecast error at the verification time in the BTH region, which represents the most sensitive initial errors in this area to PM$_{2.5}$ forecasts. The algorithms are the same as in calculating the CNOP-type errors, but the initial perturbations are restricted to only Region-W and Region-N. We also use the vertical integral of the errors (VI) to determine the observation arrays and evaluate the sensitivity of PM$_{2.5}$ forecasting uncertainties to the meteorological initial errors over these two regions. Specifically, the observation arrays in these two areas are constructed with the same configuration as in the area identified by CNOP-type errors. Then, five observation arrays are similarly obtained for Region-W and Region-N. Two groups of experiments are implemented separately for the abovementioned 4 forecasts, i.e., the forecasts aimed at the AT with lead times of 12 and 24 hours and those aimed at DT with lead times of 12 and 24 hours.

The results are shown in Tables 2 and 3. For the 24-hour lead time forecast at the AT, the five observation arrays in Region-W are assimilated, and they can improve the $PM_{2.5}$ forecast skill of the BTH region with an improved $AE_V$ ranging from 3.16% to 7.60% and $AE_M$ ranging from 5.12% to 11.30% (see Table 2). These improvements measured by $AE_V$ and $AE_M$ are approximately one-third of those in CNOP-EXPs on average for the five observation array assimilations, with the former being 5.57% and 16.48% and the latter being 8.53% and 25.17% for $AE_V$ and $AE_M$, respectively. In particular, although the observation array with a distance of 90 km has the best performance for the improvements in the $PM_{2.5}$ forecasts in Region-W, this improvement is still lower than that of the worst one among the forecasts with the five observation arrays in CNOP-EXPs. When the five observation arrays are deployed over Region-N and assimilated to forecast the $PM_{2.5}$ in the control run, the $AE_V$ values at the AT are all negative for a lead time of 24 hours, which indicates a decline in the forecasting skills for the $PM_{2.5}$ at the AT compared with the control run, whichever observation array is assimilated. For the mean of the forecast skill during the whole forecast period (as measured by $AE_M$), the observation array with an adjacent distance of 150 km presents a negative value of $AE_M$ when it is assimilated to forecast $PM_{2.5}$, while the other four observation arrays present a positive value of $AE_M$, but with the mean improvement being only 1.67%, far less than 25.17% in CNOP-EXPs. It is reasonable that assimilating observations in the Region-N may result in a worse forecast. Theoretically, if the observations in the area where the forecast is not sensitive to the initial values are assimilated, the forecasting skills will be improved slightly or neutral. However, when implementing the realistic prediction, the imperfect procedure of data assimilation, the observation errors, model errors, the unresolved scales and processes in the model and other combined effects may induce additional errors (Janjić et al., 2018), which may cause the fact that assimilating observations in the unsensitive area results in a worse forecast. That also indicates that the Region-N is not the sensitive area for the forecast at the AT. For the 12-hour lead time $PM_{2.5}$ forecast at the AT, we also show that the five observation arrays in Region-W and Region-N present far fewer improvements in $PM_{2.5}$ forecast skills than those in CNOP-EXPs when they are assimilated to forecast $PM_{2.5}$ (see Table 2). Specifically, the improvements measured by the $AE_V$ averaged for the five observation arrays in Region-W and Region-N (i.e., 14.08% and -0.33%, respectively) are approximately one-third and one hundredth of that (i.e., 36.34%) in CNOP-EXPs, and the improvements measured by $AE_M$ (i.e., 15.92% and 2.41%, respectively) are approximately one-third and one-twentieth of that (i.e., 43.62%) in CNOP-

575 EXPs, respectively. From the above experiments, it is obvious that, for the 24- and 12-hour lead time forecasts at the AT, the five observation arrays deployed in Region-W (Region–N), although they often enhance the forecast skill of $PM_{2.5}$ against the control run, present amplitudes of improvement in the $PM_{2.5}$ forecast skill significantly smaller than those in the CNOP-EXPs. This shows that the sensitive areas for targeted observations of meteorological fields associated with the $PM_{2.5}$ forecast at the AT are 580 most likely to be the ones identified by the CNOP-type errors, rather than Region-W and Region-N.

For the $PM_{2.5}$ forecasts at the DT, the results also illustrate the strong sensitivity of the targeted observations in the sensitive area identified by the CNOP-type errors. Specifically, for the 24-hour lead time forecast, the observation arrays in Region-W tend to benefit the $PM_{2.5}$ forecast, and the improvement averaged for five observation arrays is 18.68% for the $AE_V$ and 12.68% for the $AE_M$, which are both 585 nearly half of those in the CNOP-EXPs; when the five observation arrays are deployed in Region-N, they all lead to worse forecasts at the DT than the control run, with the $AE_V$ varying from -0.92% to -0.15% and the $AE_M$ from -3.43% to -0.49% (Table 3). For the 12-hour lead time forecasts, the five observation arrays deployed in Region-W do not significantly improve the $PM_{2.5}$ forecast, with $AE_V$ values ranging from -0.45% to 4.83% and $AE_M$ values ranging from -2.86% to 2.62%; in contrast, the five observation 590 arrays deployed in Region-N considerably improve the $PM_{2.5}$ forecasts, with $AE_V$ ranging from $1_{2.5}2$% to 15.07% and $AE_M$ ranging from 15.00% to 16.64%, where the observation array with an adjacent distance of 30 km shows the best performance of the 5 observation arrays for improving the $PM_{2.5}$ forecast skill. Despite this, the improvement is still less than that of the worst forecast in CNOP-EXPs with the observation array with an adjacent distance of 150 km. Specifically, the improvements in $AE_V$ 595 and $AE_M$ are 14.03% and 15.85%, respectively, which are both averaged for 5 observation arrays and approximately 50% lower than those in CNOP-EXPs. Therefore, the sensitive areas for targeted observation of meteorological fields associated with the $PM_{2.5}$ forecast at the DT are the ones identified by the CNOP-type errors, i.e., the areas from Huhhot in Inner Mongolia to the Altai Mountains in Mongolia for a lead time of 24 hours, and Zhangjiakou and Chengde cities, which lie in the northern part 600 of the BTH region for a lead time of 12 hours.

**5 Interpretation**

In this section, we further interpret why the sensitive area identified by CNOP-type errors can result in a

larger improvement of PM$_{2.5}$ forecast skill. It is known that dynamic and thermodynamic conditions are two key factors that determine the transport and deposition of pollution. With a relatively strong wind, pollution could be transported to the downwind region in a short time, while a relatively clam wind could favor ground pollution accumulation. For the BTH region, northerly winds blow away PM$_{2.5}$, while southerly winds lead to the accumulation of PM$_{2.5}$ through the blocking effect of the surrounding mountains (Zhao et al., 2009). Thermodynamic conditions such as the strong temperature inversions in the atmospheric boundary layer are also favorable for the accumulation of air pollutants to form air pollution events (Miao et al., 2015). Moreover, an increased temperature may accelerate the production rate of precursors and secondary pollutants, which contribute to variations in ground-level PM$_{2.5}$.

In this paper, we showed that the "control run" either with a lead time of 12 hours or 24 hours presents a severe underestimation of PM$_{2.5}$ at the AT, and a large overestimation of PM$_{2.5}$ at the DT for the heavy air pollution event that occurred from 30 November to 4 December 2017 (see Sect. 3.1). The assimilation runs greatly promote the skill of these PM$_{2.5}$ forecasts by assimilating the targeted observations in the sensitive areas of the meteorological fields. Now, we interpret why the assimilation runs increase the PM$_{2.5}$ forecast skill for dynamic and thermodynamic reasons. After we compare the forecast biases of the control run with lead times of 12 and 24 hours, we find that the forecast biases of the control run under the two leading times are almost the same. For simplicity, we present the forecast with a lead time of 24 hours. Figure 11 shows the differences in the wind and temperature fields between the truth run and control run at ground level at the AT and DT with a lead time of 24 hours. The truth run presents significant southerly winds with a mean speed of 2.32 m/s over the BTH region (see Fig. 11 (a)), while the control run forecasts a southerly wind with a mean speed of 0.74 m/s (see Fig. 11 (b)) and exhibits northerly wind biases, as shown in Fig. 11 (c). The weak southerly wind in the control run reduces the pollution transported from the south to the BTH region in the truth run, which results in a significant underestimation of the PM$_{2.5}$ concentration of the control run at the AT. In addition to this dynamic reason, the thermodynamical conditions are also key factors influencing the PM$_{2.5}$ forecasts. Both the truth run and the control run are able to simulate the temperature inversion layer, which prevents vertical dispersion of pollutants and promotes the accumulation of surface PM2.5. For the forecasts at the AT, the truth run has forecasted 0.11K/100m vertical temperature inversion layers at Dongsi station in Beijing City (the temperature arises 0.11K every 100m), whist the control run has forecasted

0.05K/100m. The mean lapse rate simulated by the truth run over the BTH region is 0.03K/100m and the control run has forecasted a 0.002K/100m. So the truth run simulated a more stable thermodynamic condition, which is favorable for the accumulation of surface air pollutants. Meanwhile, the negative temperature bias in the near surface of the control run decreases the production rate of precursors of $PM_{2.5}$ and the negative bias of relative humidity reduces the useful carrier of $PM_{2.5}$, causing a decrease in $PM_{2.5}$, finally favoring the underestimation of $PM_{2.5}$ at the AT in the control run.

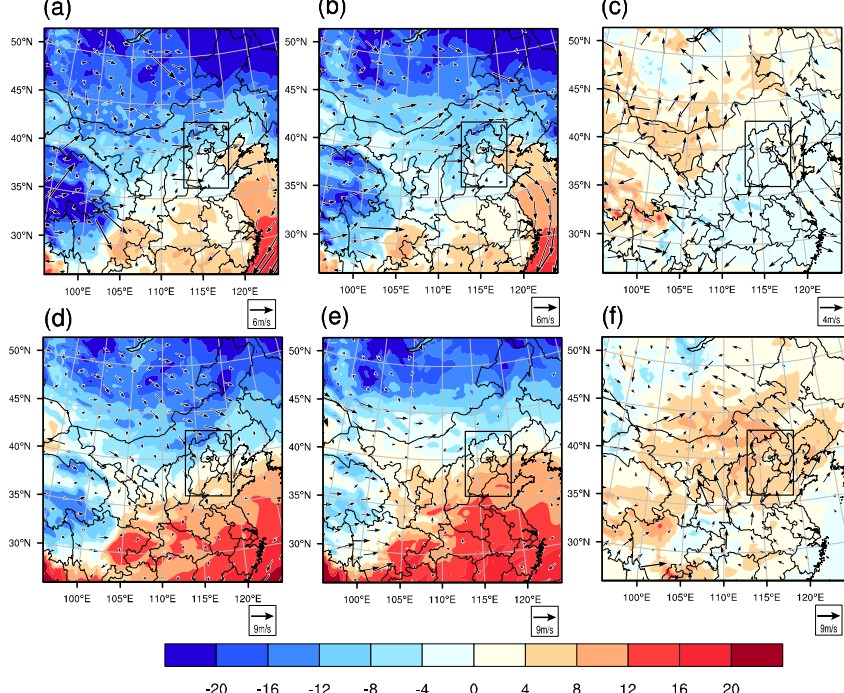

**Figure 11. The wind (vector, unit: m/s) and temperature field (shaded, unit: ℃) forecasts at the ground level at the AT with a lead time of 24 hours of the (a) truth run and (b) control run. The differences in wind and temperature fields between the truth run and control run (control run minus truth run) at the AT are shown in (c). (d-e) are the same as (a-c) but for the forests at the DT.**

From the above, it is clear that the control run exhibits northerly wind, a less table boundary layer, low temperature and relative humidity biases at the AT relative to the truth run. However, after assimilating the artificial meteorological variables over the sensitive areas determined by the CNOP-type errors to the initial analysis field of the control run, the $PM_{2.5}$ forecasts are promoted in forecasting skill. For the forecasts with lead times of 12 and 24 hours, the interpretations of why the assimilation runs increase the $PM_{2.5}$ forecast skill and its related mechanisms are similar. For simplicity, we present the interpretations in detail for the forecast with a lead time of 24 hours. In Fig. 12, we plot the spatial evolution of the 24-hour forecast differences of wind and $PM_{2.5}$ concentrations between the CNOP-EXP and control run. From Fig. 12 , we can see that the sensitive areas for the $PM_{2.5}$ forecast at the AT are

mainly located in the southern and northwestern parts of the BTH region (also see Fig. 7), and assimilating meteorological observations over the sensitive areas increases the southerly wind in the southern part of the BTH region at the initial field and finally enhances the southerly wind by 0.18 m/s

over the BTH region at the verification time, which is helpful for transporting southern pollution to the BTH region. Between the two areas, the sensitive area near the Inner Mongolia plays a more dominant role on the PM2.5 forecast of BTH region, by inducing a larger southerly wind component. In addition, the assimilation run has forecasted 0.06K/100m temperature inversion layers at Dongsi station and the mean lapse rate over the BTH region has reached to 0.004K/100m. The slightly improved thermodynamic

conditions further result in the modifications of the boundary layer structure featuring a decreased PBL height. The mean boundary layer height over the BTH region has decreased from 261m in the control run to 256m in the assimilation run, which also contributed to the increased ground level PM2.5 pollution and improved the PM2.5 forecast skill in the assimilation run. Moreover, assimilating the targeted observations increases the initial temperature and relative humidity in the western parts of the BTH

region and decreases them in the northwestern parts of the BTH region. Then, the western warm air moves easterly, and the northwestern cool air moves southeasterly, which finally decreases the temperature by 0.05 °C and the relative humidity by 0.6% at the AT over the BTH region. Decreased temperature and relative humidity are not beneficial for the formation of $PM_{2.5}$. From the above analysis, it can be found that the improvements in the $PM_{2.5}$ forecast skill in assimilation runs result from the

increased south wind and more stable boundary layer during the accumulation process.

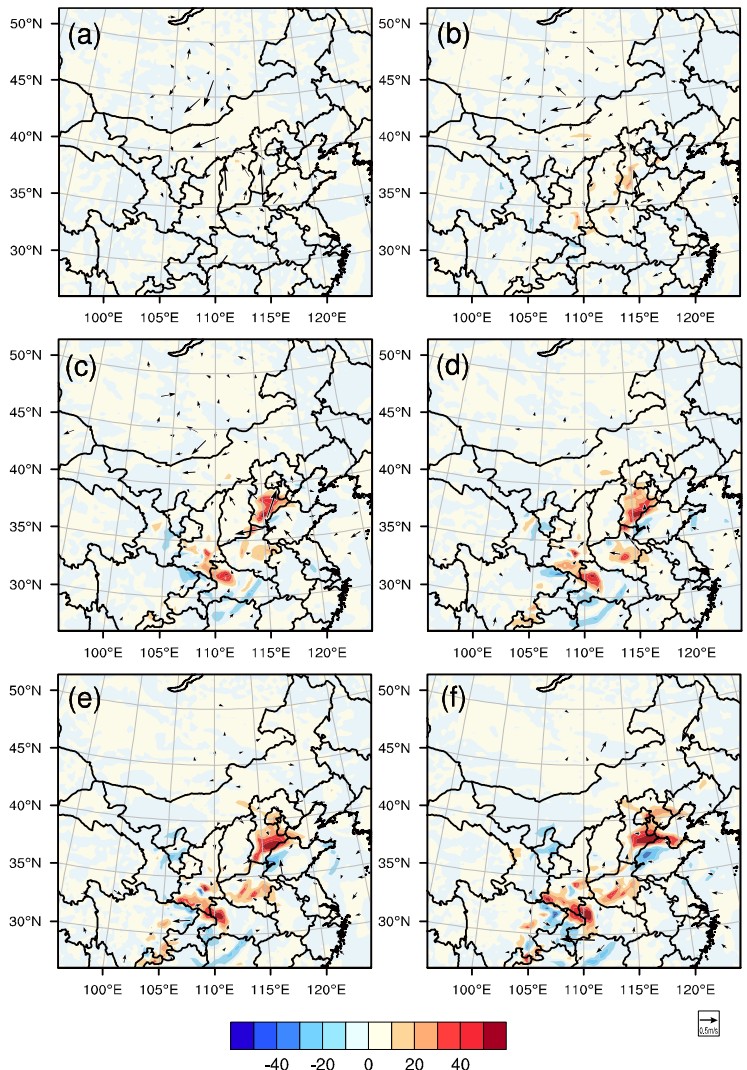

**Figure 12: The spatial evolution of the forecast differences of ground wind (vector, unit: m/s) and PM$_{2.5}$ concentrations (shaded, unit: μg/m$^3$) between the assimilation run (CNOP-EXP with the observing distance of 90km) and control run starting from 02:00 1 December with lead times of (a) 1-hour, (b) 6-hour, (c) 11-hour, (d)16-hour, (e) 21-hour, and (f) 24-hour.**

For the forecast at the DT, the truth run presents a large northerly wind with a mean speed of 5.24 m/s, as shown in Fig. 11(d), which blows the pollution from the BTH region to the south. However, the control run forecasts a southerly wind with a mean speed of 1.82 m/s (Fig. 11 (e)), which is the reverse of the truth run and might transport more pollution from the southwestern part to the BTH region than from the BTH region to the south in the truth run, finally contributing to the overestimation of the PM$_{2.5}$ concentration in the control run. Meanwhile, the control run also presents a warm temperature and much higher relative humidity biases, which prevent the dissipation of PM$_{2.5}$ over the BTH region and favors the overestimation of PM$_{2.5}$ at the DT (see Fig. 11(f)). When the targeted observations are assimilated to the control run at 24 hours before the DT and then the assimilation run is formulated, it increases the

685 northerly wind and decreases the temperature and relative humidity in the sensitive areas at the initial

time, which subsequently drives much cool and dry air in the sensitive area (i.e., the northwestern part

of the BTH region; also shown in Fig. 7) to the south and accumulates over the BTH region (see Fig. 13),

finally decreasing the temperature and relative humidity over the BTH region at the verification time,

improving the forecasts of the PM$_{2.5}$ concentrations in the assimilation run at the DT. It is obvious that

the improvement of both the dynamic and thermodynamic conditions is responsible for the increase in

the PM$_{2.5}$ forecast skill at the DT in the assimilation run.

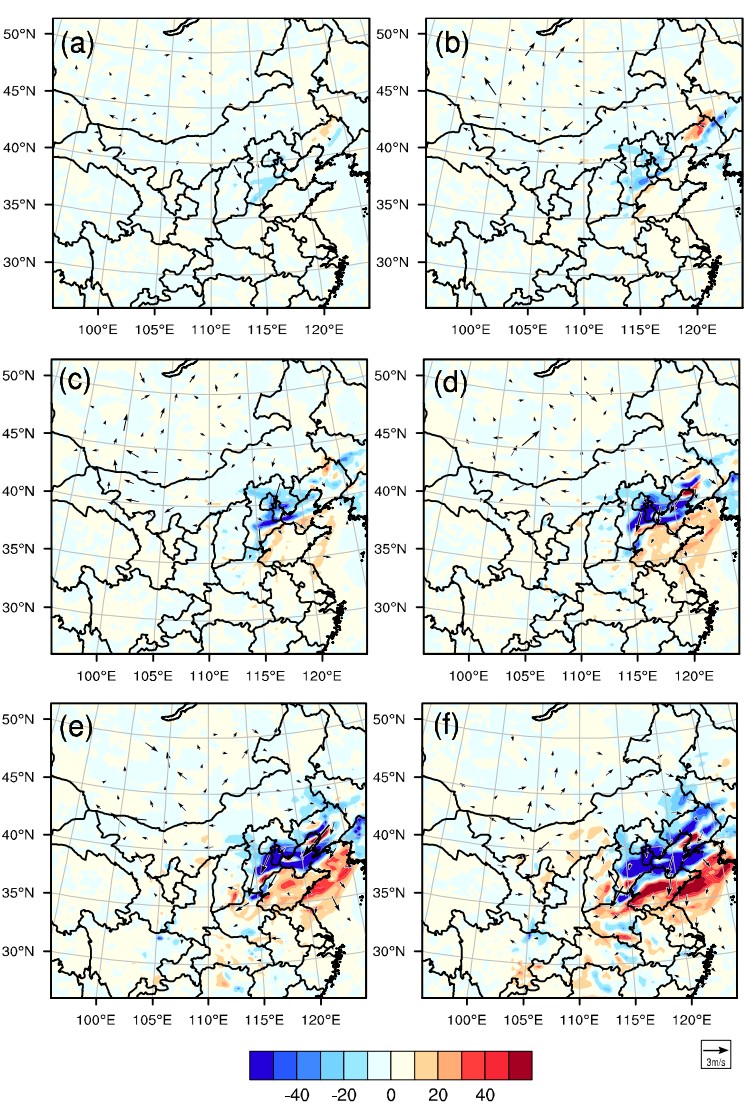

**Figure 13: The same as in Figure 12, but for the forecast starting from 14:00 on 2 December.**

## 6 Summary and discussion

Motivated by the important role of the meteorological initial field in air quality forecasts, we make the first attempt in applying the targeted observation strategy of the meteorological fields with the CNOP approach to the improvement of $PM_{2.5}$ forecasts using the WRF-NAQPMS model. By considering a heavy haze episode that occurred from 30 November to 4 December 2017 in the Beijing-Tianjin-Hebei region, we explore the effect of possible "targeted observations" on $PM_{2.5}$ forecasts during both the accumulation and dissipation periods of the haze event, where the targeted observations are represented by observation arrays consisting of 15 evenly and horizontally distributed grids through 4 pressure levels (i.e., 950, 850, 750, 500 hPa) in the sensitive areas identified by the CNOP-type errors and that include horizontal wind, temperature, and relative humidity components.

To improve the $PM_{2.5}$ forecast during the accumulation and dissipation periods of the haze event, forecasts with lead times of both 12 and 24 hours are investigated, where the AT (i.e., accumulation time, 02:00 BJT on 2 December) and DT (i.e., dissipation time, 14:00 BJT on 3 December) are selected as the verification times (i.e., the forecast times), respectively. We first calculate the CNOP-type errors for these 4 forecasts separately. Then, since the CNOP-type errors concentrate on different vertical levels and in different horizontal areas for different meteorological variables, including wind, temperature and moisture components, we propose using the vertical integral of CNOP-type errors to measure the comprehensive sensitivity of initial errors and to determine the sensitive areas for targeted observations of meteorological fields associated with the $PM_{2.5}$ forecasts. The results show that for the verification time AT, the sensitive areas identified by CNOP-type errors mainly concentrate in Dezhou city and central Inner Mongolia for a lead time of 24 hours and in Beijing and Tianjin cities for a lead time of 12 hours; for the verification time DT, the sensitive areas are determined as the region from Huhhot in Inner Mongolia to the Altai Mountains in Mongolia for a lead time of 24 hours and the region around Zhangjiakou and Chengde cities for a lead time of 12 hours.

Numerically, we conducted a series of OSSEs to explore whether the possible "targeted observations" in the above sensitive areas can improve the $PM_{2.5}$ forecasts of the BTH region and then to infer the usefulness of these sensitive areas in implementing practical field observations. For each of the 4 forecasts, we tried different observation arrays of 15 evenly and horizontally distributed grids through 4 pressure levels in the sensitive areas and assimilated them to the initial fields for evaluating the

improvement of PM$_{2.5}$ forecasting skill, finally suggesting a more useful observation array for improving

the forecasts at the AT and DT. Specifically, for the forecast at the AT, the observation array with a grid

space of 90 km in the sensitive area is more effective for a 24 hour lead time and a grid space of 150 km

performs the best for a 12 hour lead time; however, for the forecast at the DT, the observation array of a

grid space of 150 km leads to a better forecasting skill at a 24 hour lead time while that with a grid space

of 90 km results in a higher forecasting skill at a 12 hour lead time. To further confirm the usefulness of

CNOP in identifying the sensitive area for targeted observations, we compare the improvements of PM$_{2.5}$

forecasts after assimilating "targeted observations" in the sensitive areas and the additional observations

in the areas along the southwestern (Region-W) and northern (Region-N) directions of the BTH region

suggested by previous studies. The results show that the improvements of the PM$_{2.5}$ forecasting skills

with the additional observations deployed in Region-W and Region-N are significantly smaller than those

in the sensitive areas determined by the CNOP approach; in particular, assimilating the additional

observations over Region-W and Region-N cannot ensure a positive forecast benefit. All these results

indicate that preferentially implementing additional observations in the sensitive area determined by the

CNOP approach is more likely to significantly improve the PM$_{2.5}$ forecasts.

Physically, we interpret the reason why the possible targeted observations can significantly improve

the PM$_{2.5}$ forecasting skill by comparing the relevant meteorological fields before and after assimilation.

Since the interpretation and its related mechanisms are similar for the forecasts with lead times of 12 and

24 hours, we present only the interpretations in detail for the forecast with a lead time of 24 hours. During

the accumulation process, the control run forecasts a weaker southerly wind and a less stable boundary

layer at the AT, which is unfavorable for the accumulation of PM$_{2.5}$ and finally leads to a severe

underestimation of PM$_{2.5}$ at the AT. When the targeted observations are assimilated to the control run, the

southerly wind increases in the southern part of the BTH region at the initial state and finally enhances

the southerly wind over the BTH region at the verification time. The increased southerly wind transports

more PM$_{2.5}$ from the south to the BTH region and improves the PM$_{2.5}$ forecasting skills of the control

run at the AT. The assimilation also induces a more stable boundary layer in the assimilation run, which

contributed to the increased ground level PM$_{2.5}$ pollution and improved the PM$_{2.5}$ forecast skill. For the

forecast at the DT, the control run exhibits large southerly wind and positive temperature and relative

humidity biases, which prevents the dissipation of PM$_{2.5}$ and results in an overestimation of PM$_{2.5}$ at the

DT. When the targeted observations are assimilated to the control run, it increases the northerly wind and decreases the temperature and relative humidity in the sensitive areas at the initial state. The increased northerly wind drives the cool air in the sensitive area southward and finally blows more $PM_{2.5}$ from the BTH region to the south, which improves the $PM_{2.5}$ forecasting skills of the control run at the DT.

The present study provides numerical and physical evidence that the sensitive areas of meteorological initial fields associated with the PM2.5 forecasts indeed exists and deploying "targeted observations" of meteorological fields in the sensitive areas determined by the CNOP approach can significantly improve $PM_{2.5}$ forecasts. Such results formulate a theoretical basis to implement practical field campaigns associated with air quality forecasts. In the practical field campaigns, though the reanalysis data cannot be obtained in time, one can choose the forecast data from ECMWF, which are widely regarded as the best and most reliable forecast data currently, as initial field to yield a better forecast. Based on this forecast, one can compute the CNOP-type error to identify the sensitive area and design the relevant field observation networks. Such ideas have been applied on real-time typhoon forecasting and it has been verified to be able to improve greatly the typhoon forecasting skills (Duan and Qin., 2022; Qin et al., 2022). It is also noted that even if sufficient observations exist, the results in the present study can tell us which area of the observations should be preferentially assimilated to improve air quality forecasts.

As the first attempt to study the effect of targeted meteorological observations on improving air quality forecasts, we only utilized one event and, in the future, more events should be investigated to obtain a systematic and comprehensive conclusion about how to deploy targeted observations to improve $PM_{2.5}$ forecasts. Meanwhile, in the present studies, finite meteorological variables (wind, temperature, pressure, and water vapor) are selected to represent the sensitivity of meteorological initial fields on $PM_{2.5}$ forecasts. Though they are recognized as important meteorological variables on $PM_{2.5}$ forecasts over the BTH region (Chen et al., 2020), to get a comprehensive conclusion, the sensitivities of more meteorological parameters such as boundary layer height and atmospheric stability, which may not belong to an initial value problem but can be explored by the extension of CNOP method, such as CNOP-parametric perturbation (CNOP-P; Mu et al, 2010) or nonlinear forcing singular vector method (Duan and Zhou, 2013). Also, a WRF with the horizontal resolution of 30 km was preliminarily tried in the present study. Beyond doubt, this resolution is relatively low for the $PM_{2.5}$ forecasts. Nevertheless, the

sensitive areas revealed in the present study are still instructive for practical field observations of $PM_{2.5}$ forecasts because of the verifications through a series of OSSEs and reasonable physical interpretation shown in the context. In any case, a WRF with much higher resolution should be used in the future. In addition, only two verification times were adopted for determining sensitive areas and dependence of sensitive areas on forecasting times was not explored, which will be addressed in next paper.

In addition to meteorological inputs, emissions are also a key input for air quality forecasts. Accurate emission inputs are difficult enough in terms of their high uncertainties in time and 3-D space, and it is also challenging to satisfy the need for highly confident simulations of a specific event (Peng et al., 2017). Targeted observation may be a better strategy to improve the quality of emissions, and the determination of sensitive areas of emissions is certainly important. Previous studies have adopted the singular vector decomposition and adjoint sensitivity methods to identify the sensitive area for the emissions (Daescu et al., 2003; Goris and Elbern, 2013). However, it should be noted that the above two strategies are based on linear approximation of initial error evolutions and deploying the observations over the sensitive areas identified by these two strategies may not result in the largest improvement over the verification area, especially for the medium- and longer-range forecasts (Wang et al., 2011). Our current study represents the first step in studies of targeted observation strategies of meteorological variables associated with air quality forecasts with the application of CNOP, and only observations of meteorological fields are explored. Then, targeted observations of emissions based on the CNOP approach are expected to be studied for air quality forecasts in the future.

*Data availability.* Hourly surface $PM_{2.5}$ data are obtained from China National Environmental Monitoring Center (http://www.cnemc.cn/). The ERA5 reanalysis product is available at https://www.ecmwf.int/en/forecasts/datasets/reanalysis-datasets/era5. The NCEP GFS product is available at https://rda.ucar.edu/ datasets/ds084.1/. The data generated and/or analyzed during the study are stored on the computers at State Key Laboratory of Numerical Modeling for Atmospheric Sciences and Geophysical Fluid Dynamics (LASG; https://www.lasg.ac.cn) and will be available to researchers upon request.

*Author contributions.* Y.L., D.W. and W.Z. conceived the research. Y.L. and D.W designed the

experiments, performed the simulations, and analyzed the results. All authors contributed to the final drafting of the paper.

*Competing interests.* The authors declare that they have no conflict of interest.


*Acknowledgments.* The authors highly appreciate the two anonymous reviewers who provided constructive comments, that greatly improved the overall quality of the paper. The study was supported by the National Natural Science Foundation of China (Grant No. 42142039; 41930971).

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
