# Peer review of "Toward targeted observations of the meteorological initial state for improving the PM2.5 forecast of a heavy haze event that occurred in the Beijing-Tianjin-Hebei region"

_Atmospheric Chemistry and Physics, 2022_

## Author Comment (AC1)

**Response to Referee #1**

We would like to thank the referee for reviewing the manuscript and providing the valuable comments and suggestions. We are sorry that for some sentences we did not make them clear in the manuscript. We will update our manuscript following the suggestions. Below we answer the specific comments point by point. For readability the comments are shown in bold and italics.

**Review Comments:**

*The authors study the optimization of observation locations (targeted observation) to achieve an improved forecast for particulate matter. Interestingly they provide an example of a severe haze event in the Beijing area where early warnings by the authorities failed to be timely issued. This topic has attracted interest since more than a decade ago, in recent years also in the realm of atmospheric chemistry. It is strongly linked with research on predictability, observability and data assimilation. A wealth of methods has been devised, or derived from existing techniques found in the aforementioned realms.*

*In their study "Toward target observations of the meteorological initial state for improving the PM2.5 forecast of a heavy haze event that occurred in the Beijing-Tianjin-Hebei region" by Yang Lichao, Duan Wansuo, Wang Zifa, and Yang Wenyi addressed the optimisation of measurement deployment for full and atmospheric chemistry application by devising a meteorological problem of optimal measurement dislocation. In my review I question this strategy with some detail, encouraging the authors to refute my demurs.*

*Methodology:*

*The motivation of the work where's to improved aerosol forecast which failed significantly in the case study selected not fault booked. So, there might be different reasons for this failure notably a faulty emission inventory or degraded weather forecasts. In their approach the authors seeked the reason only in the weather forecast. Hence, they tried to improve the prediction by better located meteorological observations which they assimilated to obtain better initial values for the forecast. The other option let the forecast deficiencies might result from faulty emission inventories was not considered is out giving any evidence of reason. The authors quite deliberately declared a better forecast resulting from era 5 reanalysis to be the truth while another one from GSE was declared control which verse aspired to be improved by additional and optimally located observations. The resulting simulation product provided the improved forecast in relation to the control room but not as good as the truth run identity fight before. The statistical analyses of the assimilation run were then provided as quantitative proof of concept.*

**My critique addresses several items.**

***1.1 Firstly, how would the method provide reasonable results if not the meteorological forecasts are deficient but the emission inventory, which are in fact often poorly known. Figure 1 of the manuscript does not give any indication that the major discrepancy is only due to meteorological prediction flaws.***

**Response:** We agree that the uncertainties occurring in both emissions and meteorological fields cause the forecast uncertainties of PM 2.5. In fact, the errors of model itself are also influencing the forecasting uncertainty of $PM_{2.5}$. Furthermore, the forecasting errors caused by these uncertainties are interactive and it is difficult to quantify exactly their respective contribution. In Figure 1, it, as the reviewer said, does not give any indication that the major discrepancy is only due to meteorological prediction flaws. In fact, we did not state that the $PM_{2.5}$ forecasting uncertainties are solely from meteorological forecasting prediction flaws. Especially, when comparing the simulations with observations in Figure 1, we have to say that the $PM_{2.5}$ forecasting uncertainties are from the combined effect of meteorological prediction flaw, model uncertainties, and emission inventory uncertainties. In the present study, we consulted a few previous studies which have demonstrated the important role of meteorology field on $PM_{2.5}$ forecasts in the BTH region (Bei et al., 2017; Gilliam et al., 2015); meanwhile, our results also showed that the $PM_{2.5}$ forecasting uncertainties are sensitive to the initial uncertainties of meteorological field (see the two simulations in Figure 1, which were obtained by integrating the WRF-NAQPMS with different initial meteorological fields but with the same emission inventory and model) despite they are not necessarily the most important error source of PM2.6 forecasting uncertainties; and we decided to focus on the meteorology uncertainties in the present study, but leaving uncertainties of model errors and emission inventory to be explored in the future.

To identify the initial error effect of meteorology, we adopt the idea of Lorenz (1965) on two types of predictability problems. The first focuses on the effect of initial error growth with an assumption of perfect model while the second is to assume a perfect initial field for exploring the effect of model error growth. The present study follows the idea of the first type of predictability problem; and to separate the initial meteorological error effect, an assumption of perfect model is done. For the WRF-NAQPMS model, we also have to additionally assume the emission inventory is perfect and keep the emission inventory in all the simulations the same. Similar doings are also used in the previous studies for air quality simulations or forecasts, e.g. Gilliam et al. (2015) and Bei et al. (2017), etc. However, whichever it is initial field or model, even emission inventory, it certainly consists of uncertainties. So, in the present study, we have to take the better simulation initialized by the ERA5 (which was obtained by assimilating all available observations with more advanced model by ECWMF and can therefore be a good approximation to the truth except for observations) as "truth run" because we cannot obtain observations from the Monitor center; and the worse simulation initialized by GFS forecast data as "control forecast" to separate the initial error effect. It is believed that the reduction of the bias between the "truth run" and the

"assimilation run" due to the assimilation of targeted observations can indicate the decrease of the bias between the assimilation run and real observations of $PM_{2.5}$ concentration.

***1.2 In addition, if both forecasts, that is the truth and the control run, suffer from the same problem, as for example poor boundary layer height simulations, then the method proposed incapable to give any evidence of any source of error.***

**Response:** We thank the valuable comments. When both forecasts, that is the truth and the control run, suffer from the same problem, as for example poor boundary layer height simulations, then associated forecast uncertainties are included in the difference between simulations and observations. The differences of the two simulations in Figure 1 can only indicate the sensitivity of $PM_{2.5}$ forecast to the accuracy of initial meteorological field. If one will identify the role of boundary layer uncertainties in yielding $PM_{2.5}$ forecasting uncertainties along similar thought presented in this study, the $PM_{2.5}$ simulations should be made with the same initial meteorological field and the same emission inventory but different boundary layer schemes to reveal the role of boundary layer uncertainties. Based on these simulations, an extension to the CNOP method, CNOP-parametric perturbation (CNOP-P; Mu et al, 2010) or nonlinear forcing singular vector (Duan and Zhou, 2013), can be used to identify the sensitivity of boundary layer. And we thank the referee for providing us a great research idea for our future studies.

***2.1 Secondly, I put the assimilation procedure in question. So let us assume the authors are right in their suspicion, that the meteorological forecast is the source of misprediction of the aerosol concentrations. A sound synoptic description of the weather situation and its evolution is lacking as are appropriate surface weather charts.***

**Response:** As argued above, we did not emphasize that the metrological forecast uncertainties is the unique source of the misprediction of the $PM_{2.5}$ event. In the present study, we find that the $PM_{2.5}$ forecasting is sensitive to the initial uncertainties of meteorological field (see the two simulations in Figure 1) and investigate the role of the targeted observation for meteorological field in improving $PM_{2.5}$ forecasting skill, but leaving the uncertainties of model itself and emission inventory to be explored in the future. In addition, it is very difficult for us to obtain the meteorological observations from the Monitoring centers; then we adopt the more efficient ERA5 data that one often uses as initial value for the model to study meteorological predictability. Therefore, it is hard to plot the weather charts corresponding to the difference between simulation and observation. Nevertheless, when we evaluated the role of targeted observations of meteorological field in the manuscript, we plotted the weather charts before and after assimilating targeted observations and showed the weather conditions for improving $PM_{2.5}$ forecasting skill (see Figures 10-11).

*2.2 In addition, a discussion on the boundary heights and stability would be in place, as these are a critical parameters, controlling the capture of emissions.*

**Response:** We thank your valuable suggestions. We will add more discussions on the boundary heights and stability in the revised manuscript. Specifically, during the accumulation process, both the truth run and the control run are able to simulate the temperature inversion layer, which prevents vertical dispersion of pollutants and promotes the accumulation of surface $PM_{2.5}$. For the forecasts at the AT, the truth run has forecasted 0.11K/100m vertical temperature inversion layers at Dongsi station in Beijing City (the temperature arises 0.11K every 100m), whist the control run has forecasted 0.05K/100m. The mean lapse rate simulated by the truth run over the BTH region is 0.03K/100m and the control run has forecasted a 0.002K/100m. So the truth run has a more stable thermodynamic condition. After the assimilating the targeted meteorological variables to the control run, the assimilation run has forecasted 0.06K/100m temperature inversion layers at Dongsi station and the mean lapse rate over the BTH region has reached to 0.004K/100m. The slightly improved thermodynamic conditions further result in the modifications of the boundary layer structure featuring a decreased PBL height. The mean boundary layer height over the BTH region has decreased from 261m in the control run to 256m in the assimilation run, which also contributed to the increased ground level $PM_{2.5}$ pollution and improved the $PM_{2.5}$ forecast skill in the assimilation run.

*2.3 What happens, if both truths run and control run err with the stability in the same way, but differ in , say, as in this paper, in the horizontal wind direction? In this case, the CNOP type error is critically incomplete.*

**Response:** We think the comment may consist of two questions. The first is "What happens if both the truth run and control run suffer from error with stability in the same way?". When both the truth run and control run suffer from the same problem, then associated forecast uncertainties are included in the difference between simulations and observations. As we explained in Comment 1.1, the differences between the simulations and observations may be attributed to the uncertainties of meteorology, emission inventory and model itself. In the present study we only focus on the effect of meteorological initial errors on the $PM_{2.5}$ forecasts. To separate the initial effect, we adopt the same model and same emission inventory but different meteorological initial fields to verify the sensitivity of meteorological initial conditions on $PM_{2.5}$ forecasts. And the CNOP, which can represent the most sensitive initial errors, is calculated based on the better simulation, as we explained in Comment 3. As for the error related to the stability in the truth, it may be attributed to the initial errors existing in the reanalysis (ERA5) or the model itself. In that case, we admit that the CNOP-type error based on the "truth run" may be incomplete, since the reanalysis is not the real truth and the model is not a perfect model. However, as we explained in Comment 3, since we are not able to obtain the meteorological observations from the Monitoring centers, we have to adopt the more efficient ERA5 data as initial values and more advanced WRF model to study the meteorological predictability.

The second question is "the components of CNOP type error may be incomplete if only the horizonal wind is included but excluding the stability". Actually, the stability is not a direct meteorological variable, but a variable related to temperature. Meteorological conditions, such as the wind, temperature, relative humidity, stability, boundary layer height, precipitation all have significant impacts on the regional $PM_{2.5}$ forecasts in CTMs (Godowitch et al., 2011). Chen et al., (2020) reviewed hundreds of papers on the meteorological factors on $PM_{2.5}$ concentrations for the BTH region, and they concluded that the wind and humidity are the dominant meteorological factors (Table 1 in Chen et al., (2020)). In our study, to include as many meteorological variables as possible as the components of CNOP type error, we use the total energy norm, which includes the wind, temperature, water vapor mixing ratio and pressure perturbations, to constrain the CNOP. According to the results obtained by Chen et al., (2020), we think the variables considered in the CNOP are adequate.

*2.4 The method proposed by the authors is designed to deploy 15 different observation locations which might be the key to the sufficiently well performing forecast. So, all in all they select 15 times four height levels times 4 meteorological parameters that means individual 240 observations and tested the performance of these idealized network with respect to varied distances. In fact this is a variable the radiosonde network or air borne drop sonde area placed windward of the area of interest to be predicted. Leaving aside the practicability, I put into question the benefit for improved forecast with 3D-var by localized observations, given the synoptic balance conditions to be fulfilled. The authors result indicates this: Looking at Fig. 9, panels a) and b), it appears to be likely that the eastern side of a high pressure system (northerly winds) at the eastern side of the panels is shifted further eastbound in the truth run (a), than in the control run. It is not possible to correct this error by assimilation of data from a localized observation network alone.*

**Response:** Yes, the reviewer got a right conclusion. It is not possible to correct this error by assimilation of data from a localized observation network alone. In fact, the "target observation" is to deploy few additional observations in some localized areas (sensitive areas) where the additional observations are expected to just have a large contribution to reducing the prediction error in the given **verification area** (Snyder., 1996), rather than other areas. In Figure 9, we agree that the eastern side of a high pressure system at the eastern side is shifted further eastbound in the truth run than in the control run; and assimilating the targeted observation does not correct the error. Actually, the sensitive area for targeted observation in the present study is determined on the verification area, i.e. the BTH region (i.e. the black rectangle in the figure 9). That is to say, assimilating the targeted observation is to preferentially improve the meteorological field in the verification area- BTH region, rather than other regions. Therefore, it is reasonable, as the reviewer pointed, the forecast errors in the high pressure system cannot be corrected by targeted observation. This may also indicate that the high pressure system at the eastern side does not play the dominant role in influencing the PM 2.5 in BTH in this event.

3. *How should the set-up with two model runs operate practically? How do we find the "real truth"? In fact, the only thing what can be done is to achieve an optimal meteorological forecast in general, with all available observations. After this, optimal sensitivity areas can then be identified for chemical concentration measurements, not for meteorological observations, because the truth is not known. Recommendation: To account for these problems, the authors are encouraged to change their validation strategy and conduct numerical experiments, where the emission inventory is taken as true and a nature run produces artificial ("synthetic") aerosol concentration observations, which then are to be reproduced by the proposed targeted observation procedure, analog to Observation System Simulation Experiments (OSSE) made in data assimilation developments.*

**Response:** We thank your valuable comments. Yes, to apply the targeted observation procedure on the emission inventory is a good idea. As we discussed in Lines 704-710, "targeted observation may be a better strategy to improve the quality of emissions, and the determination of sensitive areas of emissions is certainly important. Implementing additional and/or optimizing environmental monitoring stations according to the sensitivity of targeted observations and obtaining more useful observations will lead to significant improvement of air quality forecasting skills". That is to say, we have realized that this is also an important work. Actually, both the emission and meteorology may substantially influence the $PM_{2.5}$ forecast. The study of targeted observations on both meteorology and emission is meaningful, but that would be accomplished step by step. In the current study, we focus on the targeted observations on meteorology forecasts first. Our current study represents a first step that the CNOP algorithm of targeted observations is applied to the studies of air quality forecasts. Then, target observations of emissions identified by CNOP are expected to be studied for air quality forecasts in the near future.

To investigate targeted observations, the Observing System Experiments (OSE) and Observing System Simulation Experiments (OSSEs) are typically designed to use the data assimilation ideas to investigate the potential impact of prospective observing systems (observation types and deployments) (Lahoz et al., 2010). The OSE consist of a control run in which all the observational data are assimilated; and an assimilation run from which the observation type under evaluation is excluded while all the other observations are kept as the same as in the control. A comparison of forecast skill between the control run and the assimilation run against the observations will be evaluated (Lahoz et al., 2010) and the role of observation under evaluation is revealed. However, under many circumstances, we are not able to obtain the observations due to varying reasons. Then the OSSE is developed. The structure of an OSSE is formally similar to that of an OSE with one important difference: OSSEs are assessment tools for new data, i.e., the data obtained by hypothetical observing systems that do not yet exist. Then an OSSE consists of a reference atmospheric state, which is usually done

with a good quality model with a good initial value. It is often called the "truth run", from which artificial "observations" are constructed and against which subsequent OSSE experiments are verified. Related to the targeted observation here, the OSSE can include a control run that is obtained by assimilating artificial observations constructed based on the "truth run" and an assimilation run that is generated by assimilating additional artificial observations. A comparison of forecast skill between the control and assimilation runs against the truth run will be evaluated to reveal the role of targeted observations (Lahoz et al., 2010). In the present study, we are not able to obtain the observation data from the Monitor Center to evaluate the simulation by OSE. So we use the OSSE instead, in which, although the "truth run" is not yielded by assimilating the observations but by directly assigning more efficient reanalysis data ERA5 to generate the initial field (which was obtained by assimilating all available observations with most advanced model by the ECMWF and can therefore be a good approximation to the truth). Particularly, we take the better simulation initialized by ERA5 as the "truth run" and the worse simulation initialized by GFS forecast data as the control run; while the run after assimilating the targeted observations is taken as the assimilation run. When we use the ERA5 reanalysis data, we are able to simulate the heavy hazy event much better, but it is failed in Monitor center. So it is acceptable to take the ERA5 simulation as an approximation to the real state of atmosphere and it will also be helpful for separating the initial effects. Then the comparison of forecast skills between the control and assimilation runs against the truth run will judge the usefulness of targeted observations associated with initial meteorological fields. It is inferred that if the forecast bias between the "truth run" and the "assimilation run" due to the assimilation of targeted observations are largely reduced, the bias between the assimilation run and real observations of $PM_{2.5}$ concentration will also decrease.

The CNOP method, similar to the approach of data assimilation, is generally operated with an assumption of perfect model. So the sensitive area for targeted observation should be identified based on a scenario of perfect model and the CNOP should be calculated by a perfect model, then such a sensitive area is the true sensitive area. However, whether it is model or initial field, there are uncertainties. Therefore, in the studies of targeted observations, one has to adopt the model of good quality and obtain a good simulation and then compute the CNOP superimposed on the simulation. In the present study, we, to achieve a good simulation, adopted the ERA5 reanalysis as initial field and integrated the advanced WRF model because we cannot obtain real observations from the Monitor Center. However, in field campaigns, one even cannot obtain reanalysis in time. In this situation, one can choose the forecast data from ECMWF, which are widely regarded as the best and most reliable forecast data currently, as initial field to yield a better forecast. Based on this forecast, one can compute the CNOP-type error to identify the sensitive area. Such idea has been applied on typhoon forecasting by the authors' group and when the useful real-time typhoon observations are obtained, it has been verified to be able to improve greatly the typhoon forecasting skill (Duan and Qin., 2022; Qin et al., 2022). In the present study, we adopt the ERA5 reanalysis, which is of less uncertainty than the forecast

data from ECMWF and is helpful for achieving a much reliable sensitive area for meteorology associated with PM$_{2.5}$ forecast.

As for the studies of targeted observations on aerosol concentrations, the strategy suggested by the reviewer is much realistic, because the emission inventories can be taken as real observations, rather than the simulated observations generated from the model like what we did in the present study due to the unavailable observations. We thank the referee's suggestion and will adopt the great research idea in the study of emission uncertainties.

**Literature:**

*4. The authors claim that they are the first to transfer the method of targeted observation to atmospheric chemistry, which does not at all apply! Regrettably, it appears that the authors are not aware of the number of meanwhile growing set of papers on this very matter. Some relevant papers are given here for convenience. Studies focusing on atmospheric chemistry observation targeting, explicitly or implicitly, are indicated by boldface letters, and merit special attention. As the authors focus on meteorological targeted observations I include several other studies on that issue, which might also be considered.*

*Recommendadtion: We strongly recommend who review this literature given below.*

Bellsky T, Kostelich EJ, Mahalov A (2014) Kalman filter data assimilation: targeting observations and parameter estimation. Chaos 24(2):024406. https://doi.org/10.1063/1.4871916

Berliner, L. M., Lu, Z., and Snyder, C.: Statistical design for Adaptive Weather Observations, J. Atmos. Sci., 56, 2536–2552, 1998.

Bishop, C. H. and Toth, Z.: Ensemble Transformation and Adaptive Observations, J. Atmos. Sci., 56, 1748–1765, 1998.

Buizza, R., Cardinali, C., Kelly, G., and Thepaut, J. N.: The value of targeted observations, ECMWF Newsletter, 111, 11–20, 2007.

Daescu, D. N. and Carmichael, G. R.: An Adjoint Sensitivity Method for the Adaptive Location of the Observations in Air Quality Modeling, J. Atmos. Sci., 60, 434–450, 2003.

Goris N, Elbern H (2013) Singular vector decomposition for sensitivity analyses of tropospheric chemical scenarios. Atmos Chem Phys 13:5063–5087. https://doi.org/10.5194/acp-13-5063-2013

Goris N, Elbern H (2015) Singular vector based targeted observations of chemical constituents: description and first application of EURAD-IM-SVA. Geosci Model Dev 8:3929–3945. https://doi.org/10.5194/gmd-8-3929-2015

Khattatov, B. V., Gille, J., Lyjak, L., Brasseur, G., Dvortsov, V., Roche, A., and Waters, J.: Assimilation of photochemically active species and a case analysis of UARS data, J. Geophys. Res., 104, 18715–18738, 1999.

Liao, W., Sandu, A., Carmichael, G. R., and Chai, T.: Singular Vector Analysis for Atmospheric Chemical Transport Models, Mon. Weather Rev., 134, 2443–2465, 2006.

Szunyogh I, Toth Z, Emanuel KA, Bishop CH, Woolen J, Marchok T, Morss R, Snyder C (1999)

Ensemble based targeting experiments during FASTEX: the impact of dropsonde data from the Lear jet. Q J R Meteorol Soc 125:3189–3218. https://doi.org/10. 1002/qj.49712556105

Wu X, Jacob B, Elbern H (2016) Optimal control and observation locations for time-varying systems on a finite-time horizon. SIAM J Control Optim 54(1):291–316. https://doi.org/10.1137/15M1014759.

Wu, Xueran; Elbern, Hendrik, Jacob, Birgit; The assessment of potential observability for joint chemical states and emissions in atmospheric modelings, Stochastic Environmental Research and Risk Assessment https://doi.org/10.1007/s00477-021-02113-, 2022.

**Response:** We are sorry that we did not conduct a fully literature review. We thank the referee for listing the related papers, especially the publications of targeted observation in atmospheric chemistry. We have read all the recommended literature carefully and will cite them in the revised manuscript. The sentences such as "the first application on the atmospheric chemistry" will be modified in the revised manuscript.

5. ***The paper is in fact about an algorithm for targeted observations. As such no results for atmospheric chemistry per se are offered and can be expected. So it is suggested to submit the manuscript to GMDD rather than ACPD.***

**Response:** We thank the referee's comment. However, we do not think our paper is just an algorithm or a technical paper. In fact, it is a study on the application of the CNOP algorithm to identify the sensitive area for targeted observations of meteorological initial fields associated with the $PM_{2.5}$ forecasts. The relevant physical process and explanations on how the targeted observations of meteorological initial conditions in the sensitive area leads to the improvement of the $PM_{2.5}$ forecasts is also investigated in the paper. According to the scope of ACP, our study contributes to understand how the meteorological initial states influence the transportation and accumulation of $PM_{2.5}$ concentrations by atmospheric dynamic and/or heating, etc., which belongs to the study of atmospheric physics processes related to the $PM_{2.5}$ variations.

Our study also provided a potential application prospect in identifying the sensitive area for emission inventories. Although other methods such as singular vector, adjoint sensitivity, and ETKF provided by literatures listed by the referee can also be used, they are approaches of linear approximation. The CNOP considers fully effect of nonlinearity and overcomes the linear limitation of the traditional approaches and presents the most sensitive initial perturbation (Mu et al., 2003), then being able to effectively identify the sensitive area for targeted observations. This argument has been verified by a lot of studies (Mu et al., 2009; Chen et al., 2013). Therefore, if this article is published in ACP, it can be expected that CNOP algorithm and its potential applications on emission inventories will be known by more researchers in the field of atmospheric chemistry. It is also expected that the CNOP can be a useful approach to addressing problems of air quality forecasts. So it is very anticipated that this article can be published in ACP after addressing all concerns of reviewers.

**Specific remarks:**

1. *The authors should use the term targeted observations throughout, as in the paper by Majumdar. (Not target observations. Majundar made only deviations by grammatical reasons.)*

**Response:** We will modify "target observations" to "targeted observations" throughout the paper.

2. *Discussion of emission inventory uncertainty and other uncertainty sources. There is a well-established corpus of literature addressing uncertainty sources of chemistry transport model, where meteorological uncertainties are only one among others. The authors' decision to solely focus on meteorology needs a sound quantification.*

**Response:** As we explained in Comment 1.1, we agree that the uncertainties occurring in emissions, meteorological fields, model itself and other sources cause the forecast uncertainties of PM 2.5. We noticed that a lot of papers emphasized the important role of meteorological field in transporting $PM_{2.5}$ and yielding $PM_{2.5}$ forecasting uncertainties in the BTH region (Bei et al., 2017; Gilliam et al., 2015; Chen et al., 2020). Furthermore, we also find that the $PM_{2.5}$ forecasting in this heavy pollution event concerned in the present study are also sensitive to the initial uncertainties of meteorological field (see the two simulations in Figure 1), despite meteorological uncertainties could not be the most important contributor to the $PM_{2.5}$ forecasting uncertainties. Therefore, we first focus on the meteorological uncertainties in the present study. This does not mean that the uncertainties of model itself and emission inventory are not important, but we think that we should address these uncertainties step by step. In the present study, we first pay attention to meteorological uncertainties and leave uncertainties of model and emission inventory to be explored in the future. It is expected that the combined effect of uncertainties of model, meteorological, and emission inventory can be finally addressed.

3. *What is the assumed dominant composition of PM 2.5 matter (mineral dust, secondary anthropogenic, …) , and is the emission inventory sufficiently resolved by 30 km grid size?*

**Response:** The components of $PM_{2.5}$ simulation here include black carbon (BC), organic carbon (OC), secondary inorganic aerosol (sulfate, nitrate, ammonium) and primary $PM_{2.5}$ emitted directly from various sources. The dominant composition of $PM_{2.5}$ varies with regions and periods. During this event, the dominant compositions are nitrate and organic carbon.

As we discussed in Line 695-696, we have realized that the resolution 30km is relatively low for $PM_{2.5}$ forecasts. Nevertheless, even thus, the simulation initialized by ERA5 can well represent variability of the accumulation and dissipation processes

of PM$_{2.5}$ despite the uncertainties against the observations (Figure 1). It indicates that the emission inventory adopted here can be resolved. Here we also present the spatial distribution of daily average PM$_{2.5}$ concentrations of observation and ERA5 simulation on Dec, 1$^{st}$ (Figure R1). It shows that the ERA5 simulation is able to produce the spatial distribution of the observed PM$_{2.5}$. This also indicates the emission inventory at 30km is acceptable for this heavy pollution event.

We agree that the emission inventory will be better resolved in a higher resolution. So we have the related discussion in the manuscript. As seen on Line 695-696, "a WRF-NAQPMS model with much higher resolution will be used in next study on PM$_{2.5}$ forecasting."

[Figure]

Figure R1 The spatial distribution of the daily average PM$_{2.5}$ concentrations of observation (circle) and ERA5 simulation (shaded) on Dec 1st. (unit: μg/m$^3$)

4. *Why is the targeted observation approach not applied to emission sources? It is well understood that emissions are rarely measurable (eddy covariance towers are a practically unavailable exemption). Yet concentration observations in the vicinity of sources could be exploited instead with some benefit.*

**Response:** We thank your valuable suggestions. Yes, it is important to apply the targeted observation approach to emission sources, as we discussed in Lines 701-710 in the manuscript. Actually, both the emission and meteorology may substantially influence the PM$_{2.5}$ forecast. The study of targeted observations on both meteorology and emission is meaningful and would be accomplished step by step. As we explained in Comment 1.1, a lot of previous studies have emphasized the important role of meteorological field on PM$_{2.5}$ forecasts in the BTH region (Liu et al., 2017; Zhang et al., 2018). Also we find that the PM$_{2.5}$ forecasts concerned in the present study are sensitive to the meteorological initial conditions (see the two simulations in Figure 1), which indicates the important role of meteorology forecast accuracy in improving

PM$_{2.5}$ forecast. Even though the meteorology may not be the first factor that influences the PM$_{2.5}$ forecasts, the large differences between the two simulations also motive us to apply the target observation strategy to improve the accuracy of the meteorological forecasts, then the PM$_{2.5}$ forecasts. So in the current study, due to important role of meteorology, and also as the first attempt to apply CNOP sensitivity to PM$_{2.5}$ forecasts, we investigate the targeted observations on meteorology forecasts associated with PM$_{2.5}$ forecasts. Then, as the referee suggested, to apply the targeted observation on emission sources, such as locating the eddy covariance tower to get the concentration observations, is a great research idea and motivate us to carry on our studies on the emission uncertainties in the near future.

**Minor issues**

1. ***Title: The typical term is. Targeted observations. It is recommended, to adapt accordingly.***

**Response:** Thank you very much for your suggestion. As expected, we will modify the "target observation" to "targeted observation" in the revised manuscript.

2. ***Feedback emissions-meteo around L 545 mentioned, but emission inventory uncertainties poorly addressed.***

**Response:** We thank the referee's comments. In the present paper, as we argued above, we only focus on the sensitivity of meteorological initial conditions on PM$_{2.5}$ forecasts, leaving the studies of emission uncertainties to be explored in the future. In the OSSEs we assume that the emission inventory is accurate and keep the emission inventory in all the simulations the same [as did in Gilliam et al. (2015), Bei et al. (2017), etc.]. So the uncertainties among the PM$_{2.5}$ simulations in the present study are only from the differences of meteorology forecasting, and in the Interpretation section, we only focus on explaining how improving the meteorological initial condition influence the PM$_{2.5}$ simulations. We will follow the referee's suggestions and add more discussions on the emission uncertainties in the "Summary and Discussion" section of the revised manuscript.

3. ***Fig. 9 Substantial differences between truth and ctrl run. How is this possible? Could be phase error. This renders the assimilation of artificial data critical as this local information is inconsistent with the synoptic situation (imbalance).***

**Response:** The differences in Figure 9 are dependent on the meteorological initial conditions, since both the truth run and control run use the same model and emission inventories. The initial meteorological condition for the truth run is generated by the ERA5 reanalysis data, which is the newest generation ECMWF reanalysis data which combines vast amounts of historical observations into global estimates using

advanced modelling and data assimilation systems. The initial meteorological condition for the control run is generated by the NCEP GFS, which is the forecast data generated by a global forecast system in NECP. The forecast data consist of larger uncertainties and very different from those of ERA5. Figure R2 shows the initial condition of WRF simulations generated by the ERA5 and NCEP GFS at the AT and DT with lead times of 24 hours. A substantial difference between the two initial conditions exists, so it is reasonable that difference of meteorological forecasts at the AT and DT between the control and truth run is large.

As for the imbalance the referee has pointed, in our opinions, does not exist in our study. Though only the observations in the sensitive area are assimilated, the initial condition outside of the sensitive area will be coordinated through the data assimilation technique. Both the initial states before and after the assimilation are the solutions to the model, they are definitely be balanced. So the assimilation of artificial data will not be imbalanced with the synoptic situation.

[Figure]

Figure R2 The initial condition of wind (vector, unit: m/s) and temperature field (shaded, unit ) for the forecast at the AT of the (a) truth run (b) control run. (c-d) are the same as (a-b) but for the

forecasts at the DT. As we can see, a substantial difference between the two initial conditions exists, so it is reasonable that difference of meteorological forecasts at the AT and DT between the control and truth run is large.

**4.** ***As meteo forecast deficits are assumed for PM prediction flaws: Validation against meteo data lacking. Why?***

**Response:** As we explained in Comment 3 (critical comments), it is difficult for us to obtain the meteorological observation from the Monitoring center. Those inspire us to design the OSSEs to study the meteorological targeted observations associated with PM 2.5 forecasts. In the structure of OSSE, the "truth run" is a reference atmospheric state that is generated by a model of good quality and a comparison of forecast skill between the control and assimilation runs against the truth run will be evaluated (Lahoz et al., 2010). In our study, the ERA5 simulation is taken as the truth run, and the subsequent experiments should be evaluated against the ERA5 simulation according to the OSSE.

**5.** ***L 39-50: Do the authors claim that this is valid for their study region, or globally? Most studies point at emission strengths uncertainties. More precisely, the uncertainties of predictions must be pondered with forecast time. On short range forecasts today's meteo forecast uncertainties are small, if not extraneous, when compared with both anthropogenic and biogenic emissions. Please discuss this with more scrutiny.***

**Response:** We thank the referee's comment. The meteorological conditions have a great impact on $PM_{2.5}$ forecasts for our study region. We will emphasize it is valid for our study region in this revised manuscript.

We did not deny the importance of emission uncertainties on $PM_{2.5}$ forecasts. As we argued above, we agree that the meteorology, emission inventories and the model itself all contribute to the $PM_{2.5}$ forecast uncertainties. Due to the former studies (Bei et al., 2017; Gilliam et al., 2015) and our results (see the two simulations in Figure 1), we first investigate the role of the meteorological targeted observation in improving $PM_{2.5}$ forecasts in the present paper although the meteorology may not be the most important for the $PM_{2.5}$ forecasts, and leave the studies on the emission uncertainties in the near future. In any case, the effect of meteorological, model itself, and emission inventories uncertainties will be studies step by step. As the first attempt to apply the CNOP sensitivity on $PM_{2.5}$ forecasts, the successful application of CNOP in meteorological targeted observations will also inspire us to apply the CNOP method in the study of emission uncertainties in the future.

We agree that the uncertainties of predictions are pondered with forecast time. For the event we studied, we showed the spatial distributions of the $PM_{2.5}$ forecast errors in the control run at the AT and the DT with the lead times of 24 hours in Figure 7. If taking the absolute value of the biases, then the mean biases of the whole

BTH region are 34.22 and 64.13 ug/m3 at the AT and DT, respectively. In some areas of BTH, the errors are more than 70 $\mu g/m^3$. For the lead time of 12 hours, the mean biases of the whole BTH region are 31.55 $\mu g/m^3$ and 54.47 $\mu g/m^3$ at the AT and DT, respectively. Though the meteorology may not the first important, the large difference of PM$_{2.5}$ forecasts caused by the meteorological initial conditions deserve studies as well.

**6. *L 71: This is not applicable. See e.g. Goris and Elbern, GMD, 2015.***

Response: We thank the referee's suggestions. We will rephrase the sentence in the revised manuscript.

**7. *L 80: "or even become worse". Theoretical justification needed.***

**Response**: We have added references here (Yu et al., 2012; Janjic et al., 2017; Zhang et al., 2018). Theoretically, if the observations in the area where the forecast is not sensitive to the initial values are assimilated, the forecasting skills might be improved slightly or neutral. However, in realistic prediction, the imperfect procedure of data assimilation, the observation errors, the unresolved scales and processes in the model and other combination effects may induce more additional errors (Janjic et al., 2017), which may cause the fact that assimilating observations in the area where the forecast is not sensitive to the initial values results in a worse forecast. Anyway, we will present a more accurate description in the revised manuscript.

**8. *L 150: Should be mentioned here that M is WRF and not the CTM, not only at line 172.***

**Response:** We thank your suggestions. However, on L146-157, we would like to introduce the general definition of the CNOP and Eq (2) is the general mathematical expression of the CNOP. In Eq(2), M presents the nonlinear propagator and can be taken as any numerical model. When the CNOP is applied on our study specifically, M means the WRF model, as we stated on Line 172. So on L150, when we present the general definition of CNOP, we think it is more appropriate to define M as nonlinear propagator.

**9. *L 160: Readers might appreciate a literature reference for the energy norm eq. (3) .o***

**Response:** We thank your suggestions and will add the reference (Ehrendorfer et al., 1999) for the energy norm eq. (3) in the revised manuscript.

**10. *L 177: Readers could be hinted that this is a realisation of the maximisation of an Oseledec operator, to familiarize with operator P. In fact, it is nevertheless a linear optimisation, linearized around the "nonlinear trajectory" of the model run, as the adjoint is used.***

**Response:** To compute the CNOP, we use the WRF nonlinear model to estimate the cost function and the adjoint model to produce the gradient of the cost function with respect to the perturbation. Yes, a linear assumption within the neighborhood of each point along the nonlinear trajectory is used when calculating the gradient of the cost function with initial perturbation at this point by adjoint model. However, such a linear assumption will not represent a linear optimization of the CNOP. In fact, the traditional singular vector approach commonly adopted in the previous studies is a linear optimization, which is obtained by a linearized model around the "nonlinear trajectory". The CNOP used here is obtained by running a nonlinear model, where the adjoint is used to calculate the gradient of the cost function with respect to initial perturbations. The CNOP is a nonlinear optimal perturbation, rather than a linear optimal perturbation (see the comparison of CNOP and singular vector in Mu et al., 2003).

***11. L 183: It is pertinent to provide a map of BTH model domain with observation sites here at latest.***

**Response**: We thank your suggestions and will add the map of BTH model domain with observation sites at latest in the revised manuscript (also see Figure R3).

[Figure]

Figure R3 The map of current environmental monitoring stations in the BTH domain.

***12. Fig, 1 caption : Add time instances AT and DT for discussion below by some tags for convenience.***

**Response**: We thank your suggestions and will add AT and DT by the tags for convenience (also see Figure R4).

[Figure]

[Figure]

Figure R4. Time series of the $PM_{2.5}$ concentrations at (a) Baoding station (Hebei Province) and (b) Dongsi station (Beijing city) of observations and simulations initialized by ERA5 and GFS meteorological reanalysis data during the period between 30 November and 4 December 2017.

***13. L 265: Please give a rigorous definition of "CNOP-type error" here, where it is mentioned first! Is it that what has been described in L 297 f?***

**Response**: We thank your suggestions. Yes, it is what described in Line 297. We will add the rigorous definition of "CNOP-type error" on L 265 in the revised manuscript.

***14. L368: Do you mean "differences" instead of "bias"?***

**Response:** Yes. Thanks for your suggestions. We will revise it to "differences" in the revised manuscript.

***15. L 393: On each level (located through the vertical 950, 850, 750 and 500 hPa levels), or only on the most sensitive level? So, are there 15 or 60 observations?***

**Response:** The observations are located at 4 levels, which are 950, 850, 750 and 500hPa. So there are totally 60 observations. We will clarify them in the revised manuscript.

***16. L461 ff: Why is this subsection reasonable, if the algorithm applied is correct, in that it infers optimal conditions? The value of the method is tested against an improvement of an control run, not against climatologically (?) selected other areas. I suggest subsection 4.3 can be omitted.***

**Response:** We think that an approach proposed based on a theory should also be verified numerically, especially by a complex model. In fact, a lot of advanced methods on predictions follow this way to show their usefulness (Zhang et al., 2019). The Region-W and Region-N here were considered being important regions for $PM_{2.5}$ forecasts of BTH region in previous studies. To emphasize the sensitivity identified by CNOP-type errors, we compared the $PM_{2.5}$ forecast skills with observations deployed over the sensitive area and Region-W and Region-N. The comparison will

further illustrate the usefulness of CNOP in identifying the sensitive area for targeted observation and make readers believe that the CNOP is indeed useful in identifying the sensitivity numerically, rather than only in theoretical consideration. So we would like to keep this section.

**17. L 500: How is a decline possible, as the sensitivity is low? It should at least be neutral.**

**Response:** Theoretically, if the observations in the area where the forecast is not sensitive to the initial values are assimilated, the forecasting skills will be improved slightly or neutral. However, in realistic prediction, the imperfect procedure of data assimilation, the observation errors, model errors, the unresolved scales and processes in the model and other combined effects may induce additional errors (Janjic et al., 2017), which may cause the fact that assimilating observations in the area where the forecast is not sensitive to the initial values results in a worse forecast.

**18. L 543: More precisely, it should be especially assigned to stagnant conditions, where a stable layer caps the boundary layer.**

**Response**: We thank your suggestions. We will add more discussions on the stability in the revised manuscript (please see the Comment 2.2). This sentence will be rephrased as well in the revised manuscript.

**19. L 560: What is the sign: truth minus control?**

**Response:** Actually, Figure 9 has 6 subfigures. Figure 9(a, d) present the meteorological condition (including wind and temperature) in the truth run at the AT (a) and DT (d). Figure (b, e) show the meteorological conditions in the control run at the AT (b) and DT(e). Figure (c) and (f) are the forecast differences (control tun minus truth run). We will clarify them in the revised manuscript.

**20. L571: But may increase stability. Further, the interpretation of observed PM values must be supported by information of being dry aerosols or with water component included. The discussion presented should be attentive to that. Otherwise the conclusions may be false.**

**Response:** We thank your valuable suggestions. We will add more discussions on the stability in the revised manuscript (see Comment 2.2). In addition, on Line 571, the interpretation is related to the $PM_{2.5}$ concentration in the two simulations. The $PM_{2.5}$ in the two simulations initialized by ERA5 and GFS is with water component included, so their comparisons are both based on aerosols with water component. We will clarify it in the revised manuscript.

**21. 2L640: What is the "vertical integer of CNOP-type errors"?**

**Response:** We are sorry for the typo. We mean the "vertical integral". We will revise it. The vertical integral of the CNOP-type errors is explained on Line 333-339 in detail.

***22. L662-667: What is the novel message of this passage there than the trivially expected?***

**Response:** The CNOP method is proposed based on an abstract concept model (Line 145-155). Whether it can be applied to identify the sensitive areas in a much realistic model, especially in a complex realistic model, should be verified numerically, despite it is reasonable in theory. Especially, the results obtained by a new method should be compared with the old perspectives to show its superiority. Therefore, the comparisons between the sensitive area identified by the CNOP and the Region-W (Region-N), which of the latter are considered being important regions in the previous studies, will further show the superiority of CNOP-type errors in identifying the sensitive area of meteorological initial fields on $PM_{2.5}$ forecasts. In fact, a lot of advanced methods on predictions follow this way to show their superiority. This is why we made this kind of comparison in the present study.

***23. L 673: "formation of*** $PM_{2.5}$***": Strictly speaking, a different local temperature and humidity dependent secondary formation of*** $PM_{2.5}$ ***must be understood, with equal gaseous precursor emissions. It appears unlikely to me, that this can substantially explain the differences given in Fig. 1. Please clarify.***

**Response:** We agree with the referee that a different local temperature and humidity has a little impact on the secondary formation of $PM_{2.5}$. And our results have also shown that the improvements in the $PM_{2.5}$ forecast skill in assimilation run are mostly attributed to dynamic and thermodynamical reasons (Line 585-602). As for the "formation of $PM_{2.5}$", we admit that we used an improper word, which may mislead the referee. We will rephrase the sentence in the revised manuscript.

Regarding the differences between the observations and the simulations in Figure 1, they, as discussed in Comment 1.1 (critical comments), are due to combined effect of uncertainties of meteorology forecast, emission inventory, and model itself. As for the differences between the two simulations in Figure 1, they, as argued in Comment 3 (Minor issues), are only attributed to the uncertainties in the meteorological initial fields; that is to say, the differences between initial wind, temperature, and moisture cause the substantially difference of the two simulations.

***24. L 687: …"then formulates a theoretical basis to implement practical field campaigns associated with air quality forecasts". Please indicate where this can be found!***

**Response:** Sorry for this ambiguous description. From the results, we showed that the sensitive areas of meteorological initial fields associated with the $PM_{2.5}$ forecasts

indeed exists; meanwhile, these sensitive areas are verified to be valid in improving $PM_{2.5}$ forecast. So the CNOP method is an effective tool to identify the sensitive areas of meteorology on $PM_{2.5}$ forecasts. These results are adequate to encourage us to implement the targeted observations of meteorological initial fields according to the CNOP sensitivity in practical field campaigns and to enhance the $PM_{2.5}$ forecasting skills, thus formulating a theoretical basis in practical field campaigns.

25. *L 697: What does "logistical verification" mean?*

**Response:** We are sorry for the improper use of the word. We would like to present that "the sensitive areas revealed in the present study are still instructive for practical field observations of $PM_{2.5}$ forecasts because of the verifications through a series of OSSEs and reasonable physical interpretation shown in the context". We will rephrase the sentence in the revised manuscript.

**References:**
Bei, N., Wu, k., Feng, T., Cao, k., Huang, R., and Long, X., and coauthors.: Impacts of meteorological uncertainties on the haze formation in Beijing-Tianjin-Hebei (BTH) during wintertime: A case study. Atmos. Chem. Phys. 17, 14579-14591, 2017.

Chen, B., M. Mu, and X. Qin.. The impact of assimilating drop windsonde data deployed at different sites on typhoon track forecasts. Mon. Wea. Rev., 141, 2669–2682, 2013.

Chen, Z., Chen, D., Zhao, C., Kwan, M., Cai, k., and coauthors.: Influence of meteorological conditions on $PM_{2.5}$ concentrations across China: A review of methodology and mechanism, Environ. Int., 139, 105558, 2020.

Duan, W., and Zhou, F.. Non-linear forcing singular vector of a two-dimensional quasi-geostrophic model. Tellus, 65(18452), 256-256, 2013.

Duan, W., Qin, X.. Application of nonlinear optimal perturbation methods in the targeting observations and field campaigns of tropical cyclones. Advances in Earth Science (in Chinese), 37(2):165-176, 2022.

Ehrendorfer, M., R. M. Errico, and K. D. Raeder. Singular-Vector Perturbation Growth in a Primitive Equation Model with Moist Physics. J. Atmos. Sci., 56, 1627-1648, 1999.

Gilliam, R. C., C. Hogrefe, J. M. Godowitch, S. Napelenok, R. Mathur, and S. T. Rao. Impact of inherent meteorology uncertainty on air quality model predictions, J. Geophys. Res. Atmos., 120, 12,259–12,280, 2015.

Janjić, T, Bormann, N, Bocquet, M, Carton, JA, Cohn, SE, Dance, SL, Losa, SN, Nichols, NK, Potthast, R, Waller, JA, Weston, P. On the representation error in data assimilation, Q J R Meteorol Soc. 144: 1257– 1278, 2018.

Lahoz, W., Khattatov, B., Menard, R. (eds) Data Assimilation. Springer Press, Berlin, Heidelberg, 2010.

Li, k., Wang, Z. F., Akimoto, H., Gao, C., Pochanart, P., Wang, X. Q.: Modeling study of ozone seasonal cycle in lower troposphere over East Asia. k. Geophys. Res: Atmos. 112, D22S25, 2007.

Liu, T., Gong, S., He, J., Yu, M., Wang, Q., Li, H., Liu, W., Zhang, J., Li, L., Wang, X., Li, S., Lu, Y., Du, H., Wang, Y., Zhou, C., Liu, H., and Zhao, Q. Attributions of meteorological and emission factors to the 2015 winter severe haze pollution episodes in China's Jing-Jin-Ji area, Atmos. Chem. Phys., 17, 2971–2980, 2017.

Lorenz, E.N.: A study of the predictability of a 28-variable atmospheric model. Tellus, 17, 321-333, 1965.

Mu, M., Duan, W. S., and Wang, B.: Conditional nonlinear optimal perturbation and its applications. Nonlinear Process Geophys., 10: 493–501, 2003.

Mu, M., Zhou, F. F., and Wang, H. L.. A method for identifying the sensitive areas in targeted observations for tropical cyclone prediction: Conditional nonlinear optimal perturbation, Monthly Weather Review, 137(5), 1623-1639, 2009.

Mu, M., W.-S. Duan, Q. Wang, and R. Zhang. An extension of conditional nonlinear optimal perturbation approach and its applications, Nonlin. Processes Geophys., 17(2), 211-220, 2010.

Park, S. K. , Xu, L.  Data assimilation for atmospheric, oceanic and hydrologic applications. Springer Press, 2016.

Privé, N., Errico, R. M. . Some General and Fundamental Requirements for Designing Observing System Simulation Experiments (OSSEs), 2018.

Qin. X., W. Duan, and M. Mu. Conditions under which CNOP sensitivity is valid for tropical cyclone adaptive observations. Quart. J. Roy. Meteor. Soc., 139, 1544–1554, 2013.

Qin, X., Duan, W., Chan, P., Chen, B., Huang, K. Effects of dropsonde data and CNOP sensitivity on the tropical cyclones forecasts in the field campaigns over the western North Pacific in 2020. Adv Atmos Sci., in second review, 2022

Snyder, C. Summary of an informal workshop on adaptive observations and FASTEX. *Bull Am Meteorol Soc,* **77**:   953–61, 1996.

Yu, Y., Mu, M., Duan, W., Gong, T.. Contribution of the location and spatial pattern of initial error to uncertainties in El Niño predictions. Journal of Geophysical Research, 117, C06018, 2012.

Zhang, F., Bei, N., Nielsen-Gammon, k. W., Li, G., Zhang, R., Stuart, A. L., and Aksoy, A.: Impacts of meteorological uncertainties on ozone pollution predictability estimated through meteorological and photochemical ensemble forecasts, J. Geophys. Res., 112, D04304, 2007.

Zhang, H., Wang, Y., Hu, J., Ying, Q., Hu, X.M.. Relationships between meteorological parameters and criteria air pollutants in three megacities in china. Environ.Res. 140, 242–254. 2015.

Zhang, H., Yuan, H., Liu, X., Yu, J., Jiao, Y.. Impact of synoptic weather patterns on 24 h-average $PM_{2.5}$ concentrations in the North China Plain during 2013–2017. Sci. Total Environ. 627, 200–210, 2018.

Zhang, K. , Mu, M. , Wang, Q. , Yin, B. , Liu, S. . CNOP-based adaptive observation network designed for improving upstream kuroshio transport prediction. *Journal of Geophysical Research: Oceans, 124*, 4350-4364, 2019.

---

## Author Response (AR2)

Dear Editor,

we highly appreciate your decision on our manuscript. We believe our study could bring new insights in the study of atmospheric physics and chemistry. For the technical corrections, we have addressed all of them in the revised manuscript. Below we answer the specific corrections point by point. For readability the corrections are shown in bold and italics.

Thank you again for accepting our manuscript, subject to technical corrections.

Sincerely,
Wansuo Duan, on behalf of all the authors

**Corrections:**

1. *Line 76: please change punctuation to "we would propose the following question: can we apply..."*

**Response:** We have changed the punctuation to "we would propose the following question: can we apply the targeted observation strategy to improve …". **Please see Line 75 on Page 3.**

2. *Line 139-140: the added text is ambiguous. I think it should read something like "so that the PM2.5 simulation includes the aqueous component". Please make clear that you are comparing like with like when comparing model to observation: i.e., if the observations are dry particle mass concentrations then the model output has been converted to dry mass concentration.*

**Response:** Sorry for confusing the editor/reviewer. In the NAQPMS model, gas-phase, aqueous and heterogeneous chemistry were included. The thermodynamic equilibrium module ISORROPIA (Nenes et al., 1998) is implemented to simulate the ammonia-sulfate-nitrate-chloride-sodium-water system. In the last round of review, we have thought the $PM_{2.5}$ simulation includes the aqueous component because the aqueous chemistry process is included in the model. Now we check the codes of NAQPMS model carefully and find that the mass of aerosol liquid water is not included in the model output of total simulated $PM_{2.5}$ mass. So the model output of $PM_{2.5}$ mass concentration is actually dry aerosol. The observations are also dry particle mass concentrations. Thus, it is reasonable to compare the observations and the model simulations of $PM_{2.5}$. To make it clear, we have deleted the sentence "so that the PM2.5 simulation includes the aqueous component" on Line 136 and added relevant explanations on Lines 142-144 and 196 in the revised manuscript. **Please see Lines 142-144 on Page 5 and Line 196 on Page 7.**

3. *Line 196: "the distribution...is shown..."*

**Response:** We have changed the sentence to "The distribution of the 80 observation sites within the BTH is shown in Figure 1". **Please see Lines 193-194 on Page 7.**

4. *Figure 1 caption. Explain the boundary lines in the figure (are they local authority boundaries for instance?) and make clear any physical boundaries (eg coast).*

**Response:** To make clear the boundary lines in Figure 1, we have updated the figure and modified the caption. The caption has been modified to "Figure 1 The map of current environmental monitoring stations (hollow circles) within the BTH domain. The black line presents the boundary of province in China, and the thick black line presents the coastline. The boundaries of the Beijing City, Tianjin City and Hebei province are marked in red.". **Please see Figure 1 and Lines 205-207 on Page 8.**

5. *Line 274: delete "much"*

**Response:** We have deleted the word "much". **Please see Line 275 on Page 11.**

6. *Line 286: please change "the unavailable observations" to "the lack of available observations"*

**Response:** We have changed the "unavailable observations" to "the lack of available observations". **Please see Line 287 on Page 11.**

7. *Line 629: make clear whether your observations are for dry or wet aerosol mass concentrations. If the former, humidity will not affect the value.*

**Response:** Sorry for the wrong expressions. The observations are for dry aerosol mass concentrations. The hygroscopic growth of aerosol particle would not affect the mass value. We have deleted the sentence and added "the PM2.5 observations are for dry mass concentrations" in Line 196. **Please see Line 196 on Page 7 and Lines 610-611 on Page 28 in the revised manuscript.**

8. *Line 802 "filed" should be "field"*

**Response:** Sorry for the typo. We have changed the "filed" to "field". **Please see Line 765 on Page 35.**

**Reference:**

Nenes A, Pandis SN, Pilinis C (1998). ISORROPIA: A new thermodynamic equilibrium model for multiphase multicomponent inorganic aerosols, Aquat.Geoch., 4, 123-152.

**Response to Reviewers:**

We highly appreciate the two anonymous reviewers who provided constructive comments, that greatly improved the overall quality of the paper. For the Response to Reviewer #2, please turn to **Page 27**.

**Response to Reviewer #1**

We would like to thank the referee for reviewing the manuscript and providing the valuable comments and suggestions. We are sorry that for some sentences we did not make them clear in the manuscript. We have updated our manuscript following the suggestions. Below we answer the specific comments point by point. For readability the comments are shown in bold and italics.

**Review Comments:**

*The authors study the optimization of observation locations (targeted observation) to achieve an improved forecast for particulate matter. Interestingly they provide an example of a severe haze event in the Beijing area where early warnings by the authorities failed to be timely issued. This topic has attracted interest since more than a decade ago, in recent years also in the realm of atmospheric chemistry. It is strongly linked with research on predictability, observability and data assimilation. A wealth of methods has been devised, or derived from existing techniques found in the aforementioned realms.*

*In their study "Toward target observations of the meteorological initial state for improving the PM2.5 forecast of a heavy haze event that occurred in the Beijing-Tianjin-Hebei region" by Yang Lichao, Duan Wansuo, Wang Zifa, and Yang Wenyi addressed the optimisation of measurement deployment for full and atmospheric chemistry application by devising a meteorological problem of optimal measurement dislocation. In my review I question this strategy with some detail, encouraging the authors to refute my demurs.*

*Methodology:*

*The motivation of the work where's to improved aerosol forecast which failed significantly in the case study selected not fault booked. So, there might be different reasons for this failure notably a faulty emission inventory or degraded weather forecasts. In their approach the authors seeked the reason only in the weather forecast. Hence, they tried to improve the prediction by better located meteorological observations which they assimilated to obtain better initial values for the forecast. The other option let the forecast deficiencies might result from faulty emission inventories was not considered is out giving any evidence of reason.*

*The authors quite deliberately declared a better forecast resulting from era 5 reanalysis to be the truth while another one from GSE was declared control which verse aspired to be improved by additional and optimally located observations. The resulting simulation product provided the improved forecast in relation to the control room but not as good as the truth run identity fight before. The statistical analyses of the assimilation run were then provided as quantitative proof of concept.*

**My critique addresses several items.**

*1.1 Firstly, how would the method provide reasonable results if not the meteorological forecasts are deficient but the emission inventory, which are in fact often poorly known. Figure 1 of the manuscript does not give any indication that the major discrepancy is only due to meteorological prediction flaws.*

**Response:** We agree that the uncertainties occurring in both emissions and meteorological fields cause the forecast uncertainties of PM 2.5. In fact, the errors of model itself are also influencing the forecasting uncertainty of $PM_{2.5}$. Furthermore, the forecasting errors caused by these uncertainties are interactive and it is difficult to quantify exactly their respective contribution. In Figure 1, it, as the reviewer said, does not give any indication that the major discrepancy is only due to meteorological prediction flaws. In fact, we did not state that the $PM_{2.5}$ forecasting uncertainties are solely from meteorological forecasting prediction flaws. Especially, when comparing the simulations with observations in Figure 1, we have to say that the $PM_{2.5}$ forecasting uncertainties are from the combined effect of meteorological prediction flaw, model uncertainties, and emission inventory uncertainties. In the present study, we read a lot of previous studies which have demonstrated the important role of meteorology field on $PM_{2.5}$ forecasts in the BTH region (Bei et al., 2017; Gilliam et al., 2015; etc.); meanwhile, our results also showed that the $PM_{2.5}$ forecasting uncertainties are sensitive to the initial uncertainties of meteorological field (see the two simulations in Figure 1, which were obtained by integrating the WRF-NAQPMS with different initial meteorological fields but with the same emission inventory and model) despite they are not necessarily the most important error source of $PM_{2.5}$ forecasting uncertainties; and we decided to focus on the meteorology uncertainties in the present study, but leaving uncertainties of model errors and emission inventory to be explored in the future. **Please see Lines 787-800 on Page 36.**

To identify the initial error effect of meteorology, we adopt the idea of Lorenz (1965) on two types of predictability problems. The first focuses on the effect of initial error growth with an assumption of perfect model while the second is to assume a perfect initial field for exploring the effect of model error growth. The present study follows the idea of the first type of predictability problem; and to separate the initial meteorological error effect, an assumption of perfect model is done; for the WRF-NAQPMS model, we acknowledge that the emission inventory is also uncertain; therefore, we further keep the emission inventory in all the simulations the same to

extract the sensitivity of initial meteorological uncertainties by investigating the difference between the two simulations. Similar doings are also used in the previous studies for air quality simulations or forecasts, e.g. Gilliam et al. (2015) and Bei et al. (2017), etc. However, whichever it is initial field or model, even emission inventory, it certainly consists of uncertainties. So, in the present study, to make much realistic we have to take the better simulation initialized by the ERA5 (which was obtained by assimilating all available observations with more advanced model by ECWMF and can therefore be a much better approximation to the truth except for observations) as "truth run" because we cannot obtain observations from the Monitor center and the worse simulation initialized by GFS forecast data as "control forecast", where these two simulations have the same emission inventory and use the same model; so the difference between them reflects the sensitivity of forecast uncertainties of $PM_{2.5}$ on initial meteorological field. We regard such sensitivity as an approximation to the real sensitivity. Therefore, when one computes the CNOP superimposed on the simulation initialized by ERA5, the CNOP may reveal a sensitive area for targeted observation, which, actually, is an approximation to the real sensitive area. If such approximation is valid, then for any forecast, preferentially assimilating additional observations in the sensitive area (i.e. the targeted observations) will help improving the forecasting skill much greatly. In fact, we need to verify the validity of the sensitive area in advance and then implement the additional observations in the sensitive area in field campaign. The validity of the above approximate sensitive area is often tested by prescribing a good simulation to observation (for example, the simulation initialized by ERA5) and then assimilating the simulated observations located in the sensitive area to examine whether a forecast (for example, the control forecast initialized by the GFS ) will be much closer to the good simulation, which, actually, is a kind of observation system simulation experiment (OSSE; see Masutani et al., 2010; Qin et al., 2013). Although the studies on the targeted observations here are associated with hindcasts of $PM_{2.5}$, it is still difficult to obtain the meteorological observations from the Monitor Center; therefore, we can only assimilate the simulated "observations" from the simulation initialized by the ERA5 to examine the validity of the sensitive area. If the sensitive area is verified to be valid along this thinking, it can be inferred that assimilating the real targeted observations to the initial field of the meteorology of the control forecast would improve the meteorological field forecasting and then the $PM_{2.5}$ forecasting greatly against the observations. **Please see lines 270-286 on Page 11.**

***1.2 In addition, if both forecasts, that is the truth and the control run, suffer from the same problem, as for example poor boundary layer height simulations, then the method proposed incapable to give any evidence of any source of error.***

**Response:** We thank the valuable comments. When both forecasts, that is the truth and the control run, suffer from the same problem, as for example poor boundary layer height simulations, then associated forecast uncertainties are included in the difference between simulations and observations. The differences of the two simulations in Figure 1 can only indicate the sensitivity of $PM_{2.5}$ forecast to the accuracy of initial meteorological field because the two simulations used the same

models and emission inventory but different initial meteorological fields; and the CNOP only considers such sensitivity from initial uncertainties. If one will identify the role of boundary layer uncertainties in yielding $PM_{2.5}$ forecasting uncertainties along similar thought presented in this study, the $PM_{2.5}$ simulations should be made with the same initial meteorological field and the same emission inventory but different boundary layer schemes to reveal the role of boundary layer uncertainties. Based on these simulations, an extension to the CNOP method, CNOP-parametric perturbation (CNOP-P; Mu et al, 2010) or nonlinear forcing singular vector (Duan and Zhou, 2013), can be used to identify the sensitivity of boundary layer uncertainties. We thank the referee for providing us a great research idea for our future studies. The related discussions have been added in the revised manuscript. **Please see lines 773-780 on Page 35.**

*2.1 Secondly, I put the assimilation procedure in question. So let us assume the authors are right in their suspicion, that the meteorological forecast is the source of misprediction of the aerosol concentrations. A sound synoptic description of the weather situation and its evolution is lacking as are appropriate surface weather charts.*

**Response:** As argued above, we did not emphasize that the metrological forecast uncertainties is the unique source of the misprediction of the $PM_{2.5}$ event. In the present study, we find that the $PM_{2.5}$ forecasting is also sensitive to the initial uncertainties of meteorological field (see the two simulations in Figure 1) and investigate the role of the targeted observation for meteorological field in improving $PM_{2.5}$ forecasting skill, but leaving the uncertainties of model itself and emission inventory to be explored in the future. In addition, it is very difficult for us to obtain the meteorological observations from the Monitoring centers; then we adopt the more efficient ERA5 data that one often uses as initial value for the model to determine the approximate sensitive area. Therefore, it is hard to plot the weather charts corresponding to the difference between simulation and observation. Nevertheless, when we evaluated the role of targeted observations of meteorological field in the manuscript, we plotted the weather charts before and after assimilating targeted observations and showed the weather conditions for improving $PM_{2.5}$ forecasting skill **Please see lines 649-691 and the Figures 12-13 on Page 29 to 32.**

*2.2 In addition, a discussion on the boundary heights and stability would be in place, as these are a critical parameters, controlling the capture of emissions.*

**Response:** We thank your valuable suggestions. We have added more discussions on the boundary heights and stability in the revised manuscript. **Please see the Lines 627-637 on Page 28, Lines 657-663, Lines 669-670 on Page 30, Lines 773-780 on Page 35.** Specifically, during the accumulation process, both the truth run and the control run are able to simulate the temperature inversion layer, which prevents vertical dispersion of pollutants and promotes the accumulation of surface $PM_{2.5}$. For the forecasts at the AT, the truth run has forecasted 0.11K/100m vertical temperature

inversion layers at Dongsi station in Beijing City (the temperature arises 0.11K every 100m), whist the control run has forecasted 0.05K/100m. The mean lapse rate simulated by the truth run over the BTH region is 0.03K/100m and the control run has forecasted a 0.002K/100m. So the truth run has a more stable thermodynamic condition. After the assimilating the targeted meteorological variables to the control run, the assimilation run has forecasted 0.06K/100m temperature inversion layers at Dongsi station and the mean lapse rate over the BTH region has reached to 0.004K/100m. The slightly improved thermodynamic conditions further result in the modifications of the boundary layer structure featuring a decreased PBL height. The mean boundary layer height over the BTH region has decreased from 261m in the control run to 256m in the assimilation run, which also contributed to the increased ground level $PM_{2.5}$ pollution and improved the $PM_{2.5}$ forecast skill in the assimilation run.

**2.3 What happens, if both truths run and control run err with the stability in the same way, but differ in , say, as in this paper, in the horizontal wind direction? In this case, the CNOP type error is critically incomplete.**

**Response:** We think the comment may consist of two questions. The first is "What happens if both the truth run and control run suffer from error with stability in the same way?". Actually, when both the truth run and control run suffer from the same problem, then associated forecast uncertainties are included in the difference between simulations and observations. As we explained in Comment 1.1, the differences between the simulations and observations may be attributed to the combined effect of the uncertainties of meteorology, emission inventory and model itself. In the present study, we only focus on the effect of meteorological initial errors on the $PM_{2.5}$ forecasts. To separate the initial effect, we adopt the same model and same emission inventory but different meteorological initial fields to verify the sensitivity of meteorological initial conditions on $PM_{2.5}$ forecasts. And the CNOP, which can represent the most sensitive initial errors, is calculated based on the better simulation, as we explained in Comment 1.1. As for the error related to the stability in the truth, it may be attributed to the initial errors existing in the reanalysis (ERA5) or the model itself. In that case, we admit that the CNOP-type error based on the "truth run" may be incomplete, since the reanalysis is not the real truth and the model is not a perfect model. However, as we explained in Comment 1.1, since we are not able to obtain the meteorological observations from the Monitoring centers, we have to adopt the more efficient ERA5 data as initial values and more advanced WRF model to determine the approximate sensitive area. In this situation, the validity of the sensitive areas should be examined; and actually, it has been verified by a series of OSSEs as described in comment 1.1 in the present study. **Please see lines 270-286 on Page 11.**

The second question is "the components of CNOP type error may be incomplete if only the horizonal wind is included but excluding the stability". Actually, the stability is not a direct meteorological variable, but a variable related to temperature. Meteorological conditions, such as the wind, temperature, relative humidity, stability,

boundary layer height, precipitation all have significant impacts on the regional $PM_{2.5}$ forecasts in CTMs (Godowitch et al., 2011). Chen et al., (2020) reviewed hundreds of papers on the meteorological factors on $PM_{2.5}$ concentrations for the BTH region, and they concluded that the wind and humidity are the dominant meteorological factors (Table 1 in Chen et al., (2020)). In our study, to include as many meteorological variables as possible as the components of CNOP type error, we use the total energy norm, which includes the wind, temperature, water vapor mixing ratio and pressure perturbations, to constrain the CNOP. According to the results obtained by Chen et al., (2020), we think the variables considered in the CNOP are adequate. **Please see lines 168-169 on Page 6, Lines 773-780 on Page 35.**

*2.4 The method proposed by the authors is designed to deploy 15 different observation locations which might be the key to the sufficiently well performing forecast. So, all in all they select 15 times four height levels times 4 meteorological parameters that means individual 240 observations and tested the performance of these idealized network with respect to varied distances. In fact this is a variable the radiosonde network or air borne drop sonde area placed windward of the area of interest to be predicted. Leaving aside the practicability, I put into question the benefit for improved forecast with 3D-var by localized observations, given the synoptic balance conditions to be fulfilled. The authors result indicates this: Looking at Fig. 9, panels a) and b), it appears to be likely that the eastern side of a high pressure system (northerly winds) at the eastern side of the panels is shifted further eastbound in the truth run (a), than in the control run. It is not possible to correct this error by assimilation of data from a localized observation network alone.*

**Response:** Yes, the reviewer got a right conclusion. It could be impossible to correct this error by assimilation of data from a localized observation network alone. In fact, the "target observation" is to deploy few additional observations in some localized areas (sensitive areas) where the additional observations are expected to just have a large contribution to reducing the prediction error in the given **verification area** (Snyder., 1996), rather than other areas. In Figure 9, we agree that the eastern side of a high pressure system at the eastern side is shifted further eastbound in the truth run than in the control run; and assimilating the targeted observation does not correct the error. Actually, the sensitive area for targeted observation in the present study is determined on the verification area, i.e. the BTH region (i.e. the black rectangle in the figure 9). That is to say, assimilating the targeted observation is to preferentially improve the meteorological field in the verification area- BTH region, rather than other regions. Therefore, it is reasonable, as the reviewer pointed, the forecast errors in the high pressure system cannot be corrected by targeted observation. This may also indicate that the high pressure system at the eastern side does not play the dominant role in influencing the $PM_{2.5}$ in BTH in this event. **Please see lines 61-67 on Page 3.**

*3. How should the set-up with two model runs operate practically? How do we find the "real truth"? In fact, the only thing what can be done is to achieve an optimal meteorological forecast in general, with all available observations. After this, optimal sensitivity areas can then be identified for chemical concentration measurements, not for meteorological observations, because the truth is not known. Recommendation: To account for these problems, the authors are encouraged to change their validation strategy and conduct numerical experiments, where the emission inventory is taken as true and a nature run produces artificial ("synthetic") aerosol concentration observations, which then are to be reproduced by the proposed targeted observation procedure, analog to Observation System Simulation Experiments (OSSE) made in data assimilation developments.*

**Response:** We thank your valuable comments. Yes, to apply the targeted observation procedure on the emission inventory is a good idea. As we discussed in **Lines 790-800 on Page 36**, "targeted observation may be a better strategy to improve the quality of emissions, and the determination of sensitive areas of emissions is certainly important…. Then, targeted observations of emissions based on the CNOP approach are expected to be studied for air quality forecasts in the future.". That is to say, we have realized that this is also an important work. Actually, both the emission and meteorology may substantially influence the $PM_{2.5}$ forecast. The study of targeted observations on both meteorology and emission is meaningful, but that would be accomplished step by step. In the current study, we focus on the targeted observations on meteorology forecasts first. Our current study represents a first step that the CNOP algorithm of targeted observations is applied to the studies of air quality forecasts. Then, target observations of emissions identified by CNOP are expected to be studied for air quality forecasts in the near future.

As we mentioned in Response to Comment 1.1, when we compute the CNOP superimposed on the simulation initialized by ERA5, the sensitive area is actually an approximation to the real sensitive area. If such approximation is valid, for any forecast, preferentially assimilating the additional observations on the sensitive area will help improve the $PM_{2.5}$ forecasts greatly. In the practical field campaigns, we also need to verify the validity of the sensitive area in advance and then implement the deployment of additional observations in the sensitive area. The validity of the above approximate sensitive area is often tested by prescribing a good simulation to observation (for example, the simulation initialized by ERA5) and then assimilating the simulated observations located in the sensitive area to examine whether the forecast (for example, the control forecast initialized by the GFS ) will be much closer to the good simulation, which, actually, is a kind of OSSEs (see Masutani et al., 2010; Qin et al., 2013). Although the studies on the targeted observations here are associated with hindcasts of $PM_{2.5}$, it is still difficult to obtain the meteorological observations from the Monitor Center; therefore, we can only assimilate the simulated "observations" from the simulation initialized by the ERA5 to examine the validity of the sensitive area. If the sensitive area is verified to be valid by the above thought, it

can be inferred that assimilating the real targeted observations to the initial field of the meteorology of the control forecast would also improve greatly the PM$_{2.5}$ forecast against the real truth (or observations). **Please see Lines 271-286 on Page 11 and Lines 392-400 on Page 19.**

According to the above interpretation, in field campaigns one is required to adopt a much better model to obtain a good forecast (for example, the "truth run" in comment 1.1) by assimilating all available observations or by a reanalysis data of high quality (for example, ERA5) and identify the sensitive area by calculating the CNOP of the good forecast. If the sensitive area is valid, it is conceivable that, for any forecast, assimilating the real additional observations on the sensitive area will improve the PM$_{2.5}$ forecasting skill much greatly against the observations. Even for the "truth run" above, which is still of obvious differences from the real observations (see the observation and simulation initialized by the ERA5 in Figure 2), assimilating the real targeted observations to the "truth run" will also make the PM$_{2.5}$ forecasts away from the "truth run" but close to the observations. Such idea has been applied on the field campaign for typhoon forecasting and the useful real-time targeted observations are obtained, which has been verified to be able to improve greatly the typhoon forecasting skill (Qin Xiaohao, Duan Wansuo, Pak Wai Chan, Chen Boyu, Kang-Ning Huang, Effects of dropsonde data in field campaigns on forecasts of tropical cyclones over the western North Pacific in 2020 and role of CNOP sensitivity, Advances in Atmospheric Sciences, 2022, accepted). **Please see Lines 757-767 on Page 35.**

As for the studies of targeted observations on aerosol concentrations, the strategy suggested by the reviewer is much realistic, because the emission inventories can be taken as real observations, rather than the simulated observations generated from the model like what we did in the present study due to the unavailable observations. We thank the referee's suggestion and will adopt the great research idea in the study of emission uncertainties. **Please see Lines 787-800 on Page 36.**

**Literature:**

4. *The authors claim that they are the first to transfer the method of targeted observation to atmospheric chemistry, which does not at all apply! Regrettably, it appears that the authors are not aware of the number of meanwhile growing set of papers on this very matter. Some relevant papers are given here for convenience. Studies focusing on atmospheric chemistry observation targeting, explicitly or implicitly, are indicated by boldface letters, and merit special attention. As the authors focus on meteorological targeted observations I include several other studies on that issue, which might also be considered.*

*Recommendadtion: We strongly recommend who review this literature given below.*

Bellsky T, Kostelich EJ, Mahalov A (2014) Kalman filter data assimilation: targeting observations and parameter estimation. Chaos 24(2):024406. https://doi.org/10.1063/1.4871916

Berliner, L. M., Lu, Z., and Snyder, C.: Statistical design for Adaptive Weather Observations, J. Atmos. Sci., 56, 2536–2552, 1998.

Bishop, C. H. and Toth, Z.: Ensemble Transformation and Adaptive Observations, J. Atmos. Sci., 56, 1748–1765, 1998.

Buizza, R., Cardinali, C., Kelly, G., and Thepaut, J. N.: The value of targeted observations, ECMWF Newsletter, 111, 11–20, 2007.

Daescu, D. N. and Carmichael, G. R.: An Adjoint Sensitivity Method for the Adaptive Location of the Observations in Air Quality Modeling, J. Atmos. Sci., 60, 434–450, 2003.

Goris N, Elbern H (2013) Singular vector decomposition for sensitivity analyses of tropospheric chemical scenarios. Atmos Chem Phys 13:5063–5087. https://doi.org/10.5194/acp-13-5063-2013

Goris N, Elbern H (2015) Singular vector based targeted observations of chemical constituents: description and first application of EURAD-IM-SVA. Geosci Model Dev 8:3929–3945. https://doi.org/10.5194/gmd-8-3929-2015

Khattatov, B. V., Gille, J., Lyjak, L., Brasseur, G., Dvortsov, V., Roche, A., and Waters, J.: Assimilation of photochemically active species and a case analysis of UARS data, J. Geophys. Res., 104, 18715–18738, 1999.

Liao, W., Sandu, A., Carmichael, G. R., and Chai, T.: Singular Vector Analysis for Atmospheric Chemical Transport Models, Mon. Weather Rev., 134, 2443–2465, 2006.

Szunyogh I, Toth Z, Emanuel KA, Bishop CH, Woolen J, Marchok T, Morss R, Snyder C (1999) Ensemble based targeting experiments during FASTEX: the impact of dropsonde data from the Lear jet. Q J R Meteorol Soc 125:3189–3218. https://doi.org/10. 1002/qj.49712556105

Wu X, Jacob B, Elbern H (2016) Optimal control and observation locations for time-varying systems on a finite-time horizon. SIAM J Control Optim 54(1):291–316. https://doi.org/10.1137/15M1014759.

Wu, Xueran; Elbern, Hendrik, Jacob, Birgit; The assessment of potential observability for joint chemical states and emissions in atmospheric modelings, Stochastic Environmental Research and Risk Assessment https://doi.org/10.1007/s00477-021-02113-, 2022.

**Response:** We are sorry that we did not conduct a fully literature review. We thank the referee for listing the related papers, especially the publications of targeted observation in atmospheric chemistry. We have read all the recommended literature carefully and cited them in the revised manuscript. **Please see Lines 791-796 on Page 36.** The sentences such as "the first application on the atmospheric chemistry" have be modified in the revised manuscript. **Please see Line 696-698 on Page 33, Lines 757-761 on Page 35.**

5. *The paper is in fact about an algorithm for targeted observations. As such no results for atmospheric chemistry per se are offered and can be expected. So it is suggested to submit the manuscript to GMDD rather than ACPD.*

**Response:** We thank the referee's comment. However, we do not think our paper is just an algorithm or a technical paper. In fact, it is a study on the application of the CNOP algorithm to identify the sensitive area for targeted observations of

meteorological initial fields associated with the $PM_{2.5}$ forecasts. The relevant physical process and explanations on how the targeted observations of meteorological initial conditions in the sensitive area leads to the improvement of the $PM_{2.5}$ forecasts is also investigated in the paper. According to the scope of ACP, our study contributes to understand how the meteorological initial states influence the transportation and accumulation of $PM_{2.5}$ concentrations by atmospheric dynamic and/or heating, etc., which belongs to the study of atmospheric physics processes related to the $PM_{2.5}$ variations.

Our study also provided a potential application prospect in identifying the sensitive area for emission inventories. Although other methods such as singular vector, adjoint sensitivity, and ETKF provided by literatures listed by the referee can also be used, they are approaches of linear approximation. The CNOP considers fully effect of nonlinearity and overcomes the linear limitation of the traditional approaches and presents the most sensitive initial perturbation (Mu et al., 2003), then being able to effectively identify the sensitive area for targeted observations. This argument has been verified by a lot of studies (Mu et al., 2009; Chen et al., 2013). Therefore, if this article is published in ACP, it can be expected that CNOP algorithm and its potential applications on emission inventories will be known by more researchers in the field of atmospheric chemistry. It is also expected that the CNOP can be a useful approach to addressing problems of air quality forecasts. **Please see Lines 787-800 on Page 36.** So it is very anticipated that this article can be published in ACP after addressing all concerns of reviewers.

**Specific remarks:**

1. *The authors should use the term targeted observations throughout, as in the paper by Majumdar. (Not target observations. Majundar made only deviations by grammatical reasons.)*

**Response:** We have modified "target observations" to "targeted observations" throughout the paper. **Please see the title, Line 11 on Page 1, Line 67 on Page 3, etc**.

2. *Discussion of emission inventory uncertainty and other uncertainty sources. There is a well-established corpus of literature addressing uncertainty sources of chemistry transport model, where meteorological uncertainties are only one among others. The authors' decision to solely focus on meteorology needs a sound quantification.*

**Response:** As we explained in Comment 1.1, we agree that the uncertainties occurring in emissions, meteorological fields, model itself and other sources cause the forecast uncertainties of $PM_{2.5}$. We noticed that a lot of papers emphasized the important role of meteorological field in transporting $PM_{2.5}$ and yielding $PM_{2.5}$ forecasting uncertainties in the BTH region (Bei et al., 2017; Gilliam et al., 2015; Chen et al., 2020). **Please see Lines 40-50 on Page 2.** Furthermore, we also find that

the $PM_{2.5}$ forecasting in this heavy pollution event concerned in the present study are also sensitive to the initial uncertainties of meteorological field (**see the two simulations in Figure 2**), despite meteorological uncertainties could not be the most important contributor to the $PM_{2.5}$ forecasting uncertainties. Therefore, we first focus on the meteorological uncertainties in the present study. This does not mean that the uncertainties of model itself and emission inventory are not important, but we should address these uncertainties step by step. In the present study, we first pay attention to meteorological uncertainties and leave uncertainties of model and emission inventory to be explored in the future. It is expected that the combined effect of uncertainties of model, meteorological, and emission inventory can be finally addressed. **Please see Lines 787-800 on Page 36.**

3. *What is the assumed dominant composition of PM 2.5 matter (mineral dust, secondary anthropogenic, …) , and is the emission inventory sufficiently resolved by 30 km grid size?*

**Response:** The components of $PM_{2.5}$ simulation here include black carbon (BC), organic carbon (OC), secondary inorganic aerosol (sulfate, nitrate, ammonium) and primary $PM_{2.5}$ emitted directly from various sources. The dominant composition of $PM_{2.5}$ varies with regions and periods. During this event, the dominant compositions are nitrate and organic carbon. We have given a definition of $PM_{2.5}$ matter in the revised manuscript. **Please see Lines 141-144 on Page 5.**

As we discussed in **Lines 780-784 on Page 35**, we have realized that the resolution 30km is relatively low for $PM_{2.5}$ forecasts. Nevertheless, even thus, the simulation initialized by ERA5 can well represent variability of the accumulation and dissipation processes of $PM_{2.5}$ despite the uncertainties against the observations (Figure 1). It indicates that the emission inventory adopted here can be resolved. Here we also present the spatial distribution of daily average $PM_{2.5}$ concentrations of observation and ERA5 simulation on Dec, $1^{st}$ (Figure R1). It shows that the ERA5 simulation is able to produce the spatial distribution of the observed $PM_{2.5}$. This also indicates the emission inventory at 30km is acceptable for this heavy pollution event.

We agree that the emission inventory will be better resolved in a higher resolution. So we have the related discussion in the manuscript. As seen on **Line 784 on Page 36**, "a WRF-NAQPMS model with much higher resolution will be used in next study on $PM_{2.5}$ forecasting."

[Figure]

Figure R1 The spatial distribution of the daily average PM$_{2.5}$ concentrations of observation (circle) and ERA5 simulation (shaded) on Dec 1st. (unit: μg/m$^3$)

**4. *Why is the targeted observation approach not applied to emission sources? It is well understood that emissions are rarely measurable (eddy covariance towers are a practically unavailable exemption). Yet concentration observations in the vicinity of sources could be exploited instead with some benefit.***

**Response:** We thank your valuable suggestions. Yes, it is important to apply the targeted observation approach to emission sources, as we discussed in **Lines 787-800 on Page 36** in the manuscript. Actually, both the emission and meteorology may substantially influence the PM$_{2.5}$ forecast. The study of targeted observations on both meteorology and emission is meaningful and would be accomplished step by step. As we explained in Comment 1.1, a lot of previous studies have emphasized the important role of meteorological field on PM$_{2.5}$ forecasts in the BTH region (Liu et al., 2017; Zhang et al., 2018). **Please see Lines 40-50 on Page 2**. Also we find that the PM$_{2.5}$ forecasts concerned in the present study are sensitive to the meteorological initial conditions (**see the two simulations in Figure 2**), which indicates the important role of meteorology forecast accuracy in improving PM$_{2.5}$ forecast. Even though the meteorology may not be the first factor that influences the PM$_{2.5}$ forecasts, the large differences between the two simulations also motive us to apply the target observation strategy to improve the accuracy of the meteorological forecasts, then the PM$_{2.5}$ forecasts. So in the current study, due to important role of meteorology, and also as the first attempt to apply CNOP sensitivity to PM$_{2.5}$ forecasts, we investigate the targeted observations on meteorology forecasts associated with PM$_{2.5}$ forecasts. Then, as the referee suggested, to apply the targeted observation on emission sources, such as locating the eddy covariance tower to get the concentration observations, is a great research idea and motivate us to carry on our studies on the emission uncertainties in the near future. **Please see Lines 222-224 on Page 9 and Lines 787-800 on Page 36.**

1. *Title: The typical term is. Targeted observations. It is recommended, to adapt accordingly.*

**Response:** Thank you very much for your suggestion. As expected, we have modified the "target observation" to "targeted observation" in the revised manuscript. **Please see the title, line 11 on Page 1, line 67 on Page 3, etc.**

2. *Feedback emissions-meteo around L 545 mentioned, but emission inventory uncertainties poorly addressed.*

**Response:** We thank the referee's comments. In the present paper, as we argued above, we only focus on the sensitivity of meteorological initial conditions on $PM_{2.5}$ forecasts, leaving the studies of emission uncertainties to be explored in the future. In the OSSEs we keep the emission inventory in all the simulations the same [as did in Gilliam et al. (2015), Bei et al. (2017), etc.]; So the uncertainties among the $PM_{2.5}$ simulations in the present study are only from the differences of meteorology forecasting **(Please see Lines 222-224 on Page 9)**, and in the Interpretation section, we have to only focus on explaining how improving the meteorological initial condition influence the $PM_{2.5}$ simulations. We have followed the referee's suggestions and add more discussions on the emission uncertainties in the "Summary and Discussion" section of the revised manuscript. **Please see Lines 787-800 on Page 36.**

3. *Fig. 9 Substantial differences between truth and ctrl run. How is this possible? Could be phase error. This renders the assimilation of artificial data critical as this local information is inconsistent with the synoptic situation (imbalance).*

**Response:** The differences in Figure 9 are dependent on the meteorological initial conditions, since both the truth run and control run use the same model and emission inventories. The initial meteorological condition for the truth run is generated by the ERA5 reanalysis data, which is the newest generation ECMWF reanalysis data which combines vast amounts of historical observations into global estimates using advanced modelling and data assimilation systems. The initial meteorological condition for the control run is generated by the NCEP GFS, which is the forecast data generated by a global forecast system in NECP. The forecast data consist of larger uncertainties and very different from those of ERA5. Figure R2 shows the initial condition of WRF simulations generated by the ERA5 and NCEP GFS at the AT and DT with lead times of 24 hours. A substantial difference between the two initial conditions exists, so it is reasonable that difference of meteorological forecasts at the AT and DT between the control and truth run is large.

As for the imbalance the referee has pointed, in our opinions, does not exist in our study. Though only the observations in the sensitive area are assimilated, the initial condition outside of the sensitive area will be coordinated through the data

assimilation technique. Both the initial states before and after the assimilation are the solutions to the model, they are definitely be balanced. So the assimilation of artificial data will not be imbalanced with the synoptic situation.

[Figure]

Figure R2 The initial condition of wind (vector, unit: m/s) and temperature field (shaded, unit ) for the forecast at the AT of the (a) truth run (b) control run. (c-d) are the same as (a-b) but for the forecasts at the DT. As we can see, a substantial difference between the two initial conditions exists, so it is reasonable that difference of meteorological forecasts at the AT and DT between the control and truth run is large.

**4. *As meteo forecast deficits are assumed for PM prediction flaws: Validation against meteo data lacking. Why?**

**Response:** As we explained in Comment 1.1 and Comment 3 (critical comments), the calculated sensitive area is actually an approximation of the real sensitive area. The validity of the above approximate sensitive area is often tested by prescribing a good simulation to observation (for example, the simulation initialized by ERA5) and then assimilating the simulated observations located in the sensitive area to a bad forecast

(for example, the control forecast) to examine whether the assimilation forecast will be much closer to the good simulation, which, actually, is a kind of OSSEs (see Masutani et al., 2010; Qin et al., 2013). In our study, to verify the validity of the sensitive area, the simulated targeted observations are assimilated to the GFS forecasts to improve their $PM_{2.5}$ forecasts, where the GFS forecasts are taken as the "control run" and those after assimilating targeted observations are regarded as the "assimilation run". If the sensitive area is valid, the $PM_{2.5}$ forecasts in the assimilation run will be much closer to the truth run. It can also be inferred that if the real observations are available, assimilating the real targeted observations to the initial field of the meteorology of the control forecast would improve the $PM_{2.5}$ forecast skill greatly against the observations. In the present study, we will adopt assimilating simulated observations to verify the validity of the sensitive area due to the lack of available observations. **Please see Lines 272-287 on Page 11.**

5. *L 39-50: Do the authors claim that this is valid for their study region, or globally? Most studies point at emission strengths uncertainties. More precisely, the uncertainties of predictions must be pondered with forecast time. On short range forecasts today's meteo forecast uncertainties are small, if not extraneous, when compared with both anthropogenic and biogenic emissions. Please discuss this with more scrutiny.*

**Response:** We thank the referee's comment. The meteorological conditions have a great impact on $PM_{2.5}$ forecasts for our study region. We have emphasized it is valid for our study region in this revised manuscript. **Please see lines 40-48 on Page 2.**

We did not deny the importance of emission uncertainties on $PM_{2.5}$ forecasts. As we argued above, we agree that the meteorology, emission inventories and the model itself all contribute to the $PM_{2.5}$ forecast uncertainties. Due to the former studies (Bei et al., 2017; Gilliam et al., 2015) and our results (see the two simulations in Figure 2), we first investigate the role of the meteorological targeted observation in improving $PM_{2.5}$ forecasts in the present paper although the meteorology may not be the most important for the $PM_{2.5}$ forecasts, and leave the studies on the emission uncertainties in the near future. In any case, the effect of meteorological, model itself, and emission inventories uncertainties will be studies step by step. As the first attempt to apply the CNOP sensitivity on $PM_{2.5}$ forecasts, the successful application of CNOP in meteorological targeted observations will also inspire us to apply the CNOP method in the study of emission uncertainties in the future. **Please see Lines 787-800 on Page 36.**

We agree that the uncertainties of predictions are pondered with forecast time. For the event we studied, we showed the spatial distributions of the $PM_{2.5}$ forecast errors in the control run at the AT and the DT with the lead times of 24 hours in Figure 7. If taking the absolute value of the biases, then the mean biases of the whole BTH region are 34.22 and 64.13 ug/m3 at the AT and DT, respectively. In some areas of BTH, the errors are more than 70 μg/m$^3$. For the lead time of 12 hours, the mean

biases of the whole BTH region are 31.55 $\mu g/m^3$ and 54.47 $\mu g/m^3$ at the AT and DT, respectively. Though the meteorology may not the first important, the large difference of PM$_{2.5}$ forecasts caused by the meteorological initial conditions deserve studies as well. **Please see Lines 403-405 on Page 19 and Figure 8 on Page 20.**

*6. L 71: This is not applicable. See e.g. Goris and Elbern, GMD, 2015.*

Response: We thank the referee's suggestions. We have rephrased the sentence in the revised manuscript as "As we stated above, the meteorological initial fields have great impacts on the PM$_{2.5}$ forecasts of the BTH region (Bei et al., 2017; Liu et al., 2018); meanwhile, our results also showed that the PM$_{2.5}$ forecasts are sensitive to the uncertainties of meteorological initial conditions (see Section 3.1). Based on these findings, we would propose the following question, can we apply the targeted observation strategy to improve the meteorological condition forecasts, which then further improve the PM$_{2.5}$ forecasts of BTH region?". **Please see lines 72-77 on Page 3.**

*7. L 80: "or even become worse". Theoretical justification needed.*

**Response**: We have added references here (Yu et al., 2012; Janjic et al., 2017; Zhang et al., 2018). Theoretically, if the observations in the area where the forecast is not sensitive to the initial values are assimilated, the forecasting skills might be improved slightly or neutral. However, in realistic prediction, the imperfect procedure of data assimilation, the observation errors, the unresolved scales and processes in the model and other combination effects may induce more additional errors (Janjic et al., 2017), which may cause the fact that assimilating observations in the area where the forecast is not sensitive to the initial values results in a worse forecast. To make the sentence clear and precise, we deleted the word "Theoretically". We revised the sentence as "When the observations in the area with high sensitivity are assimilated to the initial values of the forecast, the forecasting skills will be greatly increased; conversely, if the observations in the area where the forecast is not sensitive to the initial values are assimilated, the forecasting skills will be improved slightly or even become worse (Yu et al., 2012; Janjić et al., 2018; Zhang et al., 2018).". **Please see Lines 84-85 on Page 3.**

*8. L 150: Should be mentioned here that M is WRF and not the CTM, not only at line 172.*

**Response:** We thank your suggestions. However, **on Line 155-165**, we would like to introduce the general definition of the CNOP and Eq (2) is the general mathematical expression of the CNOP. In Eq(2), M presents the nonlinear propagator and can be taken as any numerical model. When the CNOP is applied on our study specifically, M means the WRF model, as we stated on **Line 179 on Page 7**. So **on Line 159 on Page 6,** when we present the general definition of CNOP, we think it is more appropriate to define M as a nonlinear propagator.

*9. L 160: Readers might appreciate a literature reference for the energy norm eq. (3) .o*

**Response:** We thank your suggestions and have added the reference (Ehrendorfer et al., 1999) for the energy norm eq. (3) in the revised manuscript. **Please see Line 170 on Page 6.**

*10. L 177: Readers could be hinted that this is a realisation of the maximisation of an Oseledec operator, to familiarize with operator P. In fact, it is nevertheless a linear optimisation, linearized around the "nonlinear trajectory" of the model run, as the adjoint is used.*

**Response:** To compute the CNOP, we use the WRF nonlinear model to estimate the cost function and the adjoint model to produce the gradient of the cost function with respect to the perturbation. Yes, a linear assumption within the neighborhood of each point along the nonlinear trajectory is used when calculating the gradient of the cost function with initial perturbation at this point by adjoint model. However, such a linear assumption will not represent a linear optimization of the CNOP. In fact, the traditional singular vector approach commonly adopted in the previous studies is a linear optimization, which is obtained by a linearized model around the "nonlinear trajectory". The CNOP used here is obtained by running a nonlinear model, where the adjoint is only used to calculate the gradient of the cost function with respect to initial perturbations. The CNOP is a nonlinear optimal perturbation (**see the Eq. (1) and (2) on Lines 156-164, Page 6**), rather than a linear optimal perturbation (see the comparison of CNOP and singular vector in Mu et al., 2003), whose associated gradient can be exactly derived from adjoint model.

*11. L 183: It is pertinent to provide a map of BTH model domain with observation sites here at latest.*

**Response**: We thank your suggestions and have added the map of BTH model domain with observation sites at latest in the revised manuscript (also see Figure R3). **Please see Figure 1 on the page 8 in the revised manuscript.**

[Figure]

Figure R3 The map of current environmental monitoring stations (hollow circles) within the BTH domain. The black line presents the boundary of province in China, and the thick black line presents the coastline. The boundary of the Beijing-Tianjin-Hebei region is marked in red.

**12. Fig, 1 caption : Add time instances AT and DT for discussion below by some tags for convenience.**

**Response**: We thank your suggestions and have added AT and DT by the tags for convenience (also see Figure R4). **Please see Figure 2 on Page 9.**

[Figure]

Figure R4. Time series of the dry PM$_{2.5}$ concentrations at (a) Baoding station (Hebei Province) and (b) Dongsi station (Beijing city) of observations and simulations initialized by ERA5 and GFS meteorological reanalysis data during the period between 30 November and 4 December 2017.

**13. L 265: Please give a rigorous definition of "CNOP-type error" here, where it is mentioned first! Is it that what has been described in L 297 f?**

**Response**: We thank your suggestions. Yes, it is what described in Line 297. We have added the rigorous definition of "CNOP-type error" on L 297 in the revised manuscript. **Please see Lines 297-301 on Page 11.**

*14. L368: Do you mean "differences" instead of "bias"?*

**Response:** Yes. Thanks for your suggestions. We have revised it to "differences" in the revised manuscript. **Please see Line on 399 Page 19.**

*15. L 393: On each level (located through the vertical 950, 850, 750 and 500 hPa levels), or only on the most sensitive level? So, are there 15 or 60 observations?*

**Response:** The observations are located at 4 levels, which are 950, 850, 750 and 500hPa. So there are totally 60 observations. We have clarified them in the revised manuscript. **Please see Line 424-427 on Page 20.**

*16. L461 ff: Why is this subsection reasonable, if the algorithm applied is correct, in that it infers optimal conditions? The value of the method is tested against an improvement of an control run, not against climatologically (?) selected other areas. I suggest subsection 4.3 can be omitted.*

**Response:** We think that an approach proposed based on a theory should also be verified numerically, especially by a complex model. In fact, a lot of advanced methods on predictions follow this idea to show their usefulness (Zhang et al., 2019; Feng et al., 2017). The Region-W and Region-N here were considered being important regions for $PM_{2.5}$ forecasts of BTH region in previous studies. To emphasize the sensitivity identified by CNOP-type errors, we compared the $PM_{2.5}$ forecast skills with observations deployed over the sensitive area and Region-W and Region-N. The comparison will further illustrate the usefulness of CNOP in identifying the sensitive area for targeted observation and make readers believe that the CNOP is indeed useful in identifying the sensitivity numerically, rather than only in theoretical consideration. So we would like to keep this section. **Please see Lines 520-522 on Page 25.**

*17. L 500: How is a decline possible, as the sensitivity is low? It should at least be neutral.*

**Response:** Theoretically, if the observations in the area where the forecast is not sensitive to the initial values are assimilated, the forecasting skills will be improved slightly or neutral. However, in realistic prediction, the imperfect procedure of data assimilation, the observation errors, model errors, the unresolved scales and processes in the model and other combined effects may induce additional errors (Janjic et al., 2017), which may cause the fact that assimilating observations in the area where the forecast is not sensitive to the initial values results in a worse forecast. **Please see Lines 561-568 on Page 26.**

**18. L 543: *More precisely, it should be especially assigned to stagnant conditions, where a stable layer caps the boundary layer.***

**Response**: We thank your suggestions. We have added more discussions on the stability in the revised manuscript (please see the Comment 2.2). This sentence has been rephrased as well in the revised manuscript. **Please see Lines 604-611 on Page 28, Lines 627-637 on Page 28, Lines 657-663 on Page 30.**

**19. L 560: *What is the sign: truth minus control?***

**Response:** Actually, Figure 9 has 6 subfigures. Figure 9(a, d) present the meteorological condition (including wind and temperature) in the truth run at the AT (a) and DT (d). Figure (b, e) show the meteorological conditions in the control run at the AT (b) and DT(e). Figure (c) and (f) are the forecast differences (control tun minus truth run). We have clarified them in the revised manuscript. **Please see Line 641 on Page 30.**

**20. L571: *But may increase stability. Further, the interpretation of observed PM values must be supported by information of being dry aerosols or with water component included. The discussion presented should be attentive to that. Otherwise the conclusions may be false.***

**Response:** We thank your valuable suggestions. We have added more discussions on the stability in the revised manuscript (see Comment 2.2). **Please see Lines 604-611 on Page 28, Lines 627-637 on Page 28, Lines 657-663 on Page 30.** In addition, the observed $PM_{2.5}$ and $PM_{2.5}$ simulations are both for dry mass concentrations. We have clarified it in the revised manuscript. **Please see Lines 143 on Page 5 and 196 on Page 7.** Though the hygroscopic growth of aerosol particle would not affect the mass value. However, the relative humidity may influence the aerosol components by the processes such as the heterogeneous chemistry and gas-particle partitioning.

**21. 2L640: *What is the "vertical integer of CNOP-type errors"?***

**Response:** We are sorry for the typo. We mean the "vertical integral". We have revised it. **Please see Line 711 on Page 33.** The vertical integral of the CNOP-type errors is explained **on Line 366-376 on Page 18** in detail.

**22. L662-667: *What is the novel message of this passage there than the trivially expected?***

**Response:** The CNOP method is proposed based on an abstract concept model (Line 154-166 on Page 6). Whether it can be applied to identify the sensitive areas in a much realistic model, especially in a complex realistic model, should be verified numerically, despite it is reasonable in theory. Especially, the results obtained by a new method should be compared with the old perspectives to show its superiority.

Therefore, the comparisons between the sensitive area identified by the CNOP and the Region-W (Region-N), which of the latter are considered being important regions in the previous studies, will further show the superiority of CNOP-type errors in identifying the sensitive area of meteorological initial fields on $PM_{2.5}$ forecasts. In fact, a lot of advanced methods on predictions follow this idea to show their superiority. This is why we made this kind of comparison in the present study. **Please see Lines 729-738 on Page 34, 757-769 on Page 35.**

*23. L 673: "formation of $PM_{2.5}$": Strictly speaking, a different local temperature and humidity dependent secondary formation of $PM_{2.5}$ must be understood, with equal gaseous precursor emissions. It appears unlikely to me, that this can substantially explain the differences given in Fig. 1. Please clarify.*

**Response:** We agree with the referee that a different local temperature and humidity has a little impact on the secondary formation of $PM_{2.5}$. And our results have also shown that the improvements in the $PM_{2.5}$ forecast skill in assimilation run are mostly attributed to dynamic and thermodynamical reasons (**please see Lines 670-671 on Page 30**). As for the "formation of $PM_{2.5}$", we admit that we used an improper word, which may mislead the referee. We have rephrased the sentence as "During the accumulation process, the control run forecasts a weaker southerly wind and a less stable boundary layer at the AT, which is unfavorable for the accumulation of $PM_{2.5}$ and finally leads to a severe underestimation of $PM_{2.5}$ at the AT." in the revised manuscript. **Please see Lines 742-747 on Page 34.**

Regarding the differences between the observations and the simulations in Figure 1, they, as discussed in Comment 1.1 (critical comments), are due to combined effect of uncertainties of meteorology forecast, emission inventory, and model itself. As for the differences between the two simulations in Figure 1, they, as argued in Comment 3 (Minor issues), are only attributed to the uncertainties in the meteorological initial fields; that is to say, the differences between initial wind, temperature, and moisture cause the substantially difference of the two simulations. **Please see Lines 222-224 on Page 9.**

*24. L 687: …"then formulates a theoretical basis to implement practical field campaigns associated with air quality forecasts". Please indicate where this can be found!*

**Response:** Sorry for this ambiguous description. From the results, we showed that the sensitive areas of meteorological initial fields associated with the $PM_{2.5}$ forecasts indeed exists; meanwhile, these sensitive areas are verified to be valid in improving $PM_{2.5}$ forecast. So the CNOP method is an effective tool to identify the sensitive areas of meteorology on $PM_{2.5}$ forecasts. These results are adequate to encourage us to implement the targeted observations of meteorological initial fields according to the CNOP sensitivity in practical field campaigns and to enhance the $PM_{2.5}$ forecasting skills, thus formulating a theoretical basis in practical field campaigns. We have

added the explanations in the revised manuscript. **Please see Lines 757-769 on Page 35.**

25. ***L 697: What does "logistical verification" mean?***

**Response:** We are sorry for the improper use of the word. We would like to present that "the sensitive areas revealed in the present study are still instructive for practical field observations of $PM_{2.5}$ forecasts because of the verifications through a series of OSSEs and reasonable physical interpretation shown in the context". We have rephrased the sentence in the revised manuscript. **Please see line 783 on Page 36.**

**References:**
Bei, N., Wu, k., Feng, T., Cao, k., Huang, R., and Long, X., and coauthors.: Impacts of meteorological uncertainties on the haze formation in Beijing-Tianjin-Hebei (BTH) during wintertime: A case study. Atmos. Chem. Phys. 17, 14579-14591, 2017.

Chen, B., M. Mu, and X. Qin.. The impact of assimilating drop windsonde data deployed at different sites on typhoon track forecasts. Mon. Wea. Rev., 141, 2669–2682, 2013.

Chen, Z., Chen, D., Zhao, C., Kwan, M., Cai, k., and coauthors.: Influence of meteorological conditions on $PM_{2.5}$ concentrations across China: A review of methodology and mechanism, Environ. Int., 139, 105558, 2020.

Duan, W., and Zhou, F.. Non-linear forcing singular vector of a two-dimensional quasi-geostrophic model. Tellus, 65(18452), 256-256, 2013.

Duan, W., Qin, X.. Application of nonlinear optimal perturbation methods in the targeting observations and field campaigns of tropical cyclones. Advances in Earth Science (in Chinese), 37(2):165-176, 2022.

Ehrendorfer, M., R. M. Errico, and K. D. Raeder. Singular-Vector Perturbation Growth in a Primitive Equation Model with Moist Physics. J. Atmos. Sci., 56, 1627-1648, 1999.

Gilliam, R. C., C. Hogrefe, J. M. Godowitch, S. Napelenok, R. Mathur, and S. T. Rao. Impact of inherent meteorology uncertainty on air quality model predictions, J. Geophys. Res. Atmos., 120, 12,259–12,280, 2015.

Janjić, T, Bormann, N, Bocquet, M, Carton, JA, Cohn, SE, Dance, SL, Losa, SN, Nichols, NK, Potthast, R, Waller, JA, Weston, P. On the representation error in data assimilation, Q J R Meteorol Soc. 144: 1257– 1278, 2018.

Li, k., Wang, Z. F., Akimoto, H., Gao, C., Pochanart, P., Wang, X. Q.: Modeling study of ozone seasonal cycle in lower troposphere over East Asia. k. Geophys. Res: Atmos. 112, D22S25, 2007.

Liu, T., Gong, S., He, J., Yu, M., Wang, Q., Li, H., Liu, W., Zhang, J., Li, L., Wang, X., Li, S., Lu, Y., Du, H., Wang, Y., Zhou, C., Liu, H., and Zhao, Q. Attributions of meteorological and emission factors to the 2015 winter severe haze pollution episodes in China's Jing-Jin-Ji area, Atmos. Chem. Phys., 17, 2971–2980, 2017.

Lorenz, E.N.: A study of the predictability of a 28-variable atmospheric model. Tellus, 17, 321-333, 1965.

Masutani, M. et al. (2010). Observing System Simulation Experiments. In: Lahoz, W., Khattatov, B., Menard, R. (eds) Data Assimilation. Springer, Berlin, Heidelberg. https://doi.org/10.1007/978-3-540-74703-1_24

Mu, M., Duan, W. S., and Wang, B.: Conditional nonlinear optimal perturbation and its applications. Nonlinear Process Geophys., 10: 493–501, 2003.

Mu, M., Zhou, F. F., and Wang, H. L.. A method for identifying the sensitive areas in targeted observations for tropical cyclone prediction: Conditional nonlinear optimal perturbation, Monthly Weather Review, 137(5), 1623-1639, 2009.

Mu, M., W.-S. Duan, Q. Wang, and R. Zhang. An extension of conditional nonlinear optimal perturbation approach and its applications, Nonlin. Processes Geophys., 17(2), 211-220, 2010.

Park, S. K. , Xu, L.  Data assimilation for atmospheric, oceanic and hydrologic applications. Springer Press, 2016.

Privé, N., Errico, R. M. . Some General and Fundamental Requirements for Designing Observing System Simulation Experiments (OSSEs), 2018.

Qin. X., W. Duan, and M. Mu. Conditions under which CNOP sensitivity is valid for tropical cyclone adaptive observations. Quart. J. Roy. Meteor. Soc., 139, 1544–1554, 2013.

Qin, X., Duan, W., Chan, P., Chen, B., Huang, K. Effects of dropsonde data and CNOP sensitivity on the tropical cyclones forecasts in the field campaigns over the western North Pacific in 2020. Adv Atmos Sci., accepted, 2022.

Qin X., Duan, W., Mu, M.. Conditions under which CNOP sensitivity is valid for tropical cyclone adaptive observations. Q. J. R. Meteorol. Soc., 139, 1544–1554, https://doi.org/10.1002/qj.2109, 2013.

Snyder, C. Summary of an informal workshop on adaptive observations and FASTEX. *Bull Am Meteorol Soc,* **77**:   953–61, 1996.

Yu, Y., Mu, M., Duan, W., Gong, T.. Contribution of the location and spatial pattern of initial error to uncertainties in El Niño predictions. Journal of Geophysical Research, 117, C06018, 2012.

Zhang, F., Bei, N., Nielsen-Gammon, k. W., Li, G., Zhang, R., Stuart, A. L., and Aksoy, A.: Impacts of meteorological uncertainties on ozone pollution predictability estimated through meteorological and photochemical ensemble forecasts, J. Geophys. Res., 112, D04304, 2007.

Zhang, H., Wang, Y., Hu, J., Ying, Q., Hu, X.M.. Relationships between meteorological parameters and criteria air pollutants in three megacities in china. Environ.Res. 140, 242–254. 2015.

Zhang, H., Yuan, H., Liu, X., Yu, J., Jiao, Y.. Impact of synoptic weather patterns on 24 h-average $PM_{2.5}$ concentrations in the North China Plain during 2013–2017. Sci. Total Environ. 627, 200–210, 2018.

Zhang, K. , Mu, M. , Wang, Q. , Yin, B. , Liu, S. . CNOP-based adaptive observation network designed for improving upstream kuroshio transport prediction. *Journal of Geophysical Research: Oceans, 124*, 4350-4364, 2019.

**Response to Reviewer #2:**

We would like to thank you for reviewing the manuscript and providing the valuable comments. We have updated our manuscript following the suggestions. Below we answer the specific comments point by point. For readability the comments are shown in bold and italics.

**Review comments:**

*The study made the first attempt to apply the new observation strategy "target observation" to improve the air quality forecasts. A new approach of conditional nonlinear optimal perturbation (CNOP) was applied to find the sensitive area for targeting observations associated with the* $PM_{2.5}$ *forecast of a heavy haze event that occurred in the Beijing-Tianjin-Hebei region. Then several OSSEs, with different lead times and observation distances, were designed to illustrate the sensitivity of the target observations. They also evaluate this new observation strategy through the comparison with other observation strategies revealed by other studies. In addition, they provided the physical reasons why the target observation strategy can greatly improve the PM2.5 forecasts.*

*The paper is well written, clearly structured. The study provides a new perspective on understanding the sensitivity of air quality forecasts to the meteorological initial field and can serve as a theoretical guidance on practical observation tasks for* $PM_{2.5}$ *forecast. In the summary part, the authors also present a few sound recommendations for future work, which I think are worthy in-depth study and discussed. Overall this study will make a valuable contribution to the air quality studies. I recommend acceptance after addressing the issues as listed below.*

*Response:* We appreciate your encouraging comments.

*Major comments:*
1. *The authors adopted different observing distances but the same observation number to examine the role of observing distances in the sensitive areas in improving $PM_{2.5}$ forecast. It was suggested that the observation arrays of large observing distances generally play important role in improving the forecast skill of $PM_{2.5}$. Actually, it is not surprised because the observing array with larger observation distance covers larger area and more meteorological information are captured, which are then much favorable for improving PM 2.5 forecast skill. So I suggest the authors to conduct the following experiments and further examine the validity of the sensitive areas. For a given size of sensitive area, the observing arrays of different observation distances are assimilated to evaluate the role of observing distance. If the large observing distance is still much important for improving PM25 forecast (in this situation, the number of observations is much small), the original result would be assured.*

**Response:** We thank the referee's comments. In fact, in the 4 forecasts concerned in the

manuscript, not all of the observation deployments with the large distance (150km) are the optimal deployments (Table 2 and 3 in the manuscript). For the forecasts at the AT with a 24 hour lead time and the forecast at the DT with a 12 hour lead time, the observation array with the distance of 90km shows higher $PM_{2.5}$ forecast skills than those of 150km. So the observing array with a larger observation distance will not necessarily lead to higher forecast skills.

Moreover, we also conducted the following experiment as the referee suggested. Specifically, we first select a number of 120 most sensitive grids, according to the VI value in the four forecasts. For the given size of the sensitive area, the observations with the distance of 30, 60, 90, 120 and 150km are assimilated. In this situation, the number of the observations differs at different observing distances. The $AE_V/AE_M$ of the forecast at the AT and DT with lead times of 24 and 12 hours are shown in Table R1.

For the forecasts at the AT with lead times of 24 and 12 hours, the observations with a distance of 30km shows the largest improvement in both $AE_V/AE_M$. It implies that in the given size of sensitive area, denser observation sites can better resolve the synoptic initial conditions within the sensitive area, which in turn enhance the forecasting skills more effectively. In detail, the improvements of $AE_V/AE_M$ become slightly as the observation number increases from around 15 (90km) to 120 (30km) in the two forecasts, indicating that adding more observation sites only results in a small additional benefit.

For the forecast at the DT, the observations with the distance of 30km also shows the largest improvement with the lead time of 24 hours. However, when the lead time is reduced to 12 hours, the observations with the position distance of 90km show the largest improvement. It implies that in this forecast, an appropriate observation distance is much important for improving the $PM_{2.5}$ forecasts.

Thus, it is suggested that, if we have a fixed number of observation equipment, an appropriate observing distance is essential to obtain the largest improvement of forecast skills. The observations with the large distance, which cover large areas, will not necessarily lead to higher forecast skills. If we have adequate observation equipment and the observations should be deployed in a given size of area, the observations should also be deployed carefully with an appropriate distance to get the largest benefits. The relevant discussions have been added in the revised manuscript. **Please see Lines 492-513 on Page 24.**

Table R1 The $AE_V/AE_M$ of the forecasts at the AT and DT with lead times of 24 and 12 hours, when the additional observations in the sensitive region (CNOP) are assimilated. The respective optimal observation array is marked in bold.

| Process | Lead Times | 30 km | 60 km | 90 km | 120 km | 150 km |
|---|---|---|---|---|---|---|
| Accu | 24 hour | **22.98/33.94** | 20.85/29.95 | 19.95/26.59 | 14.31/26.00 | 11.87/23.28 |
| | 12 hour | **46.50/57.62** | 43.09/54.12 | 42.98/51.88 | 40.87/49.24 | 40.72/48.00 |
| Diss | 24 hour | **58.95/49.81** | 55.18/47.41 | 51.34/44.37 | 47.28/41.66 | 42.26/41.07 |

| | 12 hour | 29.58/39.60 | 27.57/37.27 | **31.48/40.01** | 23.22/32.35 | 19.52/26.36 |

2. *Section 5, Line 593-597, the interpretations for the improvements during the accumulation process is a bit weak. Actually, there are two areas identified as sensitive areas for the forecasts at the AT. One lies in the south of BTH, the other is located at central Inner Mongolia. What role did each area play on improving the* $PM_{2.5}$ *of BTH? Are there any relation between the meteorological field on these two areas? Such details are needed to be addressed and will help understand the meaning of sensitive areas.*

**Response:** We thank your comments. We have added more discussions on the stability. **Please see Lines 627-637 on Page 28, Lines 656-663 on Page 30.** To detect each role of the meteorological initial conditions of the two sensitive areas on the $PM_{2.5}$ forecasts, we assimilated the same number of meteorological observations with the same observing distance in the two areas separately. When we only assimilated the observations in the sensitive area near the Dezhou city, which lie to the southeast of Hebei province, the forecast error of $PM_{2.5}$ decreased by 5.49% measured by $AE_V$ and 16.02% measured by $AE_M$. The assimilation run increases the southerly wind component by 0.05m/s and increases the temperature by 0.1°C at the AT over the BTH region. When we assimilated the observations in the sensitive area near the central Inner Mongolia, the values of $AE_V$ and $AE_M$ are 14.00% and 22.08%, respectively. The assimilation run increases the southerly wind component by 0.16m/s and increases the temperature by 0.21°C at the AT. So assimilating the observations in each of the two areas will result in an increase of $PM_{2.5}$ forecast skills. The sensitive area near the Inner Mongolia plays a more dominant role on the $PM_{2.5}$ forecast of BTH region, by inducing a larger southerly wind component.

We think it is hard to quantify the relations between the meteorological fields over these two regions. When we assimilated the observations on each of the two regions, only the local meteorological condition is improved. Two areas are defined as the sensitive areas because there are two sources of initial errors contributing to the forecast errors of BTH. The role of the north sensitive area is to weaken the northerly wind and the role of the south sensitive area is to strengthen the southerly wind. They both increase the southerly wind component of BTH region, which is helpful for transporting southern pollution to the BTH region in the control run.

**Minor comments:**
1. *The "PM 2.5 concentration" in the whole paper means "*$PM_{2.5}$ *surface air concentrations" (PM 2.5 can be aloft). Please define "PM 2.5 concentration" as "surface air concentrations of PM 2.5" when it is first appeared.*
   **Response:** We have defined the "PM 2.5 concentration" as "surface air concentrations of PM 2.5" when it is first appeared in the revised manuscript. **Please see Line 130 on Page 5.**

2. *Line 40, "relative moisture" is few used. Modify it to "relative humidity".*
   **Response:** We have modified the "relative moisture" to "relative humidity" in the revised manuscript. **Please see Line 41 on Page 2.**

3. *Figure 2-5, the color bars of T and QVAPOR are too small. Please modify.*
   **Response:** We have modified the size of color bars in the revised manuscript. **Please see Figure 3-6.**

4. *Line 75, "assimilating more observations may not lead to higher forecast benefits". References are needed.*
   **Response:** We have added the references (Yu et al., 2012; Janjić et al., 2018; Zhang et al., 2019) in the revised manuscript. **Please see Lines 84-85 on Page 3.**

5. *Line 339-343, this is not clear to me. Please rephrase it.*
   **Response:** We have rephrased it in the revised manuscript. "In this situation, the $PM_{2.5}$ forecast could be very sensitive to the combined effect of initial errors of the meteorological fields in the area with larger VI, and preferentially reducing the meteorological initial errors in these sensitive areas will lead much larger improvements of meteorological forecasts over the BTH region, then significantly improve the $PM_{2.5}$ forecasts." **Please see Lines 372-376 on Page 18.**

6. *Line 585. Clarify which observation array in CNOP-EXP is used when comparing the forecast differences between the CNOP-EXP and control run.*
   **Response:** We have clarified the observation array in the revised manuscript. **Please see Line 641 on Page 29.**

**References:**
Janjić, T, Bormann, N, Bocquet, M, Carton, JA, Cohn, SE, Dance, SL, Losa, SN, Nichols, NK, Potthast, R, Waller, JA, Weston, P. On the representation error in data assimilation, Q J R Meteorol Soc. 144: 1257– 1278, 2018.

Yu, Y., Mu, M., Duan, W., Gong, T.. Contribution of the location and spatial pattern of initial error to uncertainties in El Niño predictions. J. Geophy. Res., 117, C06018, https://doi.org/10.1029/2011JC007758, 2012

Zhang, K., Mu, M. , Wang, Q. , Yin, B. , Liu, S. . CNOP-based adaptive observation network designed for improving upstream kuroshio transport prediction. *Journal of Geophysical Research: Oceans, 124*, 4350-4364, 2019.